# What Drives Success in Physical Planning with Joint-Embedding Predictive World Models?

**Basile Terver**                                                                  *basile@amilabs.xyz*
*Meta FAIR*
*Inria Paris*

**Tsung-Yen Yang**                                                          *jimmytyyang@meta.com*
*Meta FAIR*

**Jean Ponce**                                                                      *jean.ponce@ens.fr*
*Ecole normale supérieure / PSL*
*New York University*

**Adrien Bardes**                                                                  *abardes@meta.com*
*Meta FAIR*

**Yann Le Cun**                                                                  *yann.lecun@nyu.edu*
*New York University*

**Reviewed on OpenReview:** *https://openreview.net/forum?id=cHZn5Gdh8e*

## Abstract

A long-standing challenge in AI is to develop agents capable of solving a wide range of physical tasks and generalizing to new, unseen tasks and environments. A popular recent approach involves training a world model from state-action trajectories and subsequently using it with a planning algorithm to solve new tasks. Planning is commonly performed in the input space, but a recent family of methods has introduced planning algorithms that optimize in the learned representation space of the world model, with the promise that abstracting irrelevant details yields more efficient planning. In this work, we characterize models from this family as JEPA-WMs and investigate the technical choices that make algorithms from this class work. We propose a comprehensive study of several key components with the objective of finding the optimal approach within the family. We conducted experiments using both simulated environments and real-world robotic data, and studied how the model architecture, the training objective, and the planning algorithm affect planning success. We combine our findings to propose a model that outperforms two established baselines, DINO-WM and V-JEPA-2-AC, in both navigation and manipulation tasks. Code, data and checkpoints are available at `https://github.com/facebookresearch/jepa-wms`.

## 1 Introduction

In order to build capable physical agents, Ha & Schmidhuber (2018) proposed the idea of a world model, that is, a model predicting the future state of the world, given a context of past observations and actions. Such a world model should perform predictions at a level of abstraction that allows training policies on top of it (Hafner et al., 2024; Mendonca et al., 2021; Guo et al., 2022) or perform planning in a sample efficient manner (Sobal et al., 2025; Hansen et al., 2024). While model-free RL (Mnih et al., 2015; Fujimoto et al., 2018; Mnih et al., 2016; Haarnoja et al., 2018; Schulman et al., 2017; Yarats et al., 2022) requires a

considerable number of samples, model-based RL (MBRL), combined with self-supervised pretraining, has led to powerful world modeling algorithms (Ha & Schmidhuber, 2018; Seo et al., 2022; Schrittwieser et al., 2020; Hafner et al., 2024; Hansen et al., 2024). More recently, large-scale world models have flourished (Hu et al., 2023; Yang et al., 2023; Brooks et al., 2024; Bruce et al., 2024; Parker-Holder et al., 2024; Bartoccioni et al., 2025; Agarwal et al., 2025; Bar et al., 2025), achieving impressive simulation accuracy in specific domains such as driving (Hu et al., 2023; Bartoccioni et al., 2025) or egocentric video games (Bruce et al., 2024; Parker-Holder et al., 2024; Ball et al., 2025).

In this presentation, we model a world in which some (robotic) agent equipped with a (visual) sensor operates as a dynamical system where the states, observations and actions are all embedded in feature spaces by parametric encoders, and the dynamics itself is also learned, in the form of a parametric predictor depending on these features. The encoder/predictor pair is what we will call a *world model*. We will focus on *action-conditioned Joint-Embedding Predictive World Models* (or *JEPA-WMs*) learned from videos (Sobal et al., 2025; Zhou et al., 2024a; Assran et al., 2025). These models adapt to the planning problem the Joint-Embedding Predictive Architectures (JEPAs) proposed by LeCun (2022), where a representation of some data is constructed by learning an encoder/predictor pair such that the embedding of one view of some data sample predicts well the embedding of a second view. We use the term JEPA-WM to refer to this family of methods, that we formalize in Equations (1) to (4) as a unified implementation recipe rather than a novel algorithm. The term JEPA-WM designates a specific recipe in which the dynamics model is trained solely through a predictive loss in embedding space, with no reconstruction, reward prediction, or value/policy heads, unlike methods such as MuZero (Schrittwieser et al., 2020), PLaNet (Hafner et al., 2019) or the Dreamer series (Hafner et al., 2020; 2021; 2024) (see Section B for a per-method comparison). Note that JEPA-WMs are not restricted to frozen encoders: PLDM (Sobal et al., 2025) and EB-JEPA (Terver et al., 2026) learn the encoder and predictor jointly. In practice, we optimize to find an action sequence without theoretical guarantees on the feasibility of the plan, which is closer to *trajectory optimization*, but we stick to the widely-used term *planning*.

Among these JEPA-WMs, PLDM (Sobal et al., 2025) shows that, on 2D navigation tasks, world models learned in a latent space, trained as JEPAs, generalize better than the GCRL baselines considered in that work (Park et al., 2025), especially on suboptimal training trajectories. DINO-WM (Zhou et al., 2024a) shows that, in absence of reward, when comparing latent world models on goal-conditioned planning tasks, a JEPA model trained on a frozen DINOv2 encoder outperforms DreamerV3 (Hafner et al., 2024) and TD-MPC2 (Hansen et al., 2024), when we deprive these of reward annotation. DINO-World (Baldassarre et al., 2025) shows the capabilities in dense prediction and intuitive physics of a JEPA-WM trained on top of DINOv2 are superior to COSMOS. The V-JEPA-2-AC (Assran et al., 2025) model is able to beat Vision Language Action (VLA) baselines like Octo (Octo Model Team et al., 2024) in greedy planning for object manipulation using image subgoals.

In this paper, we focus on the learning of the dynamics (predictor) rather than of the representation (encoder), as in DINO-WM and V-JEPA-2-AC (Zhou et al., 2024a; Assran et al., 2025). Given the increasing importance of such models, we aim at filling what we see as a gap in the literature, i.e., a thorough study answering: *how to efficiently learn a dynamics model in the embedding space of a pretrained visual encoder for manipulation and navigation planning tasks ?*

Our contributions can be summarized as follows: **(i)** We study several key components of training and planning with JEPA-WMs: multistep rollout, predictor architecture, training context length, using or not proprioception, encoder type, model size, data augmentation; and the planning optimizer. **(ii)** We use these insights to propose an optimum in the class of JEPA-WMs, outperforming DINO-WM and V-JEPA-2-AC.

## 2    Related work

**World modeling and planning.** 'A path towards machine intelligence' (LeCun, 2022) presents planning with Model Predictive Control (MPC) as the core component of Autonomous Machine Intelligence (AMI). World Models learned via Self-Supervised Learning (SSL) (Fung et al., 2025) have been used in many reinforcement learning works to control exploration using information gain estimation (Sekar et al., 2020) or curiosity (Pathak et al., 2017), to transfer to robotic tasks with rare data by first learning a world

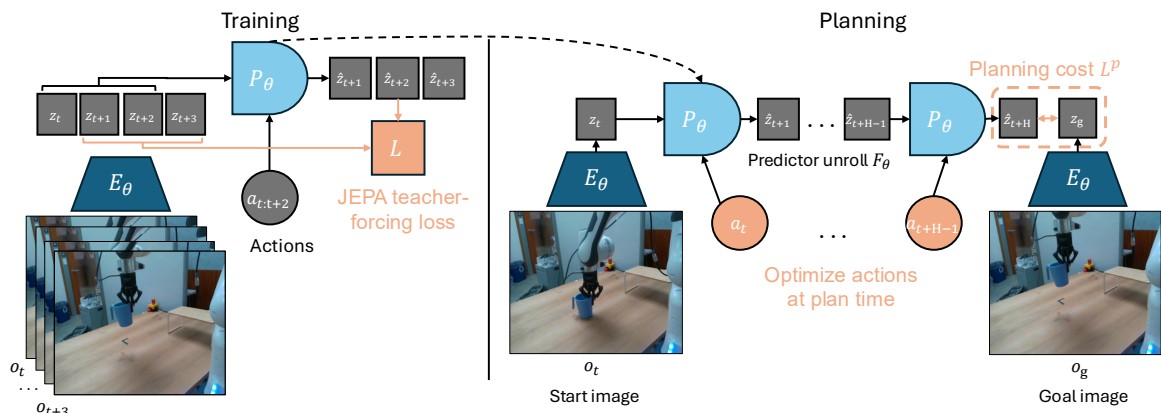

Figure 1: Left: Training of JEPA-WM: the encoder $E_{\phi,\theta}$ embeds video and optionally proprioceptive observation, which is fed to the predictor $P_\theta$, along with actions, to predict (in parallel across timesteps) the next state embedding. Right: Planning with JEPA-WM: sample action sequences, unroll the predictor on them, compute a planning cost $L^p$ for each trajectory, and use this cost to iteratively refine the action sampling. The action encoder $A_\theta$ and proprioceptive encoder $E_\theta^{prop}$ are not explicitly displayed in this figure for readability.

model (Mendonca et al., 2023) or to improve sample efficiency (Łukasz Kaiser et al., 2020). In addition, world models have been used in planning, to find sub-goals (Nair & Finn, 2020) by using the inverse problem of reconstructing previous frames to reach the objective represented as the last frame, or by imagining goals in unseen environments (Mendonca et al., 2021). World models can be generative (Brooks et al., 2024; Hu et al., 2023; Ball et al., 2025; Agarwal et al., 2025), using diffusion-based backbones to model multi-modal transition distributions, or deterministic and trained in a latent space via a JEPA loss (Garrido et al., 2024; Sobal et al., 2025; Terver et al., 2026; Assran et al., 2025; Zhou et al., 2024a; Bar et al., 2025), trading stochastic expressiveness for computational efficiency and latent abstraction. They can be used to plan in the latent space (Zhou et al., 2024a; Sobal et al., 2025; Bar et al., 2025), to maximize a sum of discounted rewards (Hansen et al., 2024; Hafner et al., 2019), or to learn a policy (Hafner et al., 2024). Other approaches for latent-space planning include locally-linear dynamics models (Watter et al., 2015), gradient-based trajectory optimization (Srinivas et al., 2018), and diffusion-based planners (Janner et al., 2022; Zhou et al., 2024b), details in Section B. While the present study focuses on the predictor given a frozen encoder, concurrent works explore lightweight adaptation of frozen VFM encoders for control, e.g. by training a bisimulation-based encoder on top of the frozen backbone (Toso et al., 2026), by learning sparse autoencoders on frozen features (Zhao et al., 2026), or by decoupling dynamics-relevant from dynamics-irrelevant representations (Yin et al., 2026). These directions are complementary to our predictor-focused investigation.

**Explicit and implicit world models.** Our study focuses on *explicit* world models with autoregressive dynamics in latent space. *Implicit* alternatives such as TD-JEPA (Bagatella et al., 2026) fold temporal abstraction into the representation via successor features; we compare both paradigms in Section B.

**Goal-conditioned RL.** Goal-conditioned RL (GCRL) offers a self-supervised approach to leverage large-scale pretraining on unlabeled (reward-free) data. Foundational methods show that goal-conditioned policies can be incorporated into planning (Nasiriany et al., 2019; Li et al., 2022), decomposed hierarchically with sub-goal generators (Fang et al., 2022a;b), or combined with offline RL (Shah et al., 2021; Xu et al., 2023; Park et al., 2023). More recent methods learn geometrically grounded distances (Park et al., 2024; Myers et al., 2025), and the OGBench benchmark (Park et al., 2025) provides a systematic evaluation of offline GCRL across locomotion and manipulation.

**Robotics.** Classical approaches rely on MPC loops (Garcia et al., 1989; Borrelli et al., 2017) leveraging analytical models of the robot Chignoli et al.; Meduri et al. (2022). For exteroception, our study relies on a camera, akin to the visual servoing problem Hutchinson et al. (1996). The current state-of-the-art in manipulation has been reached by Vision-Language-Action (VLA) models such as RT-X (Vuong et al., 2023), RT-1 (et al., 2023), and RT-2 (Zitkovich et al., 2023). Physical Intelligence's $\pi$ series (Black et al.,

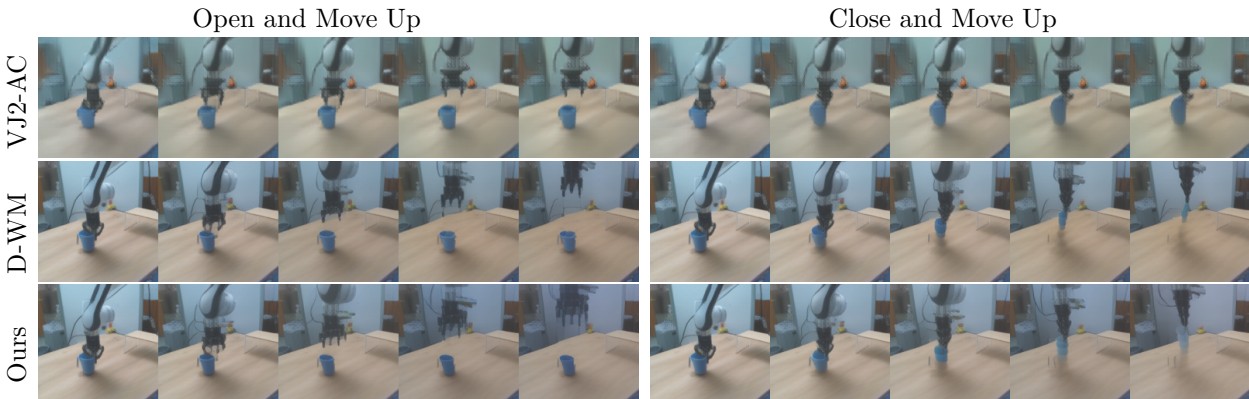

Figure 2: Comparison of different methods on the counterfactual Franka arm lift cup task, where we hardcode 2 actions, either "open and move up" or "close and move up". Each shows 5 model actions in open-loop rollout. Left: "open and move up" action. Right: "close and move up". First row: V-JEPA-2-AC. Second row: DINO-WM. Third row: our best model, described in Section 5.3.

2024; Intelligence et al., 2025b;a) uses flow matching on the Open-X embodiment dataset to generate action trajectories.

## 3 Background

This section formalizes the common setup of JEPA-WMs learned from pretrained visual encoders, but does not introduce novel methods. We summarize JEPA-WM training and planning in Figure 1. A consolidated notation reference is provided in Section A.

**Training method.** In a JEPA-WM, we embed the observations with a frozen visual encoder $E_\phi^{vis}$, and an (optional) shallow proprioceptive encoder $E_\theta^{prop}$. Applying each encoder to the corresponding modality constitutes the global state encoder, which we denote $E_{\phi,\theta} = (E_\phi^{vis}, E_\theta^{prop})$. An action encoder $A_\theta$ embeds the robotic actions. On top of these, a predictor $P_\theta$ takes both the state and action embeddings as input. $E_\theta^{prop}$, $A_\theta$ and $P_\theta$ are jointly trained, while $E_\phi^{vis}$ remains frozen. For a past window of $w$ observations $o_{t-w:t} := (o_{t-w}, \dots, o_t)$ including visual and (optional) proprioceptive input and past actions $a_{t-w:t}$, their common training prediction objective on $B$ elements of the batch is

$$\mathcal{L} = \frac{1}{B} \sum_{b=1}^{B} L[P_\theta\left(E_{\phi,\theta}(o_{t-w:t}^b), A_\theta(a_{t-w:t}^b)\right), E_{\phi,\theta}\left(o_{t+1}^b\right)], \tag{1}$$

where $L$ is a loss, computed pairwise between visual prediction and target, and proprioceptive prediction and target. In our experiments, we chose $L$ as the Mean Squared Error (MSE). The architecture chosen for the encoder and predictor in this study is ViT (Dosovitskiy et al., 2021), as in our baselines (Zhou et al., 2024a; Assran et al., 2025). In DINO-WM (Zhou et al., 2024a), the action and proprioceptive encoder are just linear layers, and their output is concatenated to the visual encoder output along the embedding dimension, which is known as *feature conditioning* (Garrido et al., 2024), as opposed to *sequence conditioning*, where the action and proprioception are encoded as tokens, concatenated to the visual tokens sequence, which is adopted in V-JEPA-2 (Assran et al., 2025). We stress that $P_\theta$ is trained with a frame-causal attention mask, thus, it is simultaneously trained to predict from all context lengths from $w = 0$ to $w = W - 1$, where $W$ is a training hyperparameter, set to $W = 3$. The causal predictor is trained to predict the outcome of several actions instead of one action only. To do so, one can skip $f$ observations and concatenate the $f$ corresponding actions to form an action of higher dimension $f \times A$, as in DINO-WM (Zhou et al., 2024a). More details on the training procedure in Section C.

**Planning.** Planning at horizon $H$ is an optimization problem over the product action space $\mathbb{R}^{H \times A}$, where each action is of dimension $A$, which can be taken to be $f \times A$ when using frameskip at training time. Given an initial and goal observation pair $o_t, o_g$, each action trajectory $a_{t:t+H-1} := (a_t, \dots, a_{t+H-1})$ should be evaluated with a planning objective $L^p$. Like at training time, consider a dissimilarity metric $L$, (e.g. the

Table 1: Summary of all candidate design choices studied in this work and recommended recipe per task category. Each row lists the options evaluated; the two rightmost columns report the empirically best option for simulated navigation and real-world manipulation, respectively.

| Component | Candidates | Best: Simu. Nav. | Best: Real Manip. |
|---|---|---|---|
| **Encoder** | | | |
| Visual encoder | DINOv2 (ViT-S/B/L), DINOv3 (L), V-JEPA (L), V-JEPA-2 (L) | **DINOv2-S** | **DINOv3-L** |
| **Predictor** | | | |
| Architecture | Feat. cond. + sincos, Seq. cond. + RoPE, Feat. cond. + RoPE, AdaLN + RoPE, AdaLN-zero + RoPE | **AdaLN + RoPE** | **AdaLN + RoPE** |
| Depth | 3, 6, 9, 12 | **6** | **12** |
| **Training** | | | |
| Rollout steps | 1-step (teacher forcing), 2-step, 3-step, 6-step | **2** | **6** |
| Context length ($W$) | 1, 2, 3, 5, 7, 9, 14 | **3** | **5** |
| Proprioception | With / Without | ✓ | ✓ |
| **Planning** | | | |
| Optimizer | CEM, NG (NeverGrad), Adam, GD | **CEM** | **CEM** |
| Cost | $L_1$, $L_2$ | $\mathbf{L_2}$ | $\mathbf{L_2}$ |

$L_1$, $L_2$ distance or minus the cosine similarity), applied pairwise on each modality, denoted $L_{vis}$ between two visual embeddings and $L_{prop}$ for proprioceptive embeddings. When planning with a model trained with both proprioception and visual input, given $\alpha \geq 0$, the planning objective $L_\alpha^p$ we aim to minimize is

$$L_\alpha^p(o_t, a_{t:t+H-1}, o_g) = (L_{vis} + \alpha L_{prop})(G_{\phi,\theta}(o_t, a_{t:t+H-1}), E_{\phi,\theta}(o_g)), \qquad (2)$$

with a function $G_{\phi,\theta}$ depending on our world model. We define recursively $F_{\phi,\theta}$ as the unrolling of the predictor from $z_t = E_{\phi,\theta}(o_t)$ on the actions, with a maximum context length of $w$, (fixed to $W^t$ at train time and to $W^p$ at test time Table 10)

$$F_{\phi,\theta} : (o_t, a_{t-w:t+k-1}) \mapsto \hat{z}_{t+k}, \qquad (3)$$
$$\hat{z}_{i+1} = P_\theta(\hat{z}_{i-w:i}, A_\theta(a_{i-w:i})), \quad i = t, \ldots, t+k-1, \quad z_t = E_{\phi,\theta}(o_t) \qquad (4)$$

In our case, we take $G_{\phi,\theta}$ to be the unrolling function $F_{\phi,\theta}$, but could choose $G_{\phi,\theta}$ to be a function of all the intermediate unrolling steps, instead of just the last one. We provide details about the planning optimizers in Section F.

## 4 Studied design choices

Our base configuration is DINO-WM without proprioception, with a ViT-S encoder and depth-6 predictor of same embedding dimension. A summary of all candidate design choices is provided in Table 1. We prioritize design choices based on their scope of impact: planning-time choices affect all evaluations, so we optimize these first and *fix the best planner for each environment for the subsequent experiments*; training and architecture choices follow; scaling experiments validate our findings. Each component is independently varied from the base configuration to isolate its effect.

**Planner.** Various optimization algorithms can be relevant to solve the problem of minimizing equation 2, which is differentiable. Zhou et al. (2024a); Hansen et al. (2024); Sobal et al. (2025); Assran et al. (2025); Bar et al. (2025) use the Cross-Entropy Method (CEM) (or a variant called Model Predictive Path Integral (MPPI) (Williams et al., 2015)), depicted in Section F. Since this is a population-based optimization method which does not rely on the gradient of the cost function, we introduce a planner that can make use of any of the optimization methods from NeverGrad (Bennet et al., 2021). For our experiments, we choose the default NGOpt optimizer (Anonymous, 2024), which is designated as a "meta"-optimizer. We do not tune any of the parameters of this optimizer. We denote this planner NG in the remainder of this paper, see details in Section F. We also experiment with gradient-based planners (GD and Adam) that directly optimize the action sequence through backpropagation, see details in Section F. The planning hyperparameters common to the four considered optimizers are those which define the predictor-dependent cost function $G_\theta$, the planning

horizon $H$, the number of actions of the plan that are stepped in the environment $m \leq H$, the maximum sliding context window size of past predictions fed to the predictor, denoted $W^p$. The ones common to either CEM and NG or to Adam and GD are the number of candidate action trajectories of which we evaluate the cost in parallel, denoted $N$, and the number of iterations $J$ of parallel cost evaluations. After some exploration of the impact of planning hyperparameters common to both CEM and NG on success, we fix them to identical values for both, as summarized in Table 10 in appendix. We plan using either the $L_1$ or $L_2$ embedding space distance as dissimilarity metric $L$ in the cost $L_\alpha^p$. The results in Figure 3 (left) are an average across the models considered in this study.

**Multistep rollout training.** At each training iteration, in addition to the frame-wise teacher forcing loss of equation 1, we compute additional loss terms as the $k$-step rollout losses $\mathcal{L}_k$, for $k \geq 1$, defined as

$$\mathcal{L}_k = \frac{1}{B} \sum_{b=1}^{B} L[P_\theta(\hat{z}^b_{t-w:t+k-1}, A_\theta(a^b_{t-w:t+k-1})), E_{\phi,\theta}(o^b_{t+k})], \tag{5}$$

where $\hat{z}^b_{t+k-1} = F_{\phi,\theta}(o_t, a_{t-w:t+k-2})$, see equation 3. We note that $\mathcal{L}_1 = \mathcal{L}$. In practice, we perform truncated backpropagation over time (TBPTT) (Elman, 1990; Jaeger, 2002), which means that we discard the accumulated gradient to compute $\hat{z}_{t+H}$ and only backpropagate the error in the last prediction. We study variants of this loss, as detailed in Section C, including the one used in V-JEPA-2-AC. We denote the model trained with a sum of loss terms up to the $\mathcal{L}_k$ loss as $k$-step. We train models with up to a 6-step loss, which requires more than the default $W = 3$ maximum context size, hence we set $W = 7$ to train them, similarly to the models with increased $W$ introduced afterwards.

**Proprioception.** We compare the standard setup of DINO-WM (Zhou et al., 2024a), where we train a proprioceptive encoder jointly with the predictor and the action encoder to a setup with visual input only. We stress that, contrary to V-JEPA-2-AC, we use both the visual and proprioceptive loss terms to train the predictor, proprioceptive encoder and action encoder.

**Training context size.** We aim to test whether allowing the predictor to see a longer context at train time allows to better unroll longer sequences of actions. We test values from $W = 1$ to $W = 14$.

**Encoder type.** As posited by Zhou et al. (2024a), local features preserve spatial details that are crucial to solve the tasks at hand. Hence we use the local features of DINOv2 and the recently proposed DINOv3 (Siméoni et al., 2025), even stronger on dense tasks. We train a predictor on top of video encoders, namely V-JEPA (Bardes et al., 2024) and V-JEPA-2 (Assran et al., 2025). We consider their ViT-L version. After exploration of the frame encoding strategy to adopt Section C, we settle on the highest performing one, which consists in duplicating each of the $o_{t-W+1}, \ldots, o_{t+1}$ frames and encoding each pair independently as a 2-frame video. Details comparing the encoding methods for all encoders considered are in Section C. The frame preprocessing and encoding is equalized to have the same number of visual embedding tokens per timestep, so the main difference lies in the weights of these encoders that we use out-of-the-box.

**Predictor architecture.** The main difference between the predictor architecture of Zhou et al. (2024a), and the one of Assran et al. (2025), is that the first uses feature conditioning, with sincos positional embedding, whereas the latter performs sequence conditioning with RoPE (Su et al., 2024). In the first, action embeddings $A_\theta(a)$ are concatenated with visual features $E_\theta(o)$ along the embedding dimension, and the hidden dimension of the predictor is increased from $D$ to $D + f_a$, with $f_a$ the embedding dimension of actions. The features are then processed with 3D sincos positional embeddings. In the second, actions are encoded as separate tokens and concatenated with visual tokens along the sequence dimension, keeping the predictor's hidden dimension to $D$ (as in the encoder). Rotary Position Embeddings (RoPE) is used at each block of the predictor. We also test an architecture mixing feature conditioning with RoPE. Another efficient conditioning technique is Adaptive Layer Normalization (AdaLN) (Xu et al., 2019), as adopted by Bar et al. (2025), which we also put to the test, using RoPE in this case. This approach allows action information to influence all layers of the predictor rather than only at input, potentially preventing vanishing of action information through the network. We also study the AdaLN-zero variant (Peebles & Xie, 2023), which initializes the

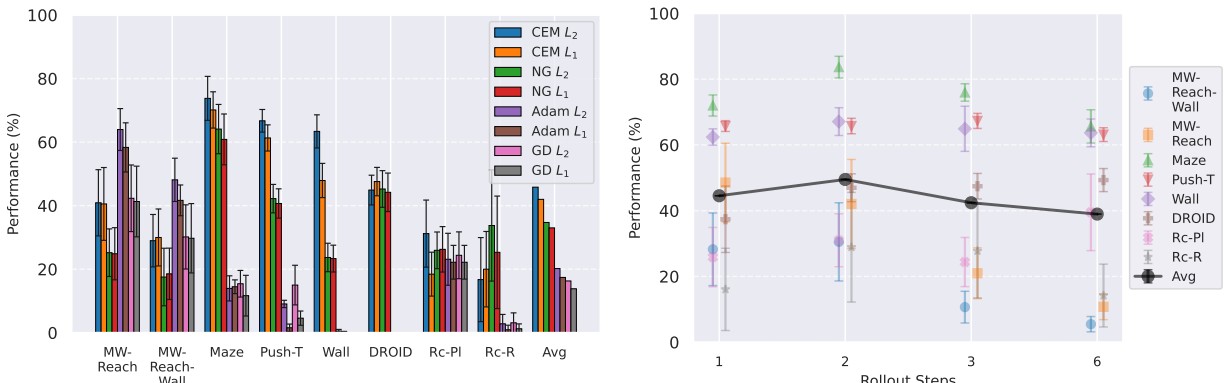

Figure 3: (a) Comparison of planning optimizers: NG is the Nevergrad-based interface that we introduce, compared to the Cross-Entropy Method (CEM), Adam, and Gradient Descent (GD), with $L_1$ or $L_2$ distance. (b) Effect of adding multistep rollout loss terms: models are trained with total loss $\mathcal{L}_1 + \cdots + \mathcal{L}_k$. Rc-Pl and Rc-R denote the Place and Reach tasks of Robocasa.

conditioning MLP to output the zero-vector, so that the predictor behaves like an unconditional ViT block at the beginning of training. Details are provided in Section C.

**Model and data scaling.** We increase the encoder size to ViT-B and ViT-L, using DINOv2 ViT-B and ViT-L with registers (Darcet et al., 2024). When increasing encoder size, we expect the prediction task to be harder and thus require a larger predictor. Hence, we increase accordingly the predictor embedding dimension to match the encoder. We also study the effect of predictor depth, varying it from 3 to 12. Regarding data scaling, we ablate the impact of these design choices on the sample-efficiency of JEPA-WMs, by training on 2%, 10%, 50% or 100% of the available training data. We compare our optimal JEPA-WM, detailed in Section 5.3 below, to the DINO-WM and V-JEPA-2-AC baselines for each of the data scale regimes considered.

## 5 Experiments

### 5.1 Evaluation Setup.

**Datasets.** For Metaworld, we gather a dataset by training TD-MPC2 (Hansen et al., 2024) online agents and evaluate two tasks, "Reach" and "Reach-Wall", denoted *MW-R* and *MW-RW*, respectively. We use the offline trajectory datasets released by Zhou et al. (2024a), namely Push-T (Chi et al., 2023), Wall and PointMaze. The train split represents 90% of each dataset. We train on DROID (et al., 2024) and evaluate zero-shot on Robocasa (Nasiriany et al., 2024) by defining custom pick-and-place tasks from teleoperated trajectories, namely "Place" and "Reach", denoted *Rc-Pl* and *Rc-R*. We *do not finetune* the DROID models on Robocasa trajectories. We also evaluate on a set of 16 videos of a real Franka arm filmed in our lab, closer to the DROID distribution, and denote this task *DROID*. On DROID, we track the $L_1$ error between the actions outputted by the planner and the groundtruth actions of the trajectory from the dataset that defines initial and goal state. We then rescale the opposite of this *Action Error*, to constitute the *Action Score*, a metric to maximize. We provide details about our datasets and environments in Section E.

**Goal definition.** We sample the goal frame from an expert policy provided with Metaworld, from the dataset for Push-T, DROID and Robocasa, and from a random 2D state sampler for Wall and Maze, more details in Section E. For the models with proprioception, we plan using proprioceptive embedding distance, by setting $\alpha = 0.1$ in equation 2, except for DROID and Robocasa, where we set $\alpha = 0$, to be comparable to V-JEPA-2-AC.

**Metrics.** The main metric we seek to maximize is success rate, but track several other metrics, that track the world model quality, independently of the planning procedure, and are less noisy than success rate. These metrics are embedding space error throughout predictor unrolling, proprioceptive decoding error throughout unrolling, visual decoding of open-loop rollouts (and the Learned Perceptual Image Patch

Similarity (LPIPS) (Zhang et al., 2018) between these decodings and the groundtruth future frames). More details in Section G.2.

**Statistical significance.** To account for training variability, we train with 3 seeds per model for our final models in Table 2. We evaluate on $e$ episodes per epoch ($e = 96$ for most environments, $e = 64$ for DROID, $e = 32$ for Robocasa) and average success over the last $n$ training epochs to obtain aggregate scores; full details on aggregation and error bars are provided in Section G.2.

### 5.2 Results

One important fact to note is that, even with models which are able to faithfully unroll a large number of actions, success at the planning task is not an immediate consequence. We develop this claim in Section G.1, and provide visualizations of rollouts of studied models and planning episodes.

**Comparing planning optimizers.** We compare four planning optimizers: Cross-Entropy Method (CEM), Nevergrad (NG), Adam, and Gradient Descent (GD). CEM is a variant of the Covariance Matrix Adaptation Evolution Strategy (CMA-ES) family (Hansen & Ostermeier, 1996; Hansen, 2023) with diagonal covariance and simplified update rules. NG uses the NGOpt wizard, which selects diagonal CMA-ES (Hansen et al., 2019) based on optimization space parametrization and budget, see Algorithm 2. We observe in Figure 3a that the CEM $L_2$ planner performs best overall.

*(i) Gradient-based methods:* Adam $L_2$ achieves the best overall performance on Metaworld, outperforming all other optimizers, and GD is also competitive with CEM. This can be explained by the nature of Metaworld tasks: they have relatively smooth cost landscapes where the goal is greedily reachable, allowing gradient-based methods to excel. In contrast, on 2D navigation tasks (Wall, Push-T, Maze) that require non-greedy planning, gradient-based methods perform very poorly compared to sampling-based ones, as GD gets stuck in local minima. On DROID, gradient-based methods also perform significantly worse than sampling-based approaches: these tasks require rich and precise understanding of complex real-world object manipulation, leading to multi-modal cost landscapes. Robocasa tasks, being simulated but closer in nature to Metaworld, allow gradient-based methods to perform reasonably well again.

*(ii) Sampling-based methods:* On 2D navigation tasks, CEM clearly outperforms NG, as these tasks require precise action sequences where CEM's faster convergence to tight action distributions is beneficial, while NG's slower, more exploratory optimization is detrimental. To compare both methods, we plot the convergence of the optimization procedure at each planning step in Figure 9, and observe that NG converges more slowly, indicating more exploration in the space of action trajectories. On DROID and Robocasa, CEM and NG perform similarly. When using NG, we have fewer planning hyperparameters than with CEM, which requires specifying the top-$K_e$ trajectories parameter and the initialization of the proposal Gaussian distribution $\mu^0, \sigma^0$—parameters that heavily impact performance. Crucially, on real-world manipulation data (DROID and Robocasa), NG performs on par with CEM while requiring no hyperparameter tuning, making it a practical alternative when transitioning to new tasks or datasets where CEM tuning would be costly.

On all planning setups and models, $L_2$ cost consistently outperforms $L_1$ cost. To minimize the number of moving parts in the subsequent study, we *fix the planning setup* for each dataset to CEM $L_2$, which is either best or competitive on all environments.

**Multistep rollout predictor training.** At planning time, the predictor is required to faithfully roll out an action sequence by predicting future embeddings from its previous predictions. We observe in Figure 3b that the performance increases when going from pure teacher-forcing models to 2-step rollout loss models, but then decreases for models trained in simulated environments. At train time, during the predictor rollout, the context window for rollout steps $k > 1$ is set to a maximum of $W^t = 3$ timesteps. At test time, we start the predictor unrolling from one groundtruth (visual and optionally proprioceptive) embedding and unroll the predictor for $H$ steps, with a predictor sliding context window of length $W^p = 2$, as explained in equation 4 and Section C. Because the predictor $P_\theta$ is a neural network with Lipschitz constant $\Lambda \geq 1$ in general (Miyato et al., 2018; Virmaux & Scaman, 2018), compounding prediction errors in continuous

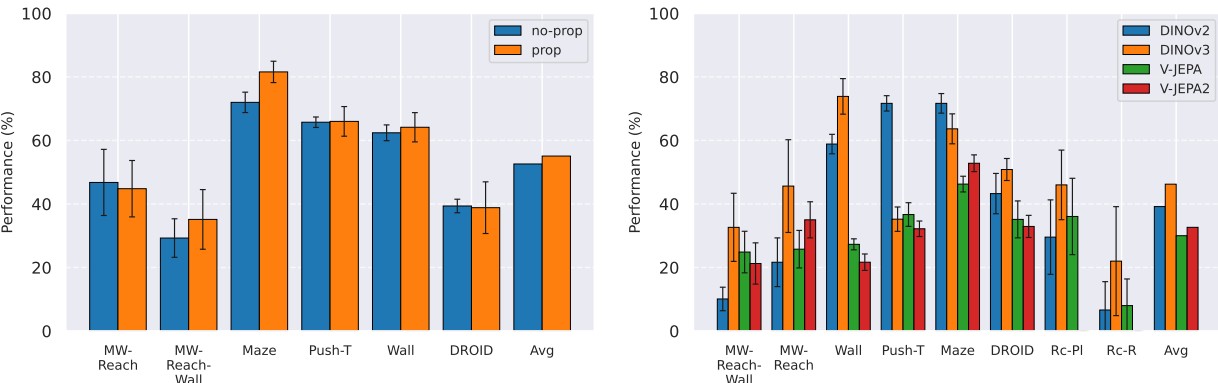

Figure 4: (a) Models trained with proprioceptive input are denoted "prop", while pure visual world models are named "no-prop". (b) Comparison of JEPA-WMs trained on top of various pretrained visual encoders,

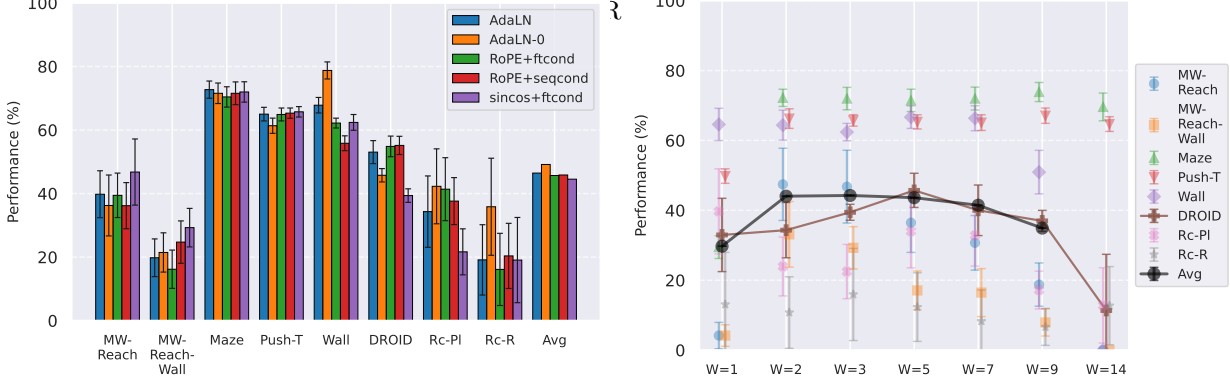

Figure 5: (a) Comparing predictor architectures: we denote positional embedding in the predictor as sincos or RoPE; the feature conditioning technique as "ftcond" and the sequence conditioning as "seqcond". The Adaptive LayerNorm conditioning technique is denoted "AdaLN". (b) Maximum number of timesteps of state embedding seen by the predictor at train time in equation 1, the predictor takes up to $(E_{\phi,\theta}(o_{t-W+1:t}), A_\theta(a_{t-W+1:t}))$ as context. Rc-Pl and Rc-R denote the Place and Reach tasks of Robocasa.

embedding space grow exponentially with the horizon $H$ (we formalize this in Section D). This creates an *accuracy-robustness tradeoff* when choosing the number of rollout steps $K$ for training: increasing $K$ raises $\delta_K$ (the one-step error on groundtruth inputs) but reduces the effective Lipschitz constant $\Lambda_K$ (see Remark 1 for a detailed analysis). The multi-step rollout loss thus acts as data augmentation against compounding error: the model learns to remain on the data manifold after several autoregressive steps, analogous to scheduled sampling (Bengio et al., 2015; Ross et al., 2011; Bengio et al., 1994). In simulated environments, the accuracy term dominates and the optimum is at small $K$; on DROID, reducing $\Lambda_K$ more than compensates the increase in $\delta_K$, and the optimal tradeoff point shifts to $K=6$.

**Impact of proprioception.** We observe in Figure 4a that models trained with proprioceptive input are consistently better than without. Visual embeddings from a frozen encoder capture appearance and coarse spatial layout, but precise metric quantities (joint positions, end-effector coordinates) are only implicitly encoded and subject to quantization by the patch-based architecture. Proprioception provides a metrically precise complement, particularly near goal states where small physical displacements yield negligible changes in embedding distance. On Metaworld, this explains the observed failure mode: without proprioception, the arm reaches the vicinity of the goal but oscillates, unable to resolve the remaining distance from vision alone. We do not display the results on Robocasa as the proprioceptive space is not aligned between DROID and Robocasa, making models using proprioception irrelevant for zero-shot transfer.

**Maximum context size.** (i) A first experiment confirms a well-known but fundamental property: the training maximum context $W$ and planning maximum context $W^p$ must be chosen so that $W^p \leq W$. Otherwise, we ask the model to perform a prediction task it has not seen at train time, and we see the predictions degrading rapidly throughout unrolling if $W^p > W$. To account for this, the $W = 1$ model

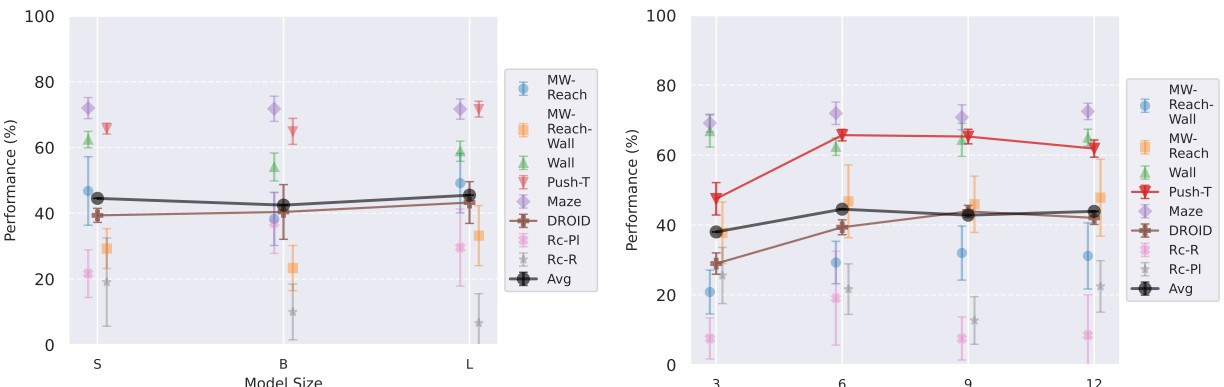

Figure 6: (a) Comparison of model size: we vary from ViT-S to ViT-L the visual encoder size, as well as the predictor embedding dimension, keeping predictor depth constant at 6. (b) Comparison of predictor depth: we vary the predictor depth from 3 to 12, keeping the encoder fixed to DINOv2-S. Rc-Pl and Rc-R denote the Place and Reach tasks of Robocasa.

performance displayed in Figure 5b is from planning with $W^p = 1$. **(ii)** We recall that we chose to plan with $W^p = 2$ in all our experiments, since it yields the maximal success rate while being more computationally efficient. The predictor needs two frames of context to infer velocity and use it for the prediction task. It requires 3 frames to infer acceleration. We indeed see in Figure 5b a big performance gap between models trained with $W = 1$ and $W = 2$, which indicates that the predictor benefits from using this context to perform its prediction. Interestingly, we observe that models trained on DROID have their optimal $W$ at 5, higher than on simulated datasets, for which it is 3. It is likely due to the more complex dynamics of DROID, requiring longer context to notably infer real-world arm and object dynamics. While longer context could in principle help capture phenomena such as object permanence or long-term momentum, occlusions are rare on DROID and Robocasa, as they can mostly occur between the arm the manipulated objects. Moreover, we sample the DROID dataset (natively at 30 fps) at 4 fps, so a training slice of $W + 1 = 8$ frames already spans over 2 seconds of video, covering most occlusion events in the dataset. **(iii)** Increasing $W$ also reduces the number of unique training slices, which, even with a fixed number of training iterations, can harm performance on small datasets (e.g. on DROID, $W = 14$ retains only 86% of videos); see Section E for details. This is in line with the observation made by Sobal et al. (2025), that world models are good at "stitching suboptimal trajectories", compared to GCRL.

**Encoder type.** In Figure 4b, we see a clear advantage of DINO encoders compared to V-JEPA encoders. We posit this is due to the well-known fact that DINO has better fine-grained object segmentation capabilities, which is crucial in tasks requiring a precise perception of the location of the agent and objects. For a frozen-encoder JEPA-WM, the predictor must learn dynamics entirely in the encoder's representation space. DINO's finer object segmentation means that distinct objects occupy distinct spatial tokens with sharp boundaries, so that object motion translates into localized, sparse token changes that the predictor can learn efficiently. Coarser segmentation, as exhibited by V-JEPA even in image mode (Section C), spreads object information across overlapping sets of tokens, making it harder for the predictor to isolate per-object dynamics. Interestingly, DINOv3 clearly outperforms DINOv2 only on the more photorealistic environments, Robocasa and DROID, likely due to the pretraining dataset of DINOv3 being more adapted to such images. On synthetic environments, DINOv2 already captures the simpler visual appearance; DINOv3's additional capacity may produce representations unnecessarily complex for the predictor, explaining why on Maze and Wall, models trained on DINOv3 take longer to converge to a lower success rate.

**Predictor architecture.** In Figure 5a, we observe that, while AdaLN with RoPE achieves the best average performance across environments, the advantage is slight, and results are task-dependent: on Metaworld, sincos+ftcond actually performs best. We do not see a substantial improvement when using RoPE instead of sincos positional embedding. As discussed when introducing AdaLN, its per-block conditioning may help avoid vanishing of the action information through the predictor's layers, while being more compute-efficient than other conditioning schemes (Peebles & Xie, 2023). We also study AdaLN-zero, following Peebles & Xie (2023)'s naming. Although Peebles & Xie (2023) find AdaLN-zero to outperform AdaLN in their setup, we

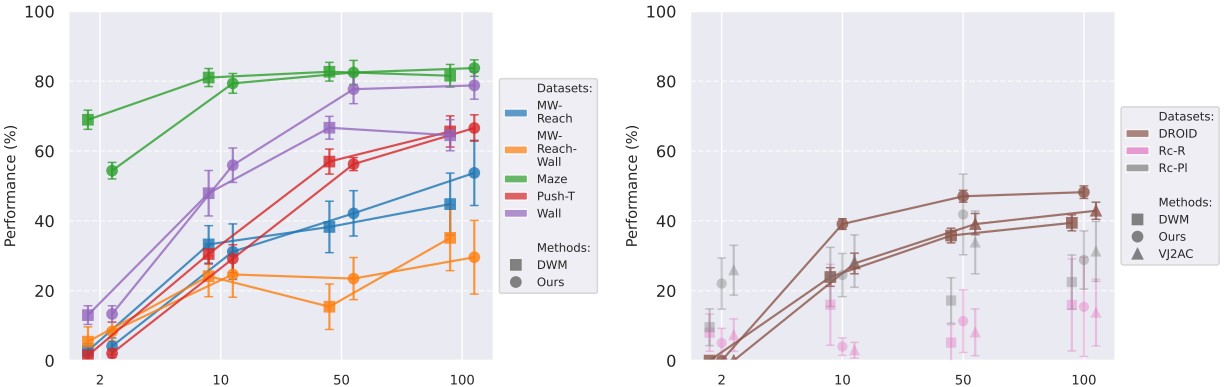

Figure 7: Data scaling ablation: we display planning performance for models trained on 2%, 10%, 50% or 100% of the entire dataset, fixing the number of iterations across runs, independently of dataset size, with dataset metrics in Table 8. (a) Models trained on simulated environments, namely Metaworld, Maze, Push-T and Wall. (b) Models trained on the real-world robotic DROID dataset, evaluated on real-world videos (DROID) and on the Robocasa simulator. Rc-Pl and Rc-R denote the Place and Reach tasks of Robocasa.

observe that, despite a higher average performance, AdaLN-zero underperforms AdaLN on the environments that provide the most reliable signal (DROID, PushT, Maze), which are less prone to noise in success rate and yield more consistent results across our other design choice experiments. Hence, we will consider, for our final JEPA-WMs optimum, the AdaLN variant. One important consideration when scaling predictor embedding dimension is maintaining the ratio of action to visual dimensions, which requires increasing the action embedding dimension in the feature conditioning case. To isolate the effect of the conditioning scheme from capacity differences due to different action ratios, we conduct additional experiments with equalized action ratios (see Section G.1), which reveal task-dependent preferences between conditioning schemes that cannot be attributed to action ratio alone.

**Model scaling.** We show in Figures 6a and 6b that increasing encoder size (with predictor width) or predictor depth does not improve performance on simulated environments. We identify three complementary hypotheses for this behavior: **(a)** simulated tasks are simple enough to saturate at small model sizes, so additional capacity brings no benefit; **(b)** larger embedding spaces make the planning optimization landscape harder to navigate, as the planner must distinguish nearby states in a higher-dimensional space (see Figure 18); **(c)** with fixed training compute, larger models see fewer gradient updates per parameter, potentially leading to underfitting. Notably, the optimal predictor depth appears to be 6 for most simulated environments, and possibly as low as 3 for the simplest 2D navigation tasks (Wall, Maze). However, on DROID, we observe a clear and consistent positive correlation between both encoder size and predictor depth with planning performance. This indicates that real-world data with complex visual dynamics benefits from higher-capacity models. This contrast provides a practical guideline for practitioners: scaling model capacity is most beneficial when the environment exhibits complex, high-dimensional dynamics (as in real-world robotics), while simulated environments with simple dynamics saturate at small model sizes.

**Data scaling.** The results in Figure 7 provide three insights. **(i)** For all datasets and methods considered, performance clearly increases when scaling data, as the world model captures more diverse dynamics and nuances of the environment, allowing for more accurate predictions and better-informed planner optimization. This is expected, as the three methods rely on the same main learning signal, which is the one-step teacher-forcing loss in state embedding space. **(ii)** Our method outperforms baselines especially on DROID and Wall, where the data scaling seems less saturated. On the Push-T and Metaworld tasks, it also seems like increasing data diversity would increase performance by a large margin, yet our method's advantage is less clear. **(iii)** The results for models trained on DROID are in Figure 7b. When evaluating these on offline planning on Franka videos (denoted DROID), we clearly see performance scaling with data quantity. Yet, for Robocasa Place and Reach, scaling is less clear, due to the domain gap between DROID and Robocasa.

Table 2: Comparison of our best model to DINO-WM and V-JEPA-2-AC. MW-R and MW-RW denote the Reach and Reach-Wall tasks of Metaworld. Rc-Pl and Rc-R denote the Place and Reach tasks of Robocasa. V-JEPA-2-AC numbers are obtained from our retrained model with the rollout-loss bug fix described in Section C, not from the public checkpoint. Best model is in **bold**.

| Model | Maze | Wall | Push-T | MW-R | MW-RW | Rc-R | Rc-Pl | DROID |
|---|---|---|---|---|---|---|---|---|
| DWM | 81.6 (3.4) | 64.1 (4.6) | 66.0 (4.7) | 44.8 (8.9) | 35.1 (9.4) | 19.1 (13.4) | 21.7 (7.2) | 39.4 (2.1) |
| VJ2AC | — | — | — | — | — | 16.2 (8.3) | **33.1 (7.2)** | 42.9 (2.5) |
| Ours | **83.9 (2.3)** | **78.8 (3.9)** | **70.2 (2.8)** | **58.2 (9.3)** | **41.6 (10.0)** | **25.4 (16.6)** | 30.7 (8.0) | **48.2 (1.8)** |

### 5.3 Our proposed optimum in the class of JEPA-WMs

We combine the findings of our study and propose optimal models for each of our robotic environments, that we compare to concurrent JEPA-WM approaches: DINO-WM (Zhou et al., 2024a) and V-JEPA-2-AC (Assran et al., 2025). For simulated environments, we use a ViT-S encoder and a ViT-S predictor with depth 6, AdaLN conditioning, and RoPE positional embeddings. We train our models with proprioception and a 2-steps rollout loss, with a maximum context of $W = 3$. For DROID and Robocasa, following our model size findings, we use a ViT-L encoder with a ViT-L predictor of depth 12, without proprioception. We plan with CEM $L_2$ for all environments. We use DINOv2 on all environments, except on the photorealistic DROID and Robocasa, where we use DINOv3. We summarize the recommended recipe per task type in Table 1. As presented in Table 2, we outperform DINO-WM and V-JEPA-2-AC in most environments. We provide a full comparison across all planner configurations in Table 11. We propose in Figure 2 a qualitative comparison of the object interaction abilities of our model against DINO-WM and V-JEPA-2-AC, in a simple counterfactual experiment, where we unroll two different action sequences from the same initial state, one where the robot lifts a cup, and one where it does not. Our model demonstrates a better prediction of the effect of its actions on the environment.

## 6 Conclusion

In this paper, we studied the effect of several training and planning design choices of JEPA-WMs on planning in robotic environments. We found that several components play an important role, such as the use of proprioceptive input, the multistep rollout loss, or the choice of visual encoder. We found that image encoders with fine object segmentation capabilities are better suited for the manipulation and navigation tasks that we considered compared to video encoders. We found that having enough context to infer velocity is important, but that too long context harms performance, obviously due to seeing less unique trajectories during training and likely also having less useful gradient from predicting from long context. On the architecture side, we found that the action conditioning technique matters, with AdaLN being a strong choice on average, compared to sequence and feature conditioning, though results are task-dependent. We found that scaling model size (encoder size with predictor width, and predictor depth) does not improve performance on simulated environments. However, on real-world data (DROID and Robocasa), both larger encoders and deeper predictors yield consistent improvements, suggesting that scaling benefits depend on task complexity. We introduced an interface for planning with Nevergrad optimizers, leaving room for exploration of optimizers and hyperparameters. On the planning side, we found that CEM $L_2$ performs best overall. The NG planner performs similarly to CEM on real-world manipulation data (DROID and Robocasa) while requiring less hyperparameter tuning, making it a practical alternative when transitioning to new tasks or datasets. Gradient-based planners (GD and Adam) excel on tasks with smooth cost landscapes like Metaworld, but fail on 2D navigation or contact-rich manipulation tasks due to local minima. Finally, we applied our learnings and proposed models outperforming concurrent JEPA-WM approaches, DINO-WM and V-JEPA-2-AC. A limitation of this class of approaches is the deterministic predictor: the MSE loss learns the conditional mean of potentially genuinely multi-modal futures. This is mitigated in our benchmarks by their deterministic dynamics, by the latent abstraction of task-irrelevant variability and closed-loop MPC providing robustness to prediction errors. For environments with aleatoric uncertainty, JEPA-WMs would require stochastic extensions such as latent variable injection (LeCun, 2022) or diffusion in latent space.

**Broader Impact Statement**

This work focuses on learning world models for physical agents, with the aim of enabling more autonomous and intelligent robots. We do not anticipate particular risk of this work, but acknowledge that further work building on it could have impact on the field of robotics, which is not exempt of risks of misuse. We also acknowledge the environmental impact of training large models, and we advocate for efficient training procedures and sharing of pretrained models to reduce redundant computation.

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

**Appendix**

**Contents**

# A Notation

Table 3: Summary of notation used throughout this paper.

| Symbol | Description |
| --- | --- |
| *Observations, actions and embeddings* | |
| $o_t$ | Observation (image frame) at time $t$ |
| $a_t$ | Action at time $t$ |
| $z_t$ | State embedding at time $t$, i.e. $z_t = E_{\phi,\theta}(o_t)$ |
| $\hat{z}_t$ | Predicted state embedding at time $t$ |
| *Model components* | |
| $E_{\phi,\theta}$ | Encoder (visual, and optionally proprioceptive) |
| $E_\theta^{prop}$ | Proprioceptive encoder |
| $P_\theta$ | Predictor |
| $A_\theta$ | Action encoder |
| $F_{\phi,\theta}$ | Predictor unrolling function (Equations (3) and (4)) |
| $G_{\phi,\theta}$ | Planning evaluation function |
| *Parameters* | |
| $\phi$ | Encoder parameters (frozen) |
| $\theta$ | Learnable parameters (predictor, action encoder, proprioceptive encoder) |
| *Losses and objectives* | |
| $\mathcal{L}$ | Teacher-forcing training loss (Equation (1)) |
| $\mathcal{L}_k$ | $k$-step rollout loss (Equation (5)) |
| $L_\alpha^p$ | Planning objective (Equation (2)) |
| $L_{vis}, L_{prop}$ | Visual and proprioceptive dissimilarity metrics |
| $\alpha$ | Proprioceptive loss weight in planning |
| *Dimensional and architectural parameters* | |
| $D$ | Embedding dimension of encoder / predictor |
| $A$ | Action dimension |
| $f$ | Frameskip factor |
| $f_a$ | Action embedding dimension (feature conditioning) |
| $h \times w$ | Number of spatial patches of the visual encoder |
| $B$ | Training batch size |
| *Training parameters* | |
| $W$ | Maximum teacher-forcing context length at training time |
| $W^t$ | Maximum rollout context length at training time |
| *Planning parameters (see also Table 10)* | |
| $W^p$ | Maximum rollout context length at planning time |
| $H$ | Planning horizon (number of predicted steps) |
| $N$ | Number of candidate action trajectories |
| $J$ | Number of optimizer iterations |
| $K_e$ | Number of top trajectories retained (CEM) |
| $m$ | Number of planned actions stepped in environment |
| $M$ | Maximum number of steps per planning episode |

# B Extended Related Work

**Alternative latent-space planning paradigms.** Several approaches have been proposed for planning in learned latent spaces, differing from JEPA-WMs in their dynamics model class, optimization strategy, or training assumptions. *Locally-linear latent dynamics models*, such as Embed to Control (E2C) (Watter et al.,

2015), learn a latent space where dynamics are locally linear, enabling the use of iterative Linear Quadratic Regulator (iLQR) for trajectory optimization. E2C is derived directly from an optimal control formulation in latent space and can operate on raw pixel observations without requiring explicit reward signals during training, using instead a reconstruction-based objective combined with dynamics constraints. *Gradient-based trajectory optimization* through learned dynamics, as in Universal Planning Networks (UPN) (Srinivas et al., 2018), uses differentiable forward models to directly backpropagate planning gradients through predicted trajectories. We compare to this paradigm in our experiments (gradient descent planner in Section 5.2), finding it effective for smooth cost landscapes but prone to local minima in navigation tasks. *Diffusion-based planners* (Janner et al., 2022; Zhou et al., 2024b) generate trajectory distributions via iterative denoising, offering multi-modal planning and implicit constraint satisfaction. While Diffuser typically requires offline RL datasets with reward annotations (Janner et al., 2022), recent work like DMPC (Zhou et al., 2024b) demonstrates diffusion-based MPC on continuous control tasks, though direct comparison with visual goal-conditioned JEPA-WMs remains challenging due to different experimental settings and assumptions. Our work focuses on systematically studying design choices within the JEPA-WM framework, which offers reward-free training from visual observations and flexible test-time goal specification—a complementary setting to these alternative paradigms. Regarding robustness to out-of-distribution trajectories, Parthasarathy et al. (2025) show that adversarial perturbations of latent states and actions during training smooth the world model's loss landscape and substantially improve gradient-based planning on tasks that we also study (Push-T, PointMaze, Wall); their approach is complementary to ours, which relies on the inherently more robust CEM planner.

**Explicit and implicit world models.** We distinguish two paradigms for learning world models in latent space. *Explicit world models* learn an autoregressive, action-conditioned predictor $\hat{z}_{t+1} = P_\theta(z_t, A_\theta(a_t))$ that generates task-agnostic state embeddings one step at a time. Because training is decoupled from any reward or task specification, test-time planning algorithms (CEM, MPPI, gradient-based optimization) can optimize arbitrary cost functions over rolled-out trajectories in embedding space. This makes explicit models highly flexible: the cost function is fully decoupled from the learned dynamics. *Implicit world models* fold multi-step temporal abstraction into the representation itself, bypassing the need for autoregressive rollouts. TD-JEPA (Bagatella et al., 2026) is a canonical example: it trains a *policy-conditioned* multi-step predictor that approximates successor features, enabling zero-shot reward optimization for any reward function lying in the span of the learned features. Note that setting the discount factor $\gamma = 0$ in TD-JEPA recovers one-step prediction, collapsing the model to the explicit, action-conditioned paradigm.

*The two paradigms trade off compute allocation.* Explicit models defer computation to test-time planning, but their training can be lightweight: JEPA-WMs that jointly learn a shallow encoder and predictor, such as PLDM (Sobal et al., 2025) or the action-conditioned video JEPA example of EB-JEPA (Terver et al., 2026), converge in as few as thousands of gradient steps on simple simulated tasks. Implicit models like TD-JEPA front-load computation into training (authors report 1-2 million gradient steps to capture long-horizon occupancies), yet at test time they can evaluate new reward functions in a single forward pass (zero-shot RL) without any search loop. *Explicit world models allow for a broader range of use cases at test time.* With an explicit world model, one can generate counterfactual trajectories in state embedding space, by hardcoding action trajectories corresponding to different future scenarios, as we showcase in Figure 2. Implicit world models like TD-JEPA are non-generative (in embedding space), hence cannot perform such counterfactual generation. Additionally, TD-JEPA relies on zero-shot RL, which allows the deployment of policies that are optimal for rewards that belong to the span of the learned feature space. With our explicit world models, since the cost functions are independent of the learned world model, we have more freedom on defining cost (reward) functions at test time, which can be any function from the feature space to $\mathbb{R}_+$. *A middle ground is occupied by hybrid methods.* TD-MPC2 (Hansen et al., 2024) learns an explicit action-conditioned latent dynamics model alongside a reward predictor and a learned policy, then uses the policy to warm-start MPPI planning at test time. This combination retains the flexibility of explicit trajectory optimization while amortizing part of the search cost through the learned policy. Our study focuses on the explicit paradigm. A direct empirical comparison between explicit and implicit world models, including training-cost, inference-cost, and generalization trade-offs, is an important direction that we leave for future work.

**Reward- and reconstruction-based latent world models vs. JEPA-WMs.** We provide here a per-method discussion of how established MBRL methods differ from the JEPA-WM recipe defined in the main text (training via embedding prediction only; planning via goal-distance minimization). *PlaNet* (Hafner et al., 2019) learns a Recurrent State-Space Model (RSSM) with a deterministic recurrent component and a stochastic latent variable, trained via a variational objective combining pixel reconstruction, reward prediction, and a KL regularizer; at test time it runs CEM to maximize predicted cumulative rewards. *The Dreamer series* (Hafner et al., 2020; 2021; 2024) inherits the RSSM but replaces online planning with an actor-critic trained entirely on imagined rollouts, still relying on reconstruction and reward prediction for the world model. *MuZero* (Schrittwieser et al., 2020) learns a latent dynamics model, a reward predictor, and value/policy heads trained end-to-end via MCTS; its latent space is optimized solely to predict policies, values, and rewards, making it fundamentally reward-driven. *TD-MPC2* (Hansen et al., 2024) pairs an action-conditioned latent dynamics model with a reward predictor and a learned policy used to warm-start MPPI planning, again maximizing predicted rewards. All four methods therefore rely on reward prediction, reconstruction, or both for training, and on reward maximization for planning. The reward-free, goal-conditioned nature of JEPA-WMs is both a strength (no reward engineering, flexible goal specification at test time) and a limitation (no straightforward way to optimize cumulative reward without a separate reward model).

## C   Training details

**Predictor.**   We train using the AdamW optimizer, with a constant learning rate on the predictor, action encoder and optional proprioceptive encoder. We use a cosine scheduler on the weight decay coefficient. For the learning rate, we use a constant learning rate without any warmup iterations. We summarize training hyperparameters common to environments in Table 4. We display the environment-specific ones in Table 5. Both the action and proprioception are first embedded with a linear kernel applied to each timestep, of input dimension action_dim or proprio_dim (equal to the unit action or proprioceptive dimension times the frameskip) and output dimension action_embed_dim or proprio_embed_dim. We stress that, for memory requirements, for our models with 6-step and $W = 7$, the batch size is half the default batch size displayed in Table 5, which leads to longer epochs, as in Table 6. For our models trained on DROID, to compare to V-JEPA-2-AC and because of the dataset complexity compared to simulated ones, we increase the number of epochs to 315, and limit the iterations per epoch to 300, as displayed in Table 5.

**Action conditioning of the predictor.**   We study four predictor conditioning variants to inject action information in Figure 5a. The conditioning method determines where and how action embeddings are incorporated into the predictor architecture:

- **Feature conditioning with sincos positional embeddings**: Action embeddings $A_\theta(a)$ are concatenated with visual token features $E_\theta(o)$ along the embedding dimension. Each timestep's concatenated features are then processed with 3D sinusoidal positional embeddings. This increases the feature dimension and the hidden dimension of the predictor from $D$ to $D + f_a$, where $f_a$ is the action embedding dimension.

- **Sequence conditioning with RoPE**: Actions are encoded as separate tokens and concatenated with visual tokens along the sequence dimension, keeping the predictor's hidden dimension to $D$ (as in the encoder). Rotary Position Embeddings (RoPE) is used at each block of the predictor.

- **Feature conditioning with RoPE**: This conditioning scheme combines feature concatenation (as in the first variant) with RoPE positional embeddings instead of sincos.

- **AdaLN conditioning with RoPE**: Action embeddings modulate the predictor through Adaptive Layer Normalization at each transformer block. Specifically, action embeddings are projected to produce scale and shift parameters that modulate the layer normalization statistics. This approach allows action information to influence all layers of the predictor rather than only at input, potentially preventing vanishing of action information through the network. Combined with RoPE for positional encoding, this design is also more compute-efficient as it avoids increasing feature or sequence dimensions.

One can estimate the strength of the action conditioning of the predictor by looking at the *action ratio*, i.e., the ratio of dimensions (processed by the predictor) corresponding to action, on the total number of dimensions. With feature conditioning, this ratio is $\frac{f_a}{D+f_a}$, where $f_a$ is the action embedding dimension. When performing sequence conditioning, this ratio is $\frac{1}{hw+1} = \frac{1}{257}$ for standard patch sizes, with $h$ and $w$ being the height and width of the token grid, namely 16 (as explained in Table 7). Thus, feature conditioning typically yields a higher action ratio than sequence conditioning.

The inductive bias we expect from these designs relates to how strongly actions can influence predictions. AdaLN's per-layer modulation should provide the most consistent action conditioning throughout the predictor depth, which may explain its superior empirical performance, see Figure 5a.

**Train time.**   We compute the average train time per epoch for each combination of world model and dataset in Table 6.

**Visual decoder.**   We train one decoder per encoder on VideoxMix2M (Bardes et al., 2024) with a sum of L2 pixel space and perceptual loss (Zhang et al., 2018). With a ViT-S encoder, we choose a ViT-S decoder

Table 4: Training hyperparameters of some of the studied models common to all environments. If left empty, the hyperparameter value is the same as the leftmost column. WM-V refers to models trained with V-JEPA and V-JEPA2 encoders.

| Hyperparameter | WM | WM-L | WM-V |
|---|---|---|---|
| *data* | | | |
| $W$ | 3 | 3 | 3 |
| $f$ | 5 | - | - |
| resolution | 224 | 224 | 256 |
| *optimization* | | | |
| lr | 5e-4 | - | - |
| start_weight_decay | 1e-7 | - | - |
| final_weight_decay | 1e-6 | - | - |
| AdamW $\beta_1$ | 0.9 | - | - |
| AdamW $\beta_2$ | 0.995 | - | - |
| clip_grad | 1 | - | - |
| *architecture* | | | |
| patch_size | 14 | - | 16 |
| pred_depth | 6 | - | - |
| pred_embed_dim | 384 | 1024 | 1024 |
| enc_embed_dim | 384 | 1024 | 1024 |
| *hardware* | | | |
| dtype | bfloat16 | - | - |
| accelerator | H100 80G | - | - |

Table 5: Environment-specific training hyperparameters. proprio_embed_dim is used only for models using proprioception. For $\text{WM}_W$-6-step, the batch size is half the default batch size displayed here. We do not train but only evaluate DROID models on Robocasa.

| Hyperparameter | Metaworld | Push-T | Maze | Wall | DROID |
|---|---|---|---|---|---|
| *optimization* | | | | | |
| batch_size | 256 | 256 | 128 | 128 | 128 |
| epochs | 50 | 50 | 50 | 50 | 315 |
| *architecture* | | | | | |
| action_dim | 20 | 10 | 10 | 10 | 7 |
| action_embed_dim | 20 | 10 | 10 | 10 | 10 |
| proprio_dim | 4 | 4 | 4 | 4 | 7 |
| proprio_embed_dim | 20 | 20 | 20 | 10 | 10 |

with depth 12. When the encoder is a ViT-L we choose a ViT-L decoder with depth 12. We train this decoder for 50 epochs with batch size 128 on trajectory slices of 8 frames.

**State decoder.** We train a depth 6 ViT-S decoder to regress the state from one `CLS` token (Darcet et al., 2024). A linear projection at the entry projects each patch token from the frozen encoder to the right embedding dimension, 384. At the exit, a linear layer projects the `CLS` token to a vector with the same number of dimensions as the state to decode.

**V-JEPA-2-AC reproduction.** To reproduce the V-JEPA-2-AC results, we find a bug in the code that yields the official results of the paper. The 2-step rollout loss is miscomputed, what is actually computed for this loss term is $\|P_\phi(a_{1:T}, s_1, z_1) - z_T\|_1$ in the paper's notations. This means that the model, when

Table 6: Model-specific training times in minutes per epoch on 16 H100 80 GB GPUs for Maze and Wall, on 32 H100 GPUs for Push-T and Metaworld. We denote WM-B, WM-L the variants of the base model with size ViT-B and ViT-L, WM-prop the variant with proprioception and WM-V the variant with V-JEPA encoders. For DROID, we display the train time for 10 epochs since we train for 315 epochs.

| Model | Metaworld | Push-T | Maze | Wall | DROID |
|---|---|---|---|---|---|
| 1-step | 23 | 48 | 5 | 1 | 7 |
| 2-step | 23 | 49 | 5 | 1 | 8 |
| 3-step | 23 | 50 | 5 | 1 | 9 |
| 6-step | 30 | 64 | 16 | 2 | 17 |
| $W = 7$ | 20 | 42 | 5 | 1 | 13 |
| WM-B | 23 | 50 | 5 | 1 | 8 |
| WM-L | 25 | 50 | 5 | 1 | 8 |
| WM-prop | 24 | 50 | 5 | 1 | 7 |
| WM-V | 25 | 60 | 7 | 2 | 9 |

receiving as input a groundtruth embedding $z_1$, concatenated with a prediction $\hat{z}_2$, is trained to output $\hat{z}_2$. We fix this bug and retrain the models. When evaluating the public checkpoint of the V-JEPA-2-AC on our DROID evaluation protocol, the action score is much lower than our retrained V-JEPA-2-AC models after bug fixing. Interestingly, the public checkpoint of the V-JEPA-2-AC predictor, although having much worse performance at planning, yields image decodings after unrolling very comparable to the fixed models, and seems to pass the simple counterfactual test, as shown in Figure 2.

Regarding planning, VJEPA2-AC does not normalize the action space to mean 0 and variance 1, contrary to DINO-WM, so we also do not normalize with our models, for comparability to V-JEPA-2-AC. The VJEPA2-AC CEM planner does clip the norm of the sampled actions to 0.1, which is below the typical std of the DROID actions. We find this clipping useful to increase planning performance and adopt it. Moreover, the authors use momentum in the update of the mean and std, which should be useful when the number of CEM iterations is high, but we do not find it to make a difference although we use 15 CEM iterations, hence do not adopt it in the planning setup on DROID. The planning procedure in V-JEPA-2-AC optimizes over four dimensions, the first three ones corresponding to the delta of the end-effector position in cartesian space, and the last one to the gripper closure. The 3 orientation dimensions of the proprioceptive state are $2\pi$-periodic, so they often rapidly vary from a negative value above $\pi$ to one positive below $\pi$. The actions do not have this issue and have values continuous in time.

**Data augmentation ablations.** In V-JEPA-2-AC, the adopted random-resize-crop effectively takes a central crop with aspect ratio 1.35, instead of the original DROID (et al., 2024) aspect ratio of $1280/720 \simeq 1.78$, and resizes it to 256x256. On simulated datasets where videos are natively of aspect ratio 1, this augmentation does not have effect. DINO-WM does not use any data augmentation. We try applying the pretraining augmentation of V-JEPA2, namely a random-resize-crop with aspect ratio in $[0.75, 1.33]$ and scale in $[0.3, 1.0]$, but without its random horizontal flip with probability 0.5 (which would change the action-state correspondence), and resizing to 256x256. We find this detrimental to performance, as the agent sometimes is not fully visible in the crop.

**Ablations on models trained with video encoders.** When using V-JEPA and V-JEPA-2 encoders, before settling on training loss and encoding procedure, we perform some ablations. First, we find that the best performing loss across MSE, $L_1$ and smooth $L_1$ is the MSE prediction error, even though V-JEPA and V-JEPA-2 were trained with an $L_1$ prediction error. Then, to encode the frame sequence, one could also leverage the ability of video encoders to model dependency between frames. To avoid leakage from information of future frames to past frames, we must in this case use a frame-causal attention mask in the encoder, just as in the predictor. We have a frameskip $f$ between the consecutive frames sampled from the trajectory dataset, considering them consecutive without duplicating them will result in $(W + 1)/2$ visual embedding timesteps. In practice, we find that duplicating each frame before encoding them as a video gives

Table 7: Detailed comparison of encoder configurations used in our experiments. All encoders use frozen weights during predictor training.

| Configuration | DINOv2 ViT-L | DINOv3 ViT-L | V-JEPA ViT-L | V-JEPA2 ViT-L |
|---|---|---|---|---|
| *Encoder Architecture* | | | | |
| Encoder type | Image | Image | Video | Video |
| Model size | ViT-L/14 | ViT-L/16 | ViT-L/16 | ViT-L/16 |
| Patch embedding | Conv2d(14, 14) | Conv2d(16, 16) | Conv3d(2, 16, 16) | Conv3d(2, 16, 16) |
| Embedding dimension | 1024 | 1024 | 1024 | 1024 |
| Patches per timestep | $16 \times 16 = 256$ | $16 \times 16 = 256$ | $16 \times 16 = 256$ | $16 \times 16 = 256$ |
| Input normalization | ImageNet stats | ImageNet stats | ImageNet stats | ImageNet stats |
| Positional encoding | Sincos | RoPE | Sincos | RoPE |
| Attention mask | Full | Full | Full | Full |
| *Input Preprocessing* | | | | |
| Input resolution | $224 \times 224$ | $256 \times 256$ | $256 \times 256$ | $256 \times 256$ |
| Input frame count | 1 per timestep | 1 per timestep | 2 per timestep | 2 per timestep |
| Frame duplication | No | No | Yes (duplicate each) | Yes (duplicate each) |

better performance than without duplication. Still, these two alternative encoding techniques yield much lower performance than using video encoders as frame encoders by duplicating each frame and encoding each pair independently. V-JEPA-2-AC (Assran et al., 2025) does use the latter encoding technique. They encode the context video by batchifying the video and duplicating each frame, accordingly to the method which we find to work best by far on all environments. In this case, for each video of $T$ frames, the encoder processes a batch of $T$ frames, so having a full or causal attention mask is equivalent.

**Encoder comparison details.** Given the above chosen encoding method for video encoders, we summarize the encoder configurations in Table 7. The key differences are: (1) encoder weights themselves—DINOv2/v3 trained with their several loss terms on images vs V-JEPA/2 trained with masked prediction on videos; (2) frame preprocessing—video encoders require frame duplication (each frame duplicated to form a 2-frame input); (3) patch sizes—14 for DINOv2 (256 tokens/frame, 224 resolution) vs 16 for others (256 tokens/frame, 256 resolution for V-JEPA/2, DINOv3). We use raw patch tokens without aggregation or entry/exit projections and use all encoders frozen, without any finetuning. DINOv2/v3's superior performance on our tasks likely stems from better fine-grained object segmentation capabilities crucial for manipulation and navigation, as discussed in the main text.

**Multistep rollout variants ablations.** We ablate several rollout strategies as illustrated in Figure 8, following the scheduled sampling (Bengio et al., 2015) and TBPTT (Jaeger, 2002) literature for sequence prediction in embedding space. When using transformers, one advantage we have compared to the classical RNN architecture, is the possibility to perform next-timestep prediction **in parallel for all timesteps** in a more computationally efficient way, thanks to a carefully designed attention mask. In our case, each timestep is a frame, made of $H \times W$ patch tokens. We seek to train a predictor to minimize rollout error, similarly to training RNNs to generate text (Bengio et al., 2015). One important point is that, in our planning task, we feed a context of one state (frame and optionally proprioception) $o_t$, then recursively call the predictor as described in equation 3, equation 4 to produce a sequence of predictions $\hat{z}_{t+1}, \ldots, \hat{z}_{t+k}$. Since our predictor is a ViT, the input and output sequence of embeddings have same length. At each unrolling step, we only take the last timestep of the output sequence and concatenate it to the context for the next call to the predictor. We use a maximum sliding window $W^p$ of two timesteps in the context at test time, see Section 4 and Table 10. At training time, we add multistep rollout loss terms, defined in equation 5 to better align training task and unrolling task at planning time. Let us define the *order* of a prediction as the number of calls to the predictor function required to obtain it from a groundtruth embedding. For a predicted embedding $z_t^{(k)}$, we denote the timestep it corresponds to as $t$ and its prediction order as $k$. There are various ways to implement such losses with a ViT predictor.

1. Increasing order rollout illustrated in Figure 8. In this setup, the prediction order is increasing with the timestep. This strategy has two variants.

    (a) The "Last-gradient only" variant is the most similar to the unrolling at planning time. We concatenate the latest timestep outputted by the predictor to the context for the next unrolling step.

    (b) The "All-gradients" variant generalizes the previous variant, by computing strictly more (non-redundant) additional loss terms although using the same number of predictor unrolling steps. These additional loss terms correspond to other combinations of context embeddings.

2. "Equal-order": In this variant, at each unrolling step $k$, the predictor input is the full output of the previous unrolling step, denoted $z_t^{(k-1)}, \ldots, z_{t+\tau}^{(k-1)}$, deprived of the rightmost timestep $z_{t+\tau}^{(k-1)}$ since it has no matching target groundtruth embedding $z_{t+\tau}$.

In all the above methods, we can add sampling schedule (Bengio et al., 2015), i.e. have a probability $p$ to flip one of the context embeddings $z_t^{(k)}$ to the corresponding groundtruth embedding $z_t$.

The takeaways from our ablations are the following:

- The "Equal-order" strategy gives worse results. This is due to the fact that, with this implementation, the predictor does not take as input a concatenation (over time dimension) of ground truth embeddings and its predictions. Yet, at planning time, the unrolling function keeps a sliding context of ground truth embeddings as well as predictions. Hence, although this strategy uses more gradient (more timesteps have their loss computed in parallel) than the "Last-gradient only" variant, it is less aligned with the task expected from the predictor at planning time.

- The strategy that yields best success rate is the 2-step "Last-gradient only" variant with random initial context.

- Even though the "All-gradients" variant has an ensemble of loss terms that strictly includes the ones of the "Last-gradient only" strategy, it does not outperform it.

- Across all strategies, we find simultaneously beneficial in terms of success rate and training time to perform TBPTT (Jaeger, 2002), detaching the gradient on all inputs before each pass in the predictor.

In a nutshell, what matters is to train the predictor to receive as input a mix of encoder outputs and predictor outputs. This makes the predictor more aligned with the planning task, where it unrolls from some encoder outputs, then concatenates to it its own predictions.

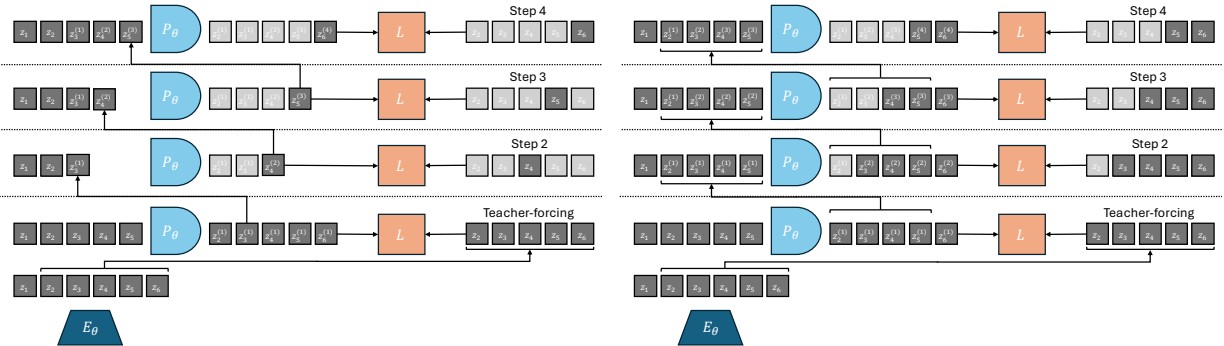

Figure 8: Two rollout strategies with any predictor network predicting the next timestep (frame) simultaneously for all timesteps. Predictor is used as an RNN by recursively feeding it a mix of its previous predictions and groundtruth embeddings. We vertically represent the "predictor unrolling step" dimension. For a predicted embedding $z_t^{(k)}$, we denote the timestep it corresponds to as $t$ and its prediction order as $k$. The embeddings that enter in the loss computation are in grey whereas those which do not are in light grey. Left: "Last-gradient-only" strategy. We sample a random groundtruth embedding prefix, $(z_1, \ldots, z_t)$ (in this figure $t = 2$), and concatenate only the latest prediction to the predictor context at the next unrolling step. Strategy used in V-JEPA-2-AC with a groundtruth embedding prefix always equal to $z_1$. Right: "All-gradients" strategy, we compute all available prediction tasks without redundancies, e.g. we exclude from loss computation prediction tasks that have already been included in loss computation at previous timesteps, i.e. $L(z_t^{(t-1)}, z_t)$.

## D    Error propagation in autoregressive latent prediction

We formalize the exponential error growth claim made in the main text. Consider the autoregressive predictor recurrence from equation 4:

$$\hat{z}_{i+1} = P_\theta(\hat{z}_{i-w:i}, A_\theta(a_{i-w:i})), \quad i = t, \ldots, t + H - 1, \quad \hat{z}_t = E_{\phi,\theta}(o_t). \tag{6}$$

Here $w$ denotes the maximum context length during rollout (set to $W^t$ at training time and to $W^p$ at planning time, see Table 10), while $W+1$ denotes the number of frames per training slice. Let $z_i = E_{\phi,\theta}(o_i)$ denote the groundtruth embedding at step $i$, and define the prediction error $\epsilon^{(k)} = \|\hat{z}_{t+k} - z_{t+k}\|$ at horizon $k$. We denote the intrinsic one-step prediction error (the residual on groundtruth inputs) by $\delta = \sup_i \|P_\theta(z_{i-w:i}, A_\theta(a_{i-w:i})) - z_{i+1}\|$.

**Proposition 1** (Error bound, simplified case $w = 0$). *Assume $P_\theta(\cdot, A_\theta(a_i))$ is $\Lambda$-Lipschitz with respect to its state input for every action $a_i$, with $\epsilon^{(0)} = 0$ (the initial embedding is groundtruth). Then:*

$$\epsilon^{(H)} \leq \begin{cases} \delta \dfrac{\Lambda^H - 1}{\Lambda - 1} & \text{if } \Lambda \neq 1, \\ H\,\delta & \text{if } \Lambda = 1. \end{cases} \tag{7}$$

*For $\Lambda > 1$, the error grows as $\mathcal{O}(\Lambda^H)$, i.e., exponentially in the horizon $H$. For $\Lambda = 1$ (isometric predictor), the error grows linearly: $\epsilon^{(H)} \leq H\,\delta$. For $\Lambda < 1$ (contractive predictor), the error saturates at $\delta/(1 - \Lambda)$, independently of $H$.*

*Proof.* From the Lipschitz property and the triangle inequality:

$$
\begin{aligned}
\epsilon^{(k+1)} &= \|\hat{z}_{t+k+1} - z_{t+k+1}\| \\
&= \|P_\theta(\hat{z}_{t+k}, A_\theta(a_{t+k})) - z_{t+k+1}\| \\
&\leq \|P_\theta(\hat{z}_{t+k}, A_\theta(a_{t+k})) - P_\theta(z_{t+k}, A_\theta(a_{t+k}))\| + \|P_\theta(z_{t+k}, A_\theta(a_{t+k})) - z_{t+k+1}\| \\
&\leq \Lambda\,\epsilon^{(k)} + \delta.
\end{aligned}
\tag{8}
$$

Solving the linear recurrence $\epsilon^{(k+1)} \leq \Lambda\,\epsilon^{(k)} + \delta$ with $\epsilon^{(0)} = 0$ yields equation 7. □

**Proposition 2** (Tightness of the bound). *The upper bound in Proposition 1 is tight in the following senses.*

*(a) **Achievability by affine predictors.** For any $\Lambda > 0$ and $\delta > 0$, there exist a $\Lambda$-Lipschitz affine predictor $P_\theta$ on $\mathbb{R}$ with one-step error $\delta$ and a groundtruth trajectory for which the bound in equation 7 is achieved with equality: $\epsilon^{(k+1)} = \Lambda\,\epsilon^{(k)} + \delta$ for all $k \geq 0$, giving $\epsilon^{(H)} = \delta\,(\Lambda^H - 1)\,/\,(\Lambda - 1)$ when $\Lambda \neq 1$ and $\epsilon^{(H)} = H\,\delta$ when $\Lambda = 1$.*

*(b) **Nonlinear predictors, lower bound.** Suppose the Jacobian $J_{P_\theta}$ along the error trajectory has a minimal singular value $\sigma_{\min}(J_{P_\theta}) \geq \ell > 1$, and the one-step error has a component of magnitude at least $\delta' > \delta/(\ell - 1)$ aligned with the right singular vector corresponding to $\sigma_{\min}(J_{P_\theta})$. Then $\epsilon^{(H)} \geq \left(\delta' - \delta/(\ell - 1)\right) \ell^{H-1} = \Omega(\ell^H)$, giving a lower bound that matches the $\mathcal{O}(\Lambda^H)$ upper bound in growth rate.*

*Proof.* **Part (a).** We construct a one-dimensional example that saturates the bound. Let $z \in \mathbb{R}$, define $P_\theta(z) = \Lambda z + \delta$, and let the groundtruth trajectory be constant: $z_{t+k} = 0$ for all $k \geq 0$. The predictor is $\Lambda$-Lipschitz since $|P_\theta(x) - P_\theta(y)| = \Lambda\,|x - y|$, and the one-step error on groundtruth inputs is $|P_\theta(0) - 0| = \delta$. Since $\hat{z}_{t+0} = z_t = 0$, an induction gives $\hat{z}_{t+k} \geq 0$ for all $k$ (because $\hat{z}_{t+k+1} = \Lambda\,\hat{z}_{t+k} + \delta \geq 0$). At each step, the two terms in the triangle inequality equation 8 are $P_\theta(\hat{z}_{t+k}) - P_\theta(0) = \Lambda\,\hat{z}_{t+k} \geq 0$ and $P_\theta(0) - z_{t+k+1} = \delta > 0$. Because both terms are non-negative, the triangle inequality holds with equality: $\epsilon^{(k+1)} = \Lambda\,\epsilon^{(k)} + \delta$. Solving this recurrence with $\epsilon^{(0)} = 0$ yields the claimed formula by direct computation.

**Part (b).** Define the error vector $e_k = \hat{z}_{t+k} - z_{t+k}$ and the one-step residual $r_k = P_\theta(z_{t+k}, A_\theta(a_{t+k})) - z_{t+k+1}$, so that $\|r_k\| \leq \delta$ for all $k$. Decomposing the error at step $k+1$:

$$
\begin{aligned}
e_{k+1} &= P_\theta(\hat{z}_{t+k}) - z_{t+k+1} \\
&= \left[P_\theta(\hat{z}_{t+k}) - P_\theta(z_{t+k})\right] + \underbrace{\left[P_\theta(z_{t+k}) - z_{t+k+1}\right]}_{r_k}.
\end{aligned}
\tag{9}
$$

By the mean value theorem, $P_\theta(\hat{z}_{t+k}) - P_\theta(z_{t+k}) = J_{P_\theta}(\xi_k)\,e_k$ for some $\xi_k$ on the segment between $z_{t+k}$ and $\hat{z}_{t+k}$. Since $\sigma_{\min}(J_{P_\theta}(\xi_k)) \geq \ell$ along the error trajectory by assumption, the first term satisfies $\|J_{P_\theta}(\xi_k)\,e_k\| \geq \ell\,\|e_k\|$. Rearranging equation 9 as $J_{P_\theta}(\xi_k)\,e_k = e_{k+1} - r_k$ and applying the triangle inequality gives $\|J_{P_\theta}(\xi_k)\,e_k\| \leq \|e_{k+1}\| + \|r_k\|$, hence:

$$
\|e_{k+1}\| \geq \|J_{P_\theta}(\xi_k)\,e_k\| - \|r_k\| \geq \ell\,\|e_k\| - \delta.
\tag{10}
$$

By assumption the initial error satisfies $\|e_1\| \geq \delta'$. Unrolling the recurrence equation 10 from $k = 1$ to $k = H-1$:

$$
\begin{aligned}
\|e_H\| &\geq \ell^{H-1}\,\|e_1\| - \delta \sum_{j=0}^{H-2} \ell^j \\
&= \ell^{H-1}\,\|e_1\| - \delta\,\frac{\ell^{H-1} - 1}{\ell - 1} \\
&\geq \ell^{H-1}\,\delta' - \delta\,\frac{\ell^{H-1} - 1}{\ell - 1} \\
&= \left(\delta' - \frac{\delta}{\ell - 1}\right)\ell^{H-1} + \frac{\delta}{\ell - 1}.
\end{aligned}
\tag{11}
$$

Since $\delta' > \delta/(\ell-1)$ by assumption, the leading coefficient is strictly positive, giving $\|e_H\| \geq (\delta' - \delta/(\ell - 1)) \ell^{H-1} = \Omega(\ell^H)$. Combined with the $\mathcal{O}(\Lambda^H)$ upper bound from Proposition 1, this confirms that the error grows exponentially. Note, however, that the bases may differ: $\ell$ lower-bounds $\sigma_{\min}(J_{P_\theta})$ while $\Lambda$ upper-bounds $\sigma_{\max}(J_{P_\theta})$, so in general $\ell \leq \Lambda$ and a gap remains between the lower bound $\Omega(\ell^H)$ and the upper bound $\mathcal{O}(\Lambda^H)$. The two rates coincide when the Jacobian is conformal ($\ell = \Lambda$), e.g. for the scalar predictor constructed in Part (a). $\qquad\square$

**Proposition 3** (Error bound with sliding window $w \geq 1$)**.** *Let $P_\theta$ be $\Lambda_j$-Lipschitz with respect to its $j$-th state input ($j = 0, \ldots, w$) for fixed actions. Then the error dynamics satisfy:*

$$\epsilon^{(k+1)} \leq \sum_{j=0}^{w} \Lambda_j\, \epsilon^{(k-j)} + \delta, \tag{12}$$

*which is a $(w{+}1)$-th order linear recurrence. The asymptotic growth rate is governed by the spectral radius $\rho$ of the companion matrix*

$$\mathbf{C} = \begin{pmatrix} \Lambda_0 & \Lambda_1 & \cdots & \Lambda_w \\ 1 & 0 & \cdots & 0 \\ \vdots & \ddots & & \vdots \\ 0 & \cdots & 1 & 0 \end{pmatrix}. \tag{13}$$

*If $\rho > 1$, errors grow as $\mathcal{O}(\rho^H)$. For typical neural networks without explicit contractivity regularization, $\rho > 1$ (Miyato et al., 2018; Virmaux & Scaman, 2018; Pascanu et al., 2013).*

*Proof.* We adopt the convention $\epsilon^{(k)} = 0$ for $k \leq 0$, since all inputs at or before time $t$ are groundtruth embeddings. For $k \geq 0$, the predicted embedding at step $t+k+1$ is $\hat{z}_{t+k+1} = P_\theta(\hat{z}_{t+k-w:t+k}, A_\theta(a_{t+k-w:t+k}))$, and the corresponding groundtruth embedding satisfies $z_{t+k+1} = P_\theta(z_{t+k-w:t+k}, A_\theta(a_{t+k-w:t+k})) + r_k$ with $\|r_k\| \leq \delta$. By the triangle inequality:

$$\begin{aligned} \epsilon^{(k+1)} &= \|\hat{z}_{t+k+1} - z_{t+k+1}\| \\ &\leq \|P_\theta(\hat{z}_{t+k-w:t+k}, A_\theta(a_{t+k-w:t+k})) - P_\theta(z_{t+k-w:t+k}, A_\theta(a_{t+k-w:t+k}))\| + \delta. \end{aligned} \tag{14}$$

To bound the first term, we change one state argument at a time. Define intermediate sequences $(v^{(m)})_{m=0}^{w+1}$ by setting $v^{(0)} = (\hat{z}_{t+k}, \hat{z}_{t+k-1}, \ldots, \hat{z}_{t+k-w})$ and, for $m = 1, \ldots, w+1$, replacing the $m$-th most recent predicted state with its groundtruth value:

$$v_j^{(m)} = \begin{cases} z_{t+k-j} & \text{if } j < m, \\ \hat{z}_{t+k-j} & \text{if } j \geq m, \end{cases} \qquad j = 0, \ldots, w. \tag{15}$$

Note that $v^{(w+1)} = (z_{t+k}, z_{t+k-1}, \ldots, z_{t+k-w})$. By telescoping and using the $\Lambda_j$-Lipschitz property of $P_\theta$ in its $j$-th state argument (the other arguments being held fixed):

$$\begin{aligned} \|P_\theta(v^{(0)}) - P_\theta(v^{(w+1)})\| &\leq \sum_{m=0}^{w} \|P_\theta(v^{(m)}) - P_\theta(v^{(m+1)})\| \\ &\leq \sum_{j=0}^{w} \Lambda_j\, \|\hat{z}_{t+k-j} - z_{t+k-j}\| = \sum_{j=0}^{w} \Lambda_j\, \epsilon^{(k-j)}. \end{aligned} \tag{16}$$

Substituting equation 16 into equation 14 gives the recurrence $\epsilon^{(k+1)} \leq \sum_{j=0}^{w} \Lambda_j\, \epsilon^{(k-j)} + \delta$.

It remains to relate the growth rate to the spectral radius of $\mathbf{C}$. Consider the auxiliary sequence $(\bar{\epsilon}^{(k)})$ satisfying the equality recurrence $\bar{\epsilon}^{(k+1)} = \sum_{j=0}^{w} \Lambda_j\, \bar{\epsilon}^{(k-j)} + \delta$ with the same initial conditions $\bar{\epsilon}^{(k)} = 0$ for $k \leq 0$. We have $\epsilon^{(k)} \leq \bar{\epsilon}^{(k)}$ for all $k \geq 0$. Setting $\bar{\mathbf{e}}_k = (\bar{\epsilon}^{(k)}, \bar{\epsilon}^{(k-1)}, \ldots, \bar{\epsilon}^{(k-w)})^\top$, the equality recurrence can be written in matrix form as

$$\bar{\mathbf{e}}_{k+1} = \mathbf{C}\,\bar{\mathbf{e}}_k + \delta\,\mathbf{b}, \tag{17}$$

where $\mathbf{b} = (1, 0, \ldots, 0)^\top$ and $\mathbf{C}$ is the companion matrix from the proposition statement. By unrolling, $\bar{\mathbf{e}}_k = \delta \sum_{i=0}^{k-1} \mathbf{C}^i \mathbf{b}$. The spectral radius $\rho = \rho(\mathbf{C})$ governs the asymptotic growth of $\|\mathbf{C}^k\|$: for any $\varepsilon > 0$, there exists a constant $M_\varepsilon$ such that $\|\mathbf{C}^k\| \leq M_\varepsilon (\rho + \varepsilon)^k$ for all $k \geq 0$ (Gelfand's formula). Hence $\bar{\epsilon}^{(k)} \leq \|\bar{\mathbf{e}}_k\| = \mathcal{O}(\rho^k)$, so $\epsilon^{(H)} \leq \bar{\epsilon}^{(H)} = \mathcal{O}(\rho^H)$, which is exponential when $\rho > 1$. $\qquad\square$

**Remark 1** (Accuracy-robustness tradeoff and dependence on $K$). *The $k$-step rollout loss $\mathcal{L}_k$ (see equation 5) trains $P_\theta$ on inputs $\hat{z}$ that are $(k-1)$-order predictions. For small $k$, $\epsilon^{(k-1)}$ is small and the training inputs lie near the encoder manifold; for large $k$, $\epsilon^{(k-1)}$ may be large and the inputs drift off-manifold. Including large-$k$ losses thus improves robustness to out-of-distribution (predicted) inputs but may reduce accuracy on in-distribution (groundtruth) inputs, a phenomenon also observed in imitation learning (Ross et al., 2011) and model-based reinforcement learning (Talvitie, 2016; Asadi et al., 2019). The error bound in Proposition 1 involves two quantities that both depend on the converged predictor $P_\theta$, and therefore on the number of rollout steps $K$ used for training. We write $\delta_K$ and $\Lambda_K$, where the subscript indicates that the predictor was trained with the aggregate loss $\sum_{k=1}^{K} \mathcal{L}_k$. The test-time error satisfies $\epsilon_K^{(H)} \leq \delta_K (\Lambda_K^H - 1) / (\Lambda_K - 1)$.*

**Effect of $K$ on $\delta_K$.** *The predictor $P_\theta$ is a single function optimized under the aggregate loss $\sum_{k=1}^{K} \mathcal{L}_k$. Each term $\mathcal{L}_k$ trains $P_\theta$ on a different input distribution: $\mathcal{L}_1$ uses groundtruth embeddings $z \in \mathcal{M}$ (the encoder manifold), while $\mathcal{L}_k$ for $k > 1$ uses $(k-1)$-order predictions $\hat{z}$ that lie at distance $\epsilon^{(k-1)}$ from $\mathcal{M}$. With finite model capacity, distributing the optimization across input distributions that are increasingly far from $\mathcal{M}$ reduces specialization on $\mathcal{M}$ itself. We therefore expect $\delta_K$, the one-step error on groundtruth inputs, to be non-decreasing in $K$.*

**Effect of $K$ on $\Lambda_K$.** *Note that we use truncated backpropagation through time (TBPTT, see Section 4): the gradient of $\mathcal{L}_k$ does not flow back through all $k$ applications of $P_\theta$, so the multi-step loss does not directly penalize the Jacobian norm via the chain rule. Nevertheless, including $\mathcal{L}_k$ for large $k$ still reduces the effective Lipschitz constant in the region visited during rollout. To see why, note that $\mathcal{L}_k$ trains the predictor to map $(k-1)$-order predictions, which lie at distance $\epsilon^{(k-1)}$ from the encoder manifold $\mathcal{M}$, to the correct targets on $\mathcal{M}$. As a result, $P_\theta$ produces accurate outputs over a wider neighborhood of $\mathcal{M}$, which reduces the rate of variation of $P_\theta$ in directions away from $\mathcal{M}$, i.e., reduces the effective Lipschitz constant $\Lambda$ in the neighborhood of the rollout trajectory. We therefore conjecture that the effective $\Lambda_K$ is non-increasing in $K$ under favorable optimization conditions.*

**Environment-dependent tradeoff.** *The test-time error $\epsilon_K^{(H)} \leq \delta_K (\Lambda_K^H - 1) / (\Lambda_K - 1)$ reflects the competition between these two effects. For clean simulated dynamics (deterministic transitions, noise-free rendering), $\delta_1$ is small and $\Lambda_1$ is already moderate: increasing $K$ raises $\delta_K$ without proportionally reducing $\Lambda_K$, so the accuracy term dominates, which suggests the optimum is at small $K$, consistent with our experimental observations. For noisy real-world data (sensor noise, physical uncertainty, visual variability), $\delta_1$ is larger and $\Lambda_1$ is large: training on off-manifold inputs from large-$K$ rollouts reduces the effective $\Lambda_K$, which more than compensates the increase in $\delta_K$, consistent with the experimental shift of the optimum to larger $K$.*

**Remark 2** (Growth regimes and Lipschitz regularization). *Proposition 1 distinguishes three regimes: **(i)** $\Lambda < 1$ (contractive): errors saturate and long rollouts are stable; **(ii)** $\Lambda = 1$ (isometric): errors grow linearly with the horizon, $\epsilon^{(H)} \leq H\delta$; **(iii)** $\Lambda > 1$ (expansive): errors grow exponentially as $\mathcal{O}(\Lambda^H)$. For transformer-based predictors with residual connections, the per-block Jacobian is $I + J_{block}$, whose spectral norm is generically greater than one. Without explicit Lipschitz constraints, the overall constant satisfies $\Lambda \gg 1$, which places the predictor in regime **(iii)**. Applying spectral normalization (Miyato et al., 2018) to each layer constrains $\Lambda \approx 1$, transitioning the predictor to regime **(ii)** and reducing error growth from exponential to linear. In the sliding window case (Proposition 3), polynomial growth $\mathcal{O}(H^{m-1})$ can arise when the spectral radius $\rho = 1$ and the companion matrix $\mathbf{C}$ has a Jordan block of size $m > 1$; however, this is a degenerate (measure-zero) configuration that is unlikely to occur in practice. Exploring Lipschitz-constrained predictors to enable longer stable rollouts is an interesting direction for future work.*

# E    Planning environments and datasets

At train time, we normalize the action and optional proprioceptive input by subtracting the empirical mean and dividing by the empirical standard deviation, which are computed on each dataset. At planning time, we sample candidate actions in the normalized action space directly. When stepping the plan in the simulator, we thus denormalize the plan resulting from the optimization before stepping it in the environment. For comparability with V-JEPA-2-AC, we do not normalize actions for DROID. We stress that, in all environments considered, we are scarce in data, except for Push-T, where we have a bigger dataset, compared to the task complexity.

**Data slicing.**    We summarize dataset statistics in Table 8. Each *trajectory dataset* (set of trajectories) is transformed into a *dataset of trajectory slices*, with each slice of length $W + 1$ for training. We split, at the level of trajectories, the dataset into training and validation sets, with a ratio of 0.9/0.1. Within each split, we split trajectories as follows. For each trajectory, we take the maximum number of contiguous slices of length $W + 1$ that can be extracted from it. For a trajectory of length $L$, we thus have $L - (W + 1)f$ such slices. For DROID only, for consistency with V-JEPA-2-AC (Assran et al., 2025), we do not preslice the dataset into a slice dataset as above. This means that, after having split the set of trajectories into training and validation sets, at dataloading time, we sample uniformly a slice of length $W + 1$ from each trajectory among the $L - (W + 1)f$ valid slices. If the trajectory is shorter than $W + 1$ frames, we sample a new random trajectory index and retry, until a maximum number of attempts.

Hence, varying $W$ has a different effect on the dataset size for both the above categories. For DROID, $W$ does affect the actual number of unique trajectories seen, but does not affect the number of training iterations. For the other environments, $W$ would affect the number of training iterations, with a larger $W$ leading to fewer slices, thus smaller dataset and fewer iterations per epoch. In any case, we account for this by setting the number of training iterations per epoch to a fixed value, for all values of $W$. We choose this value to be the number of iterations for one epoch with the default $W = 3$ and total batch size detailed in Table 5, namely 3543 for Metaworld, 1139 for Maze, 7741 for Push-T, 418 for Wall and the default 300 for DROID, as in V-JEPA-2-AC. On DROID, setting too high $W$ leads to discarding videos shorter than $W + 1$ frames: at $W = 3$, we retain 99.2% of the dataset, at $W = 9$, 96.4%, and at $W = 14$, only 86%. For the Wall dataset, which has trajectories of length $L = 50$, we can only increase $W$ to 9, as this requires data slices of length $(W + 1)f = 10 \times 5 = 50$, where $f$ is the frameskip parameter. For datasets with longer trajectories (Push-T) or more data relative to task complexity (Maze), this effect of dataset reduction is not present and performance is essentially constant beyond $W = 3$.

Table 8: Datasets statistics. We denote the number of trajectories in the dataset under *Dataset Size*, the length of trajectories under *Traj. Len. L.*

|  | Dataset Size | Traj. Len. $T$ |
|---|---|---|
| PointMaze | 2000 | 100 |
| Push-T | 18500 | 100-300 |
| Wall | 1920 | 50 |
| Metaworld | 12600 | 100 |
| DROID | 8000 | 20-50 |

**DROID.**    We use the same dataloader as in V-JEPA-2-AC, which defines actions as delta in measured robot positions. One could either feed all three available cameras of DROID (left, right, wrist) simultaneously (e.g. by concatenating them) or alternatively to the model. We choose to use only one view point as simultaneous input. For training, we find that allowing the batch sampler to sample from either the left or right camera allows for slightly lower validation loss than using only one of them.

For evaluation, we collected a set of 16 videos with our own DROID setup, positioning the camera to closely match the left camera setup from the original DROID dataset. These evaluation videos specifically focus on

object interaction and arm navigation scenarios, allowing us to assess the model's performance on targeted manipulation tasks.

As discussed in Section 5.1, we define the *Action Score* as a rescaling of the opposite of the Action Error, namely $800(0.1 - E)$ if $E < 0.1$ else 0, where $E$ is the Action Error. We display the Action Score in all figures discussed in Section 5.2.

**Robocasa.** Robocasa (Nasiriany et al., 2024) is a simulation framework, based on Robosuite (Zhu et al., 2020), with several robotic embodiments, including the Franka Panda Arm, which is the robot used in the DROID dataset. Robocasa features over 2,500 3D assets across 150+ object categories and numerous interactable furniture pieces. The framework includes 100 everyday tasks and provides both human demonstrations and automated trajectory generation to efficiently expand training data. It is licensed under the MIT License.

We evaluate DROID models on Robocasa. The already existing pick-and-place tasks require too long horizons to be solved by our current planning procedure. Hence, we need to define custom easier pick-and-place tasks where the arm and target object start closer to the target position. To get a goal frame, we need to teleoperate a trajectory to obtain successful pick-and-place trajectories. We can then use the last frame of these trajectories as goal frame for planning. We needed to tune the camera view point to roughly correspond to the DROID left or right camera viewpoint, otherwise our models were not able to unroll well a sequence of actions. We also customize the gripper to use the same RobotiQ gripper as in DROID. We collect 16 such trajectories to form our evaluation set in the kitchen scene with various object classes. We define the "Reach" condition as having the end-effector at less than 0.2 (in simulator units, corresponding to roughly 5 cms in DROID) from the target object, the "Pick" condition as having lifted the object at more than 0.05 from its initial altitude, and the "Place" condition as having the object at less than 0.15 from the target position of the object. Our teleoperated trajectories all involve three segments, namely reaching the object (segment 1), picking it up (segment 2), and placing it at the target position (segment 3), delimited by these conditions. These three segments allow to define 6 subtasks, namely "Reach-Pick-Place", "Reach-Pick", "Pick-Place", "Reach", "Pick", and "Place". The success definition of each of these tasks is as follows:

- "Reach-Pick-Place": starting from the beginning of segment 1, success is 1 if the "Pick" and "Place" conditions are met.

- "Reach-Pick": starting from the beginning of segment 1, success is 1 if the "Pick" condition is met.

- "Pick-Place": starting from the beginning of segment 2, success is 1 if the "Place" condition is met.

- "Reach": starting from the beginning of segment 1, success is 1 if the "Reach" condition is met.

- "Pick": starting from the beginning of segment 2, success is 1 if the "Pick" condition is met.

- "Place": starting from the beginning of segment 3, success is 1 if the "Place" condition is met.

We focus on the "Place" and "Reach" tasks. Our models have low success rate on the "Pick" task, as they slightly misestimate the position of the end-effector, which proves crucial, especially for small objects.

To allow for zero-shot transfer from DROID to Robocasa, we perform 5 times action repeat of the actions outputted by our DROID model, since we trained on DROID sampled at 4 fps and the control frequency of Robocasa is 20 Hz. We also rescale the actions outputted by our planner to match the action magnitude of Robocasa, namely [-1, 1] for the delta position of the end-effector in cartesian space, and [0, 1] for the gripper closure.

**Metaworld.** The Metaworld (Yu et al., 2019) environment is licensed under the MIT License. The 42 Metaworld tasks we consider are listed in Table 9. We gather a Metaworld dataset via TD-MPC2 online agents trained on the visual and full state (39-dimensional) input from the Metaworld environment, on 42 Metaworld tasks, listed in Table 9. We launch the training of each TD-MPC2 agent for three seeds per task. The re-initialization of the environment at each new training episode is therefore different, even within a

given seed and task. This randomness governs the initial position of the arm and of the objects present in the scene, as well as the goal positions of the arm and potential objects. Each episode has length 100. We keep the first 100 episodes of each combination of seed and task, to limit the proportion of "expert" trajectories in the dataset, thus promoting data diversity. This results in 126 buffers, each of 100 episodes, hence 12600 episodes of length 100.

We introduce a planning evaluation procedure for each of the Metaworld tasks considered. These are long-horizon tasks that require to perform at least 60 actions to be solved, meaning it should be solvable if planning at horizon $H = 60/f$, if using frameskip $f$. This allows us to explore the use of JEPA-WMs in a context where MPC is a necessity. At planning time, we reset the environment with a different seed for each episode, randomizing the initial position of the arm, of the objects present in the scene, as well as the goal positions of the arm and potential objects. We then play the expert policy provided in the open-source Metaworld package for 100 steps. The last frame (and optionally proprioception) of this episode is set as the goal $o_g$ for the planning objective of equation 2. We then reset the environment again with the same random seed, and let the agent plan for 100 steps to reach the goal.

| Task | Description |
| --- | --- |
| turn on faucet | Rotate the faucet counter-clockwise. Randomize faucet positions |
| sweep | Sweep a puck off the table. Randomize puck positions |
| assemble nut | Pick up a nut and place it onto a peg. Randomize nut and peg positions |
| turn off faucet | Rotate the faucet clockwise. Randomize faucet positions |
| push | Push the puck to a goal. Randomize puck and goal positions |
| pull lever | Pull a lever down 90 degrees. Randomize lever positions |
| push with stick | Grasp a stick and push a box using the stick. Randomize stick positions. |
| get coffee | Push a button on the coffee machine. Randomize the position of the coffee machine |
| pull handle side | Pull a handle up sideways. Randomize the handle positions |
| pull with stick | Grasp a stick and pull a box with the stick. Randomize stick positions |
| disassemble nut | pick a nut out of the a peg. Randomize the nut positions |
| place onto shelf | pick and place a puck onto a shelf. Randomize puck and shelf positions |
| press handle side | Press a handle down sideways. Randomize the handle positions |
| hammer | Hammer a screw on the wall. Randomize the hammer and the screw positions |
| slide plate | Slide a plate into a cabinet. Randomize the plate and cabinet positions |
| slide plate side | Slide a plate into a cabinet sideways. Randomize the plate and cabinet positions |
| press button wall | Bypass a wall and press a button. Randomize the button positions |
| press handle | Press a handle down. Randomize the handle positions |
| pull handle | Pull a handle up. Randomize the handle positions |
| soccer | Kick a soccer into the goal. Randomize the soccer and goal positions |
| retrieve plate side | Get a plate from the cabinet sideways. Randomize plate and cabinet positions |
| retrieve plate | Get a plate from the cabinet. Randomize plate and cabinet positions |
| close drawer | Push and close a drawer. Randomize the drawer positions |
| press button top | Press a button from the top. Randomize button positions |
| reach | reach a goal position. Randomize the goal positions |
| press button top wall | Bypass a wall and press a button from the top. Randomize button positions |
| reach with wall | Bypass a wall and reach a goal. Randomize goal positions |
| insert peg side | Insert a peg sideways. Randomize peg and goal positions |
| pull | Pull a puck to a goal. Randomize puck and goal positions |
| push with wall | Bypass a wall and push a puck to a goal. Randomize puck and goal positions |
| pick out of hole | Pick up a puck from a hole. Randomize puck and goal positions |
| pick&place w/ wall | Pick a puck, bypass a wall and place the puck. Randomize puck and goal positions |
| press button | Press a button. Randomize button positions |
| pick&place | Pick and place a puck to a goal. Randomize puck and goal positions |
| unplug peg | Unplug a peg sideways. Randomize peg positions |
| close window | Push and close a window. Randomize window positions |
| open door | Open a door with a revolving joint. Randomize door positions |
| close door | Close a door with a revolving joint. Randomize door positions |
| open drawer | Open a drawer. Randomize drawer positions |
| close box | Grasp the cover and close the box with it. Randomize the cover and box positions |
| lock door | Lock the door by rotating the lock clockwise. Randomize door positions |
| pick bin | Grasp the puck from one bin and place it into another bin. Randomize puck positions |

Table 9: A list of all of the Meta-World tasks and a description of each task.

**Push-T.** In this environment introduced by (Chi et al., 2023) (MIT License), a pusher ball agent interacts with a T-shaped block. Success is achieved when both the agent and the T-block, which start from a randomly initialized state, reach a target position. For Push-T, the dataset provided in DINO-WM is made of 18500 samples, replays of the original released expert trajectories with various levels of noise. At evaluation time, we sample an initial and goal state from the validation split, such that the initial state attains the goal in $H$ steps, with $H$ the planning horizon. Indeed, otherwise, the task can require very long-horizon planning, and is not well solved with our planners.

**PointMaze.** In this environment introduced by (Fu et al., 2020) (Apache 2.0 license), a force-actuated 2-DoF ball in the Cartesian directions $x$ and $y$ must reach a target position. The agent's dynamics incorporate its velocity, acceleration, and inertia, making the movement realistic. The PointMaze train set is made of 2000 fully random trajectories. At evaluation time, we sample a random initial and goal state from the simulator's sampler.

**Wall.** This 2D navigation environment introduced in (Zhou et al., 2024a) (MIT License) features two rooms separated by a wall with a door. The agent's task is to navigate from a randomized starting location in one room to a goal in one of the two rooms, potentially passing through the door. The Wall dataset is made of 1920 random trajectories each with 50 time steps. At planning time, we also sample a random initial and goal state from the simulator's sampler.

# F  Planning Optimization

In this section, we detail the optimization procedures for planning in our experiments. Given a modeling function $F_{\phi,\theta}$, a dissimilarity criterion $(L_{vis} + \alpha L_{prop})$, and an initial and goal observation pair $o_t, o_g$, we recall the objective function $L_\alpha^p(o_t, a_{t:t+H-1}, o_g) = (L_{vis} + \alpha L_{prop})(F_{\phi,\theta}(o_t, a_{t:t+H-1}), E_{\phi,\theta}(o_g))$.

**Model Predictive Control.**  In Metaworld only we perform MPC, a procedure where replanning is allowed after executing a plan in the environment. We set the maximum number of actions that can be stepped in the environment to 100, which constitutes an episode. At each step of the episode where we plan, we use either the CEM or NG planner.

**Cross-Entropy Method.**  The CEM optimisation algorithm proceeds as in Algorithm 1. In essence, we fit parameters of a time-dependent multivariate Gaussian with diagonal covariance.

---

**Algorithm 1** Cross-Entropy Method

---

1: $\mu^0 \in \mathbb{R}^{H \times A}$ is zero and covariance matrix $\sigma^0 I \in \mathbb{R}^{(H \times A)^2}$ is the identity. Number of optimisation steps $J$.

2: **for** $j = 1$ to $J$ **do**

3:    Sample $N$ independent trajectories $(\{a_t, \ldots, a_{t+H-1}\}) \sim \mathcal{N}(\mu^j, (\sigma^j)^2 I)$

4:    For each of the $N$ trajectories, unroll predictor to predict the resulting trajectory, $\hat{z}_i = P_\theta(\hat{z}_{i-1}, a_{i-1}), \quad i = t+1, \ldots, t+H$. Compute cost $L_\alpha^p(o_t, a_{t:t+H-1}, o_g)$ for each candidate trajectory.

5:    Select top $K_e$ action sequences with the lowest cost, denote them $(\{a_t, \ldots, a_{t+H-1}\})_{1,\ldots,K_e}$. Update

$$\mu^{j+1} = \frac{1}{K_e} \sum_{k=1}^{K_e} (\{a_t, \ldots, a_{t+H-1}\})_k$$

$$\sigma^{j+1} = \sqrt{\frac{1}{K_e - 1} \sum_{k=1}^{K_e} [(\{a_t, \ldots, a_{t+H-1}\})_k - \mu^{j+1}]^2}$$

6: **end for**

7: Step the first $m$ actions of $\mu^J$, where $m \leq H$ is a planning hyperparameter in the environment. If we are in MPC mode, the process then repeats at the next time step with the new context observation.

---

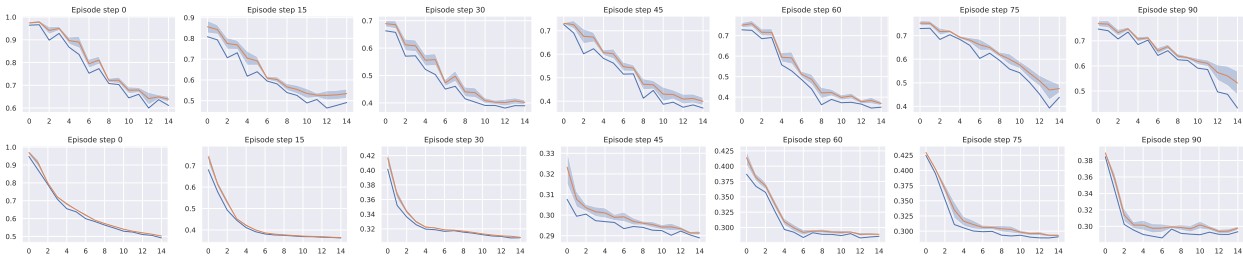

Figure 9: Planning a 100-steps Metaworld episode with the base DINO-WM at the end of training of WM, for the same Metaworld environment episode seed, with our two planners. We display the average objective of the top $K_e$ imagined trajectories and its standard deviation (orange), and the best imagined trajectory's planning loss $L_\alpha^p$ (blue). Bottom: Planning with CEM. Top: Planning with NG. Failure episode for both: with NG the arms stays stuck against the wall, hence the higher planning objective, whereas with CEM this episode fails because of imprecision around the goal position.

**NG Planner.**  We design a procedure to use any NeverGrad optimizer with our planning objective $L_\alpha^p(o_t, a_{t:t+H-1}, o_g)$, with the same number of action trajectories evaluated in parallel and total budget as CEM, as detailed in Algorithm 2. As discussed in Section 5.2, in all the evaluation setups we consider in

this study, the NGOpt meta-optimizer always chooses the diagonal variant of the CMA-ES algorithm with a particular parameterization. The diagonal version of CMA is advised when the search space is big. We stress that after trying other parameterizations of the Diagonal CMA algorithm, like its elitist version (with a scale factor of 0.895 instead of 1, which is the default value), success rate can drop by 20% on Wall, Maze and Push-T.

---

**Algorithm 2** NeverGrad planner

---

1: `optimizer` chosen by `nevergrad.optimizers.NGOpt` from budget $N \times J$, on space $\mathbb{R}^{H \times A}$, with $N$ workers.
2: **for** $j = 1$ to $J$ **do**
3:     `optimizer.ask()` $N$ trajectories sequentially.
4:     For each of the $N$ trajectories, unroll predictor to predict the resulting trajectory, $\hat{z}_i = P_\theta(\hat{z}_{i-1}, a_{i-1}), \quad i = t+1, \ldots, t+H$. Compute cost $L_\alpha^p(o_t, a_{t:t+H-1}, o_g)$ for each candidate trajectory.
5:     `optimizer.tell()` the cost $L_\alpha^p(o_t, a_{t:t+H-1}, o_g)$ of the $N$ trajectories sequentially.
6: **end for**
7: Step the first $m$ actions of `optimizer.provide_recommendation()`, where $m \leq H$ is a planning hyper-parameter in the environment. If we are in MPC mode, the process then repeats at the next time step with the new context observation.

---

**Gradient Descent Planner.** We also experiment with gradient-based planners that directly optimize the action sequence through backpropagation. Unlike sampling-based methods (CEM, NG), these planners leverage the differentiability of the world model to compute gradients of the planning objective $L_\alpha^p$ with respect to actions. The Gradient Descent (GD) planner initializes actions either from a standard Gaussian (scaled by $\sigma_0$) or from zero, then performs $J$ gradient descent steps with learning rate $\lambda$. After each gradient step, Gaussian noise with standard deviation $\sigma_{\text{noise}}$ is added to the actions to encourage exploration and help escape local minima. Action clipping is applied to enforce bounds on specific action dimensions. The default hyperparameters are $J = 500$ iterations, $\lambda = 1$, $\sigma_0 = 1$, and $\sigma_{\text{noise}} = 0.003$.

**Adam Planner.** The Adam planner extends the GD planner by using the Adam optimizer instead of vanilla stochastic gradient descent. Adam maintains exponential moving averages of the gradient (first moment) and squared gradient (second moment), which can provide more stable optimization dynamics. The default hyperparameters are $\beta_1 = 0.9$, $\beta_2 = 0.995$, $\epsilon = 10^{-8}$, with the same defaults as GD for other parameters ($J = 500$, $\lambda = 1$, $\sigma_0 = 1$, $\sigma_{\text{noise}} = 0.003$).

**Planning hyperparameters.** We display in Table 10 the hyperparameters used to plan on each environment. We keep the planning hyperparameters of DINO-WM (Zhou et al., 2024a) for Push-T, Wall and Maze, but reduce the number of "top" actions, denoted $K_e$, to 10 instead of 30. We obtain these parameters after careful grid search on DINO-WM. The success rate is very sensitive to these parameters, keeping the world model fixed.

Table 10: Environment-specific hyperparameters for planning, corresponding to the notations of Section F. The number of steps per planning episode is denoted $M$ and the frameskip is denoted $f$. $H$ is the planning horizon, $m$ the number of actions to step in the environment, $K_e$ the number of top actions in CEM, $N$ the number of trajectories evaluated in parallel, $J$ the number of iterations of the optimizer. The total number of replanning steps for an evaluation episode is $\frac{M}{fm}$.

|           | $N$ | $H$ | $m$ | $K_e$ | $J$ | $W^p$ | $f$ | $M$ |
|-----------|-----|-----|-----|-------|-----|-------|-----|-----|
| PointMaze | 300 | 6   | 6   | 10    | 30  | 2     | 5   | 30  |
| Push-T    | 300 | 6   | 6   | 10    | 30  | 2     | 5   | 30  |
| Wall      | 300 | 6   | 6   | 10    | 30  | 2     | 5   | 30  |
| Metaworld | 300 | 6   | 3   | 10    | 15  | 2     | 5   | 100 |
| Robocasa  | 300 | 3   | 1   | 10    | 15  | 2     | 5   | 60  |
| DROID     | 300 | 3   | 3   | 10    | 15  | 2     | 1   | $m$ |

## G  Additional experiments

### G.1  Additional results

**Equalized action ratio experiments.**  To isolate the effect of the conditioning scheme from capacity differences due to action ratio, we train new models for each of the considered environments where we downscale the image resolution to $128 \times 128$ with a DINOv3-L encoder (patch size 16), yielding $8 \times 8 = 64$ visual patches. This produces matched action ratios: sequence conditioning achieves $1/(hw+1) = 1/65$, while feature conditioning achieves $f_a/(D + f_a) = 16/1040$. Both variants use the same predictor architecture, RoPE positional encoding, frozen encoder, and training data.

*Results reveal task-dependent preferences*:  Sequence conditioning outperforms feature conditioning on DROID, Robocasa, and Metaworld (3D manipulation tasks). Conversely, feature conditioning significantly outperforms on Wall (2D navigation). Performance is comparable on Maze and Push-T. Notably, the Wall result replicates the trend from Figure 5a (different resolution/model size), confirming statistical robustness.

We cannot provide a precise explanation of the underlying mechanism explaining why we observe such differences. What we can say for sure is that, despite matched action ratios, the conditioning schemes differ fundamentally in how action information propagates:

- *Feature conditioning*: Action embeddings are concatenated to each visual token. At the first transformer block, action-visual mixing occurs only within the MLP (after self-attention), which combines information across embedding dimensions but processes each spatial token separately.

- *Sequence conditioning*: The action token participates in self-attention from the first block, allowing immediate broadcasting to all visual tokens through attention mechanisms.

*Observations on Wall task*: Training and validation rollout losses are identical between methods, indicating both models predict dynamics equally well. However, during planning, sequence-conditioned models occasionally select very high magnitude actions, while feature-conditioned models consistently choose moderate actions. This suggests the conditioning scheme affects not only prediction but also the structure of the learned latent dynamics relevant for planning.

We hypothesize that for spatially simple tasks (Wall), feature conditioning's direct action-to-token pathway yields dynamics better suited for planning optimization, while sequence conditioning's attention-based routing is more beneficial for complex spatial reasoning in manipulation. However, further analysis would be needed to conclusively establish this mechanism.

**Additional planner ablations.**  In Table 11, we compare the performance of our model to the DINO-WM and VJEPA-2-AC baselines across all planner configurations tested in Figure 3. Our optimal JEPA-WM consistently outperforms the baselines on most tasks and planners, as in Table 2.

Table 11: Comparison of different models across all planner configurations. MW-R and MW-RW denote the Reach and Reach-Wall tasks of Metaworld. Rc-Pl and Rc-R denote the Place and Reach tasks of Robocasa. In **bold** is the best of the three models for a given planning optimizer, underlined is the best across all planning optimizers.

| Model | Planner | Maze | Wall | Push-T | MW-R | MW-RW | Rc-R | Rc-Pl | DROID |
|---|---|---|---|---|---|---|---|---|---|
| CEM $L_2$ | DWM | 81.6 (3.4) | 64.1 (4.6) | 66.0 (4.7) | 44.8 (8.9) | 35.1 (9.4) | 19.1 (13.4) | 21.7 (7.2) | 39.4 (2.1) |
| | VJ2AC | — | — | — | — | — | 16.2 (8.3) | **33.1 (7.2)** | 42.9 (2.5) |
| | Ours | **83.9 (2.3)** | **78.8 (3.9)** | **70.2 (2.8)** | **58.2 (9.3)** | **41.6 (10.0)** | **25.4 (16.6)** | 30.7 (8.0) | **48.2 (1.8)** |
| CEM $L_1$ | DWM | 78.8 (2.9) | **48.7 (4.0)** | 61.6 (4.5) | 45.1 (10.4) | 34.0 (9.1) | 14.4 (11.3) | 19.6 (7.4) | 41.7 (2.7) |
| | VJ2AC | — | — | — | — | — | 13.2 (10.3) | **30.9 (7.3)** | 38.5 (5.7) |
| | Ours | **79.7 (3.1)** | 46.7 (3.6) | **63.4 (2.1)** | **55.1 (8.5)** | **40.8 (9.2)** | **17.6 (12.3)** | 30.5 (8.7) | **47.2 (1.8)** |
| NG $L_2$ | DWM | 54.2 (3.8) | 25.3 (4.2) | 47.6 (5.4) | 28.1 (7.8) | **27.9 (10.0)** | 25.8 (16.7) | 27.1 (8.6) | 36.0 (3.6) |
| | VJ2AC | — | — | — | — | — | 22.1 (8.9) | 31.1 (7.0) | 36.2 (3.2) |
| | Ours | **72.7 (4.5)** | **35.4 (5.9)** | **48.0 (3.0)** | **29.3 (8.6)** | 21.0 (7.7) | **30.5 (10.7)** | **36.9 (9.1)** | **39.9 (1.9)** |
| NG $L_1$ | DWM | 52.3 (4.0) | 24.6 (5.2) | 46.2 (5.1) | 27.5 (8.5) | **28.6 (8.9)** | 21.6 (15.5) | 26.0 (8.0) | 35.4 (3.3) |
| | VJ2AC | — | — | — | — | — | 18.4 (11.8) | 29.4 (5.1) | 33.7 (3.9) |
| | Ours | **69.5 (2.1)** | **30.9 (4.1)** | **48.2 (5.1)** | **27.8 (8.2)** | 21.3 (8.9) | **25.7 (12.0)** | **32.0 (9.9)** | **40.0 (2.2)** |
| Adam $L_2$ | DWM | 14.8 (1.5) | 0.1 (0.3) | 8.0 (2.4) | **62.0 (8.3)** | 49.9 (9.9) | **2.8 (3.7)** | **21.5 (8.6)** | 0.0 (0.0) |
| | VJ2AC | — | — | — | — | — | 0.0 (0.0) | 5.9 (5.1) | 0.0 (0.0) |
| | Ours | **17.7 (3.3)** | **1.1 (1.1)** | **9.0 (2.1)** | 55.3 (9.9) | 39.7 (9.7) | 0.0 (0.0) | 10.5 (5.0) | 0.0 (0.0) |
| Adam $L_1$ | DWM | 12.6 (2.8) | **0.1 (0.3)** | **2.1 (1.5)** | 50.7 (8.5) | 37.2 (8.3) | **0.9 (1.4)** | **22.1 (7.5)** | 0.0 (0.0) |
| | VJ2AC | — | — | — | — | — | 0.0 (0.0) | 3.5 (2.8) | 0.0 (0.0) |
| | Ours | **16.1 (3.5)** | 0.0 (0.2) | 0.9 (0.9) | **53.2 (7.4)** | **43.1 (6.4)** | 0.2 (0.7) | 7.5 (3.9) | 0.0 (0.0) |
| GD $L_2$ | DWM | 14.5 (1.9) | 0.1 (0.3) | 8.1 (2.4) | **62.2 (8.2)** | **50.0 (10.1)** | **2.9 (3.8)** | **21.9 (8.4)** | 0.0 (0.0) |
| | VJ2AC | — | — | — | — | — | 0.0 (0.0) | 8.1 (3.6) | 0.0 (0.0) |
| | Ours | **17.7 (3.3)** | **1.1 (1.1)** | **9.1 (1.9)** | 55.3 (9.9) | 39.7 (9.7) | 0.0 (0.0) | 10.5 (5.0) | 0.0 (0.0) |
| GD $L_1$ | DWM | 12.4 (2.6) | **0.1 (0.3)** | **2.0 (1.3)** | 50.9 (8.5) | 37.1 (8.4) | **1.0 (1.4)** | **22.2 (7.3)** | 0.0 (0.0) |
| | VJ2AC | — | — | — | — | — | 0.0 (0.0) | 4.0 (2.6) | 0.0 (0.0) |
| | Ours | **16.1 (3.5)** | 0.0 (0.2) | 0.9 (0.9) | **53.3 (7.3)** | **40.1 (12.3)** | 0.2 (0.7) | 7.5 (3.9) | 0.0 (0.0) |

For completeness, we also report in Table 12 the per-seed variance at the final checkpoint, which isolates reproducibility across random seeds from training stability. Results are consistent with Table 11, confirming that our method reliably outperforms baselines across both metrics.

In Figure 11 (Left), we display Figure 3 again for completeness, along with Figure 11 (Right), where we compare the performance of all planners on all environments but *with proprioception*. We adopt the default configuration described in the beginning of Section 4, namely DINO-WM ViT-S but with proprioception. We can draw the same conclusions as for without proprioception, in Figure 3.

Let us detail the failure cases of the GD planner. On the Wall task, the GD planner gets zero performance, although the task is visually simplistic. We identify two main failure cases. Either the agent goes into the wall without being able to pass the door, which is a classical failure case for better CEM or NG planners. Or the agent finds a local planning cost minimum by going to the borders of the image, when starting close to them. We illustrate both of these in Figure 10.

**Object manipulation on Robocasa and DROID.** We show in Figure 13 a successful planning episode with our proposed JEPA-WM on Robocasa on the "Place" task. Our model is able to perform the "Place" task but has a much lower success rate at the "Pick" task, as it misestimates the position of the arm. We illustrate the shift in camera calibration / action calibration in Figure 12 on the "Reach" task. On all episodes, the model always predicts a state shifted to the left compared to the ground-truth. This phenomenon is less clear on DROID, as we illustrate in Figure 14.

**Object manipulation on Metaworld.** Our agent solves the pose control tasks like reach and reach-wall. In addition, long-term action unrolling with object interaction seems to be well-captured by our models, as shown in Figure 15 for the bin-picking task. Yet, for tasks involving object manipulation, it hallucinates grasping the object. In Figure 16, the visual decoding of the unrolling of the action plan shows a gap between

Table 12: Comparison of the final checkpoint of three methods, displaying the variance across three seeds, for all planner configurations. MW-R and MW-RW denote the Reach and Reach-Wall tasks of Metaworld. Rc-Pl and Rc-R denote the Place and Reach tasks of Robocasa. In **bold** is the best of the three models for a given planning optimizer, underlined is the best across all planning optimizers.

| Model | Planner | Maze | Wall | Push-T | MW-R | MW-RW | Rc-R | Rc-Pl | DROID |
|---|---|---|---|---|---|---|---|---|---|
| CEM $L_2$ | DWM | 79.2 (2.9) | 63.1 (2.3) | 63.3 (3.9) | 44.4 (5.5) | **31.9 (2.6)** | 11.5 (7.8) | 20.8 (9.7) | 40.4 (1.6) |
| | VJ2AC | — | — | — | — | — | 7.3 (6.4) | 33.3 (5.9) | 43.2 (1.9) |
| | Ours | **83.3 (3.1)** | **80.9 (4.9)** | **69.4 (4.0)** | **49.0 (9.4)** | 29.2 (11.5) | **22.9 (21.4)** | **34.4 (7.7)** | **46.5 (0.4)** |
| CEM $L_1$ | DWM | 78.3 (2.1) | **50.0 (3.7)** | **60.6 (2.2)** | **49.3 (5.5)** | **38.2 (10.4)** | 13.5 (8.2) | 14.6 (5.3) | 41.6 (1.8) |
| | VJ2AC | — | — | — | — | — | 8.3 (3.9) | **31.2 (6.8)** | 40.9 (3.1) |
| | Ours | **81.2 (4.7)** | 47.6 (2.1) | 55.9 (3.0) | 48.4 (10.7) | 30.7 (7.0) | **20.8 (22.9)** | 29.2 (11.8) | **45.0 (0.3)** |
| NG $L_2$ | DWM | 55.6 (2.3) | 25.2 (4.3) | 48.8 (3.0) | **25.7 (2.0)** | **27.1 (1.7)** | 20.8 (7.8) | 33.3 (8.2) | 35.4 (2.2) |
| | VJ2AC | — | — | — | — | — | 18.8 (4.4) | 29.2 (2.9) | 35.7 (2.1) |
| | Ours | **76.4 (0.5)** | **38.9 (4.2)** | **49.7 (2.7)** | 23.4 (8.4) | 13.0 (5.6) | **26.0 (16.6)** | **35.4 (6.4)** | **39.8 (0.7)** |
| NG $L_1$ | DWM | 52.3 (2.0) | 23.1 (6.6) | 42.1 (4.4) | **31.2 (6.1)** | **29.2 (3.4)** | **24.0 (8.2)** | 28.1 (15.9) | 38.4 (3.2) |
| | VJ2AC | — | — | — | — | — | 9.4 (2.6) | 28.1 (2.6) | 34.0 (4.9) |
| | Ours | **73.6 (3.0)** | **30.6 (1.3)** | **43.8 (0.9)** | 27.6 (6.5) | 12.5 (6.4) | 18.8 (19.9) | **32.3 (12.6)** | **39.9 (1.3)** |
| Adam $L_2$ | DWM | 14.6 (3.1) | 0.0 (0.0) | **8.9 (0.5)** | **60.4 (2.9)** | **44.4 (7.7)** | **7.3 (2.9)** | **24.0 (10.3)** | 0.0 (0.0) |
| | VJ2AC | — | — | — | — | — | 0.0 (0.0) | 12.5 (5.1) | 0.0 (0.0) |
| | Ours | **21.5 (4.2)** | **1.0 (0.9)** | 8.3 (0.9) | 45.8 (3.9) | 25.5 (10.6) | 3.1 (3.1) | 9.4 (3.1) | 0.0 (0.0) |
| Adam $L_1$ | DWM | 12.0 (1.6) | 0.0 (0.0) | **3.1 (1.0)** | 45.8 (6.8) | **34.7 (5.2)** | **3.1 (4.4)** | **21.9 (0.0)** | 0.0 (0.0) |
| | VJ2AC | — | — | — | — | — | 0.0 (0.0) | 4.2 (1.5) | 0.0 (0.0) |
| | Ours | **17.7 (1.5)** | 0.0 (0.0) | 1.0 (0.0) | **45.8 (2.6)** | 33.9 (3.1) | 0.0 (0.0) | 7.8 (4.7) | 0.0 (0.0) |
| GD $L_2$ | DWM | 13.5 (2.1) | 0.0 (0.0) | 8.3 (1.0) | 61.8 (2.0) | 45.8 (6.8) | **7.3 (2.9)** | **25.0 (9.2)** | 0.0 (0.0) |
| | VJ2AC | — | — | — | — | — | 0.0 (0.0) | 5.2 (1.5) | 0.0 (0.0) |
| | Ours | **21.5 (4.2)** | **1.0 (0.9)** | **8.3 (0.9)** | 45.8 (3.9) | 25.5 (10.6) | 3.1 (3.1) | 9.4 (3.1) | 0.0 (0.0) |
| GD $L_1$ | DWM | 12.0 (1.6) | 0.0 (0.0) | **2.1 (0.0)** | 45.8 (6.8) | **34.7 (5.2)** | **3.1 (4.4)** | **21.9 (0.0)** | 0.0 (0.0) |
| | VJ2AC | — | — | — | — | — | 0.0 (0.0) | 4.2 (1.5) | 0.0 (0.0) |
| | Ours | **17.7 (1.5)** | 0.0 (0.0) | 1.0 (0.0) | **45.8 (2.6)** | 33.9 (3.1) | 0.0 (0.0) | 7.8 (4.7) | 0.0 (0.0) |

the imagined consequences of the actions and their consequences in the simulator. This calls for a separate optimization procedure for the action dimension that corresponds to the end-effector's gripper.

**Action precision Push-T.** The results in Figure 21 clearly show that CEM performs better than NG for the Push-T task. The Push-T task requires very precise actions, since if the ball is slightly off the position where it should be to push the T-shape, it misses the shape and fails at the task. Hence it proves essential to only step actions after convergence of the planner. Yet, we see in Figure 17 that the NG planner is more explorative and should therefore be parametrized differently for this type of task. Interestingly, the larger model converges faster and brings higher maximal success rate on this task.

**Embedding space and model size.** We see in Figure 18 that the relative difference in embedding space distance to the goal is approximately ten times smaller in the ViT-L model than in the ViT-S model. This is consistent with the fact that the ViT-L model has a higher capacity and can therefore embed more information in its latent space, in which two states of Metaworld are closer to each other than in the ViT-S model embedding space.

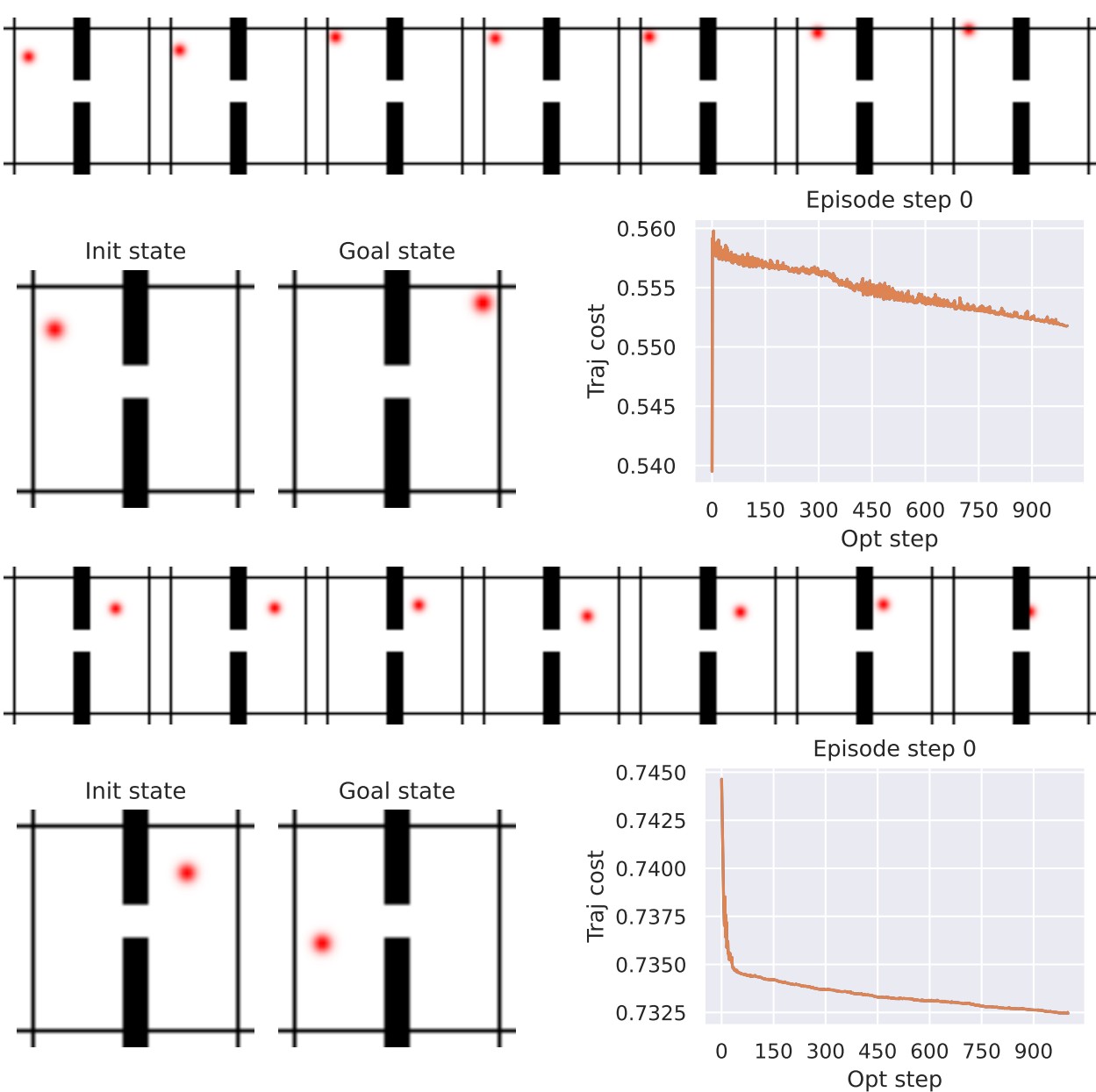

Figure 10: Two typical failure cases with the Gradient-Descent (GD) planner on the Wall task. For each failure case, we show: (top) the planned trajectory visualization, (bottom left) initial and goal states, (bottom right) planning cost evolution throughout gradient descent iterations. First failure case (top 3 subfigures): the agent finds a local planning cost minimum by going to the borders of the image when starting close to them. Second failure case (bottom 3 subfigures): the agent goes into the wall without being able to pass the door.

## G.2 Evaluation metrics

**Statistical significance.** To account for the evaluation variability, at each epoch, we launch $e = 96$ episodes, each with a different initial and goal state, either sampled from the dataset (Push-T, Robocasa, DROID) or by the simulator (Metaworld, PointMaze, Wall). We take $e = 64$ for evaluation on DROID, which proves essential to get a reliable evaluation, even though we compare a continuous action score metric. We use $e = 32$ for Robocasa given the higher cost of a planning episode, which requires replanning 12 times, as

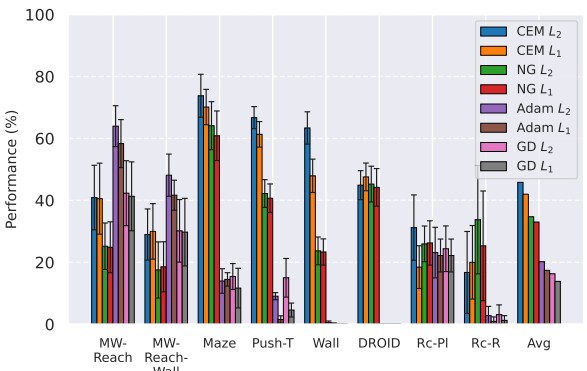
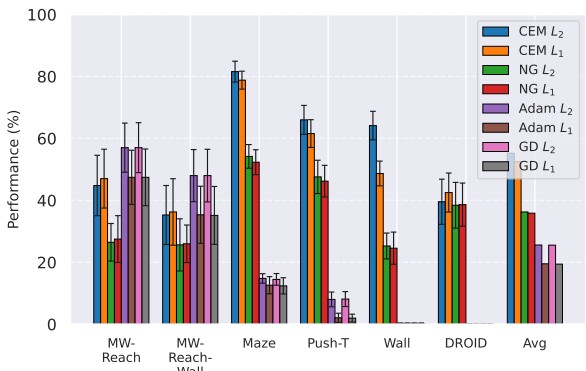

Figure 11: Comparison of planning optimizers. NG is the Nevergrad-based interface for trajectory optimization that we introduce, CEM is the Cross-Entropy Method, and GD is the Gradient Descent Planner with $L_1$ or $L_2$ distance. Left: comparison of the GD, CEM and NG planners on DINO-WM without proprioception. Right: comparison of CEM and NG planners on DINO-WM with proprioception.

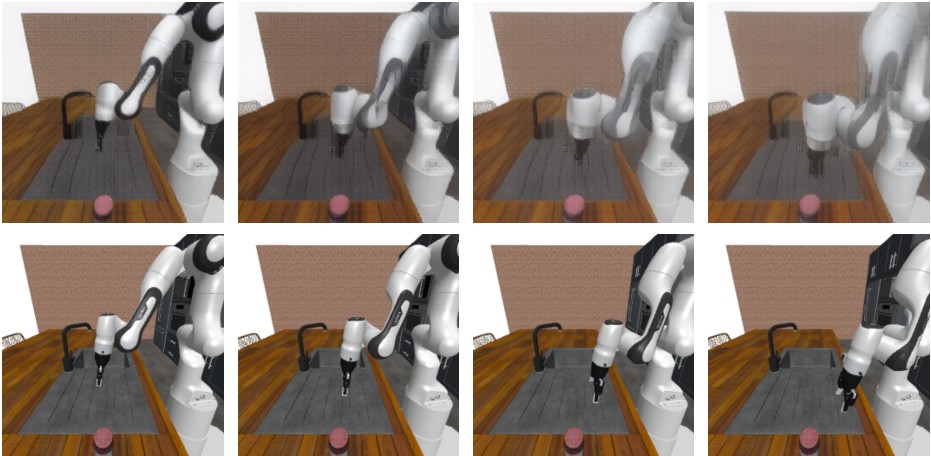

Figure 12: Planning at horizon 3 with our proposed JEPA-WM on Robocasa, with our best model presented in Section 5.3. The model is trained on DROID and evaluated zero-shot on Robocasa on the "Reach" task, where the goal is to reach the object. Top: the model's visual decoding of the action plan. Bottom: the ground-truth action stepping in simulator. The model predicts a state shifted to the left compared to the ground-truth.

explained in Table 10. We average over these episodes to get a success rate. Although we average success at each epoch over three seeds and their evaluation episodes, we still find high variability throughout training. Hence, to get an aggregate score per model, we average success over the last $n$ training epochs, with $n = 10$ for all datasets, except for models trained on DROID, for which $n = 100$. The error bars displayed in the plots comparing design choices are the standard deviation across the last epochs' success rate, to reflect this variability only.

The metric we seek to optimize for planning tasks is the success rate, which can be noisy, highly dependent on the initial seed, and sparse at the beginning of training. We therefore derive several other useful metrics and study their correlation with the success rate.

**Embedding space error throughout unrolling.** Throughout training, every 300 training iterations, we unroll the predictor on a batch of the validation split for $n$ steps. For each of these steps, we compute the $L_1$ and $L_2$ loss between the predicted embedding and the embedding of the ground truth corresponding future frame. The $L_2$ loss at step 1 is the teacher-forcing loss term $\mathcal{L}$.

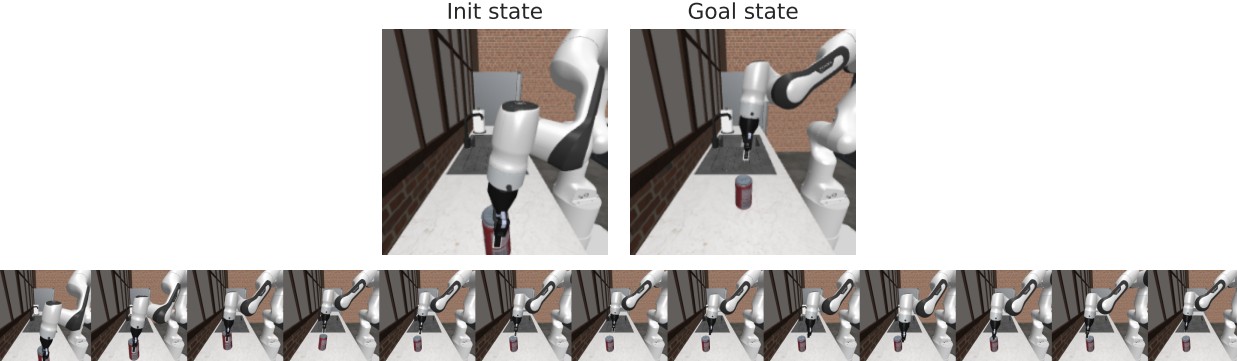

Figure 13: Planning episode with our proposed JEPA-WM on Robocasa, with DINOv2 ViT-S encoder, 2-step rollout, AdaLN predictor. The model is trained on DROID and evaluated zero-shot on Robocasa on the 'place' task. The planning cost is the embedding space distance to the goal frame embedding.

**Proprioceptive decoding error throughout unrolling.** Prior to training of the predictors, we train a small ViT probe, called "state decoder" on top of the frozen encoder to regress the state, i.e. proprioception and optionally other simulator state information. Then, when training the predictors we study in this paper, every 300 training iterations, we unroll the predictor for $n$ steps on a batch of the validation split and use our state decoder on the predicted features. This yields $n$ state predictions, of which we compute the distance to the ground truth future $n$ states.

**Visual decoder.** Just like the state decoder, we train a visual decoder to reconstruct, from the embeddings outputted by the frozen encoder, the associated frames. We decode each frame independently to avoid artificial consistency across frames, as the decoder is a probing tool. Indeed, in models like COSMOS (Agarwal et al., 2025), the powerful diffusion decoder accounts for beautiful visualisations, although the underlying latent world model might not be as accurate. Every 300 training iterations, we unroll the predictor on a given sequence of actions, and compare the decoding of the predicted embeddings to the ground truth future frames, both qualitatively and with the LPIPS metric (Zhang et al., 2018).

**Action error.** To evaluate models without having to step actions in a real robot or a simulator, we compare the actions outputted by the planner and the groundtruth actions of the trajectory sampled from the dataset to define initial and goal state. On DROID, throughout training, the total action error increases, and even more if we do not clip the actions magnitude in the planner, as done in V-JEPA-2-AC, since we do not normalize the actions. This is because most of the action error comes from the gripper closure and gripper orientation action dims, as detailed in Figure 19. Hence, on DROID, we track the action error on the three first dimensions, corresponding to the end-effector position control, which is more relevant for the tasks we consider.

### G.3 Is there a proxy for success rate?

Evaluating several independent planning recipes is compute-intensive, even more so as the model size increases, as well as the planning budget $N \times H \times J \times W^p$, see Table 10. Hence, we look for validation metrics to track throughout the training that correlate well with the success rate. Each epoch of each model and evaluation setup (among four) is a data point with a value for a validation metric and an associated success rate. Considering each epoch as an independent sample allows us to compute the Spearman correlation between each quantitative metric and the success rate. The results in Table 13, Table 16, Table 14 and Table 15 first show that the correlation with training loss (Vis emb) is higher for the easier Wall task. Since we want to find the metric that correlates most to the success rate, we average the Spearman correlations instead of computing them on the union of data points, to avoid Simpson's paradox. This yields the rightmost column of each table. In both environments, the metric most correlated with the success rate is the planning objective, that is, the Vis Emb loss. Interestingly, only in Metaworld, which requires better long-horizon unrolling capability, do the unroll metrics at step $H > 1$ correlate better than step-1 metrics.

Table 13: Negative Spearman Correlation Coefficients between smoothed success rate and several validation metrics on Metaworld. In **bold** is highest value, underlined is 2nd highest. We denote the visual embedding prediction errors Vis Emb and the proprioceptive decoding error Proprio dec, at horizons $H$ from one to three. We display the mean success rate over the last 10 epochs averaged over the four eval setups in the last row.

| Model name | WM | $\text{WM}_W$ | WM-prop | WM-2-step | WM-L | Mean |
|---|---|---|---|---|---|---|
| Proprio dec $H = 1$ | 0.40 | 0.44 | 0.40 | 0.26 | 0.20 | 0.34 |
| Proprio dec $H = 2$ | 0.41 | 0.34 | 0.45 | 0.27 | 0.21 | 0.34 |
| Proprio dec $H = 3$ | 0.38 | 0.44 | 0.40 | 0.33 | 0.18 | 0.35 |
| Vis emb $L_2$ $H = 1$ | 0.44 | 0.51 | 0.62 | 0.39 | 0.23 | 0.44 |
| Vis emb $L_2$ $H = 2$ | 0.42 | 0.46 | 0.62 | 0.39 | 0.24 | 0.42 |
| Vis emb $L_2$ $H = 3$ | 0.36 | 0.44 | 0.59 | 0.37 | 0.20 | 0.39 |
| Vis emb $L_1$ $H = 1$ | 0.45 | **0.55** | 0.69 | 0.41 | 0.24 | 0.47 |
| Vis emb $L_1$ $H = 2$ | **0.45** | 0.52 | 0.72 | **0.43** | **0.25** | **0.47** |
| Vis emb $L_1$ $H = 3$ | 0.42 | 0.52 | **0.72** | 0.42 | 0.22 | 0.46 |
| SR | 29.7 $\pm$3.8 | 24.9 $\pm$7.6 | **39.2 $\pm$4.1** | 28.7 $\pm$5.8 | 19.4 $\pm$5.8 | |

Table 14: Negative Spearman Correlation Coefficients across data points of the four eval setups between smoothed success rate and several validation metrics on the Push-T task. The rightmost mean column is the average of Spearman correlation of each model. In **bold** is highest value, underlined is 2nd highest. We display the mean success rate over the last 10 epochs in the last row.

| Model name | WM | $\text{WM}_W$ | WM-prop | WM-2-step | WM-L | Mean |
|---|---|---|---|---|---|---|
| Proprio dec $H = 1$ | 0.70 | 0.73 | 0.85 | 0.71 | 0.81 | 0.76 |
| Proprio dec $H = 2$ | 0.78 | 0.66 | 0.87 | 0.79 | 0.84 | 0.79 |
| Proprio dec $H = 3$ | 0.76 | 0.70 | 0.78 | 0.83 | 0.86 | 0.79 |
| Vis emb $L_2$ $H = 1$ | 0.80 | 0.73 | 0.88 | 0.82 | 0.87 | 0.82 |
| Vis emb $L_2$ $H = 2$ | 0.85 | 0.76 | 0.91 | **0.84** | 0.87 | 0.85 |
| Vis emb $L_2$ $H = 3$ | 0.86 | 0.70 | 0.89 | 0.80 | 0.87 | 0.82 |
| Vis emb $L_1$ $H = 1$ | 0.87 | **0.79** | 0.92 | 0.83 | **0.90** | **0.86** |
| Vis emb $L_1$ $H = 2$ | **0.88** | 0.78 | **0.93** | 0.84 | 0.88 | 0.86 |
| Vis emb $L_1$ $H = 3$ | 0.87 | 0.75 | 0.91 | 0.83 | 0.87 | 0.85 |

Since we are essentially in a supervised learning setting, training a regressor of future embeddings, it is clear that lower validation prediction losses (at all unrolling steps) means a more accurate world model. This is best observed in the visual decodings of validation rollouts throughout training.

Why the success rate does not correlate well to these losses is due to several factors. The validation prediction task is not fully aligned with the goal-conditioned planning task. The planning optimization task we use to evaluate models is a heuristic, where the objective is to minimize the embedding space distance of the last imagined state to the goal.

As we see in DROID experiments Figure 19, letting the planner sample actions that are OOD for the predictor can severely harm the plan accuracy and occult the improvement of the predictor throughout training. Another caveat is that a better world model that does not prevent the planning procedure from getting stuck in local cost minima, as we see with the score obtained with gradient-based planners in Table 11.

Table 15: Negative Spearman Correlation Coefficients across data points of the eight eval setups between smoothed success rate and several validation metrics on the Wall task. In **bold** is highest value, underlined is 2nd highest. We denote the visual embedding prediction errors Vis Emb and the proprioceptive decoding error Proprio dec, at horizons $H$ from one to three.

| Model name | WM | $WM_W$ | WM-prop | WM-2-step | WM-3-step | WM-L | Mean |
|---|---|---|---|---|---|---|---|
| Proprio dec $H = 1$ | 0.25 | 0.38 | 0.19 | 0.23 | 0.26 | 0.16 | 0.25 |
| Proprio dec $H = 2$ | 0.23 | 0.51 | 0.22 | 0.30 | 0.14 | 0.28 | 0.28 |
| Proprio dec $H = 3$ | 0.22 | 0.49 | 0.39 | 0.26 | 0.34 | 0.23 | 0.32 |
| Vis emb $L_2$ $H = 1$ | 0.72 | 0.94 | 0.74 | 0.73 | 0.69 | 0.75 | 0.76 |
| Vis emb $L_2$ $H = 2$ | 0.69 | 0.94 | 0.69 | 0.70 | 0.61 | 0.74 | 0.73 |
| Vis emb $L_2$ $H = 3$ | 0.68 | 0.94 | 0.65 | 0.69 | 0.56 | 0.70 | 0.70 |
| Vis emb $L_1$ $H = 1$ | **0.78** | **0.96** | **0.81** | **0.77** | **0.73** | **0.80** | **0.81** |
| Vis emb $L_1$ $H = 2$ | 0.74 | 0.96 | 0.76 | 0.72 | 0.69 | 0.79 | 0.78 |
| Vis emb $L_1$ $H = 3$ | 0.73 | 0.96 | 0.73 | 0.72 | 0.64 | 0.75 | 0.75 |

Table 16: Negative Spearman Correlation Coefficients across data points of the eight eval setups between smoothed success rate and several validation metrics on the Point Maze environment. The rightmost mean column is the average of Spearman correlation of each model. In **bold** is highest value, underlined is 2nd highest.

| Model name | WM | $WM_W$ | WM-prop | WM-2-step | WM-3-step | WM-L | Mean |
|---|---|---|---|---|---|---|---|
| Proprio dec $H = 1$ | 0.15 | 0.19 | 0.15 | 0.14 | 0.25 | 0.09 | 0.16 |
| Proprio dec $H = 2$ | 0.13 | 0.21 | 0.10 | 0.35 | 0.23 | 0.21 | 0.21 |
| Proprio dec $H = 3$ | 0.10 | 0.34 | 0.25 | 0.10 | 0.27 | 0.10 | 0.19 |
| Vis emb $L_2$ $H = 1$ | 0.43 | 0.50 | 0.82 | 0.53 | 0.42 | 0.19 | 0.48 |
| Vis emb $L_2$ $H = 2$ | 0.23 | 0.24 | 0.74 | 0.50 | **0.53** | 0.03 | 0.38 |
| Vis emb $L_2$ $H = 3$ | 0.21 | 0.30 | 0.67 | 0.46 | 0.49 | 0.17 | 0.38 |
| Vis emb $L_1$ $H = 1$ | **0.58** | **0.68** | **0.86** | 0.54 | 0.38 | **0.36** | **0.57** |
| Vis emb $L_1$ $H = 2$ | 0.37 | 0.49 | 0.78 | **0.55** | 0.50 | 0.19 | 0.48 |
| Vis emb $L_1$ $H = 3$ | 0.29 | 0.38 | 0.74 | 0.51 | 0.50 | 0.29 | 0.45 |

### G.4 Success over epochs

We display in Figure 20 and Figure 21 the evolution of success rate over training epochs for some of the models that compose the design choice study of this paper.

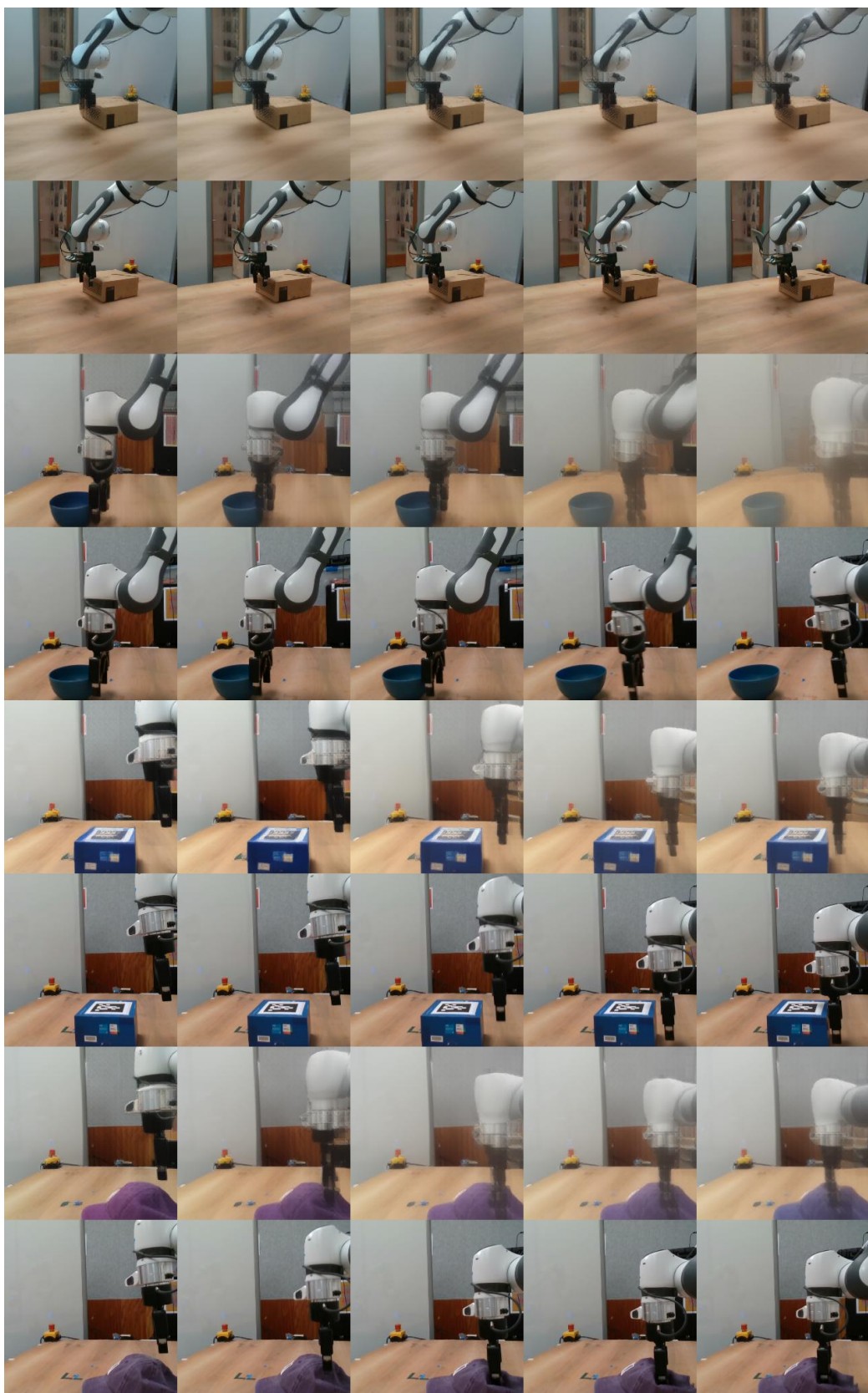

Figure 14: Unrolling a trajectory with our proposed JEPA-WM on some of our collected Franka arm trajectories, with our best model presented in Section 5.3. For each pair of rows, the top one is the model's visual decoding of the action unrolling, the bottom one is the ground-truth trajectory of the dataset. The model sometimes does not grasp well interaction with objects, which is the most frequent failure case.

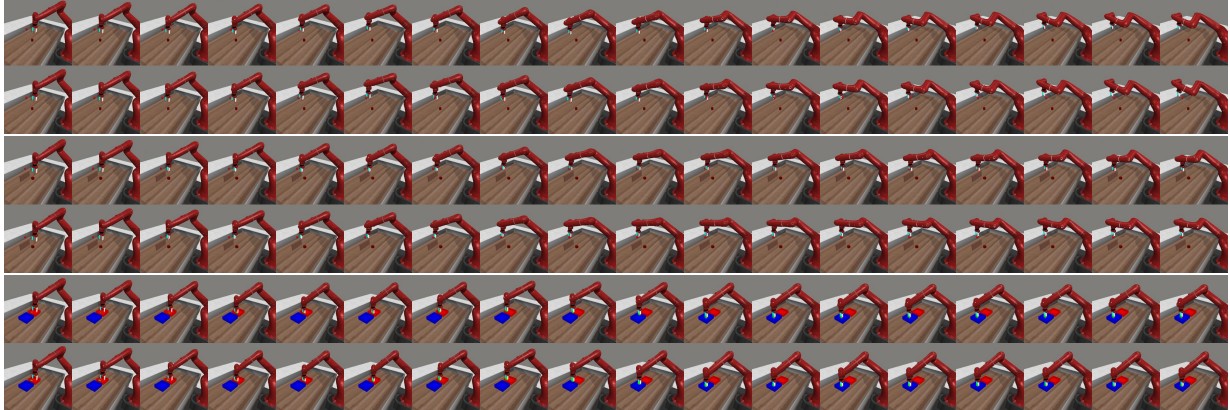

Figure 15: Samples of DINO-WM open-loop rollouts on the validation split, each on 18 model actions corresponding to 90 elementary actions in simulator. For each of the three pairs of rows, the groundtruth action stepping in simulator is below the decoding of the predictor rollout.

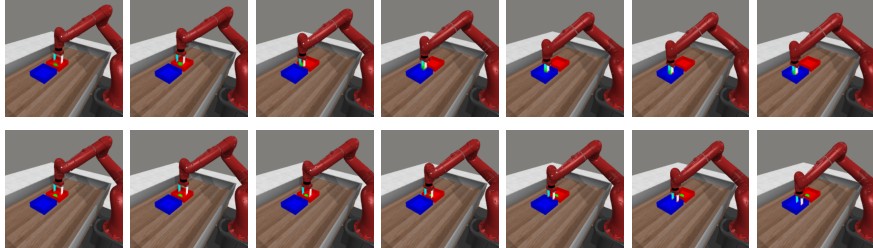

Figure 16: Upper row: Visual decoding of the unrolling of the action plan outputted by the NG planner, at step 1 of the Metaworld episode, on the bin-picking task. The world model's predictions resulting from the plan indicate that the object is picked. Lower row: Stepping of half of the plan in the simulator. The object is not picked and the plan leads the robotic arm to the target location without the object.

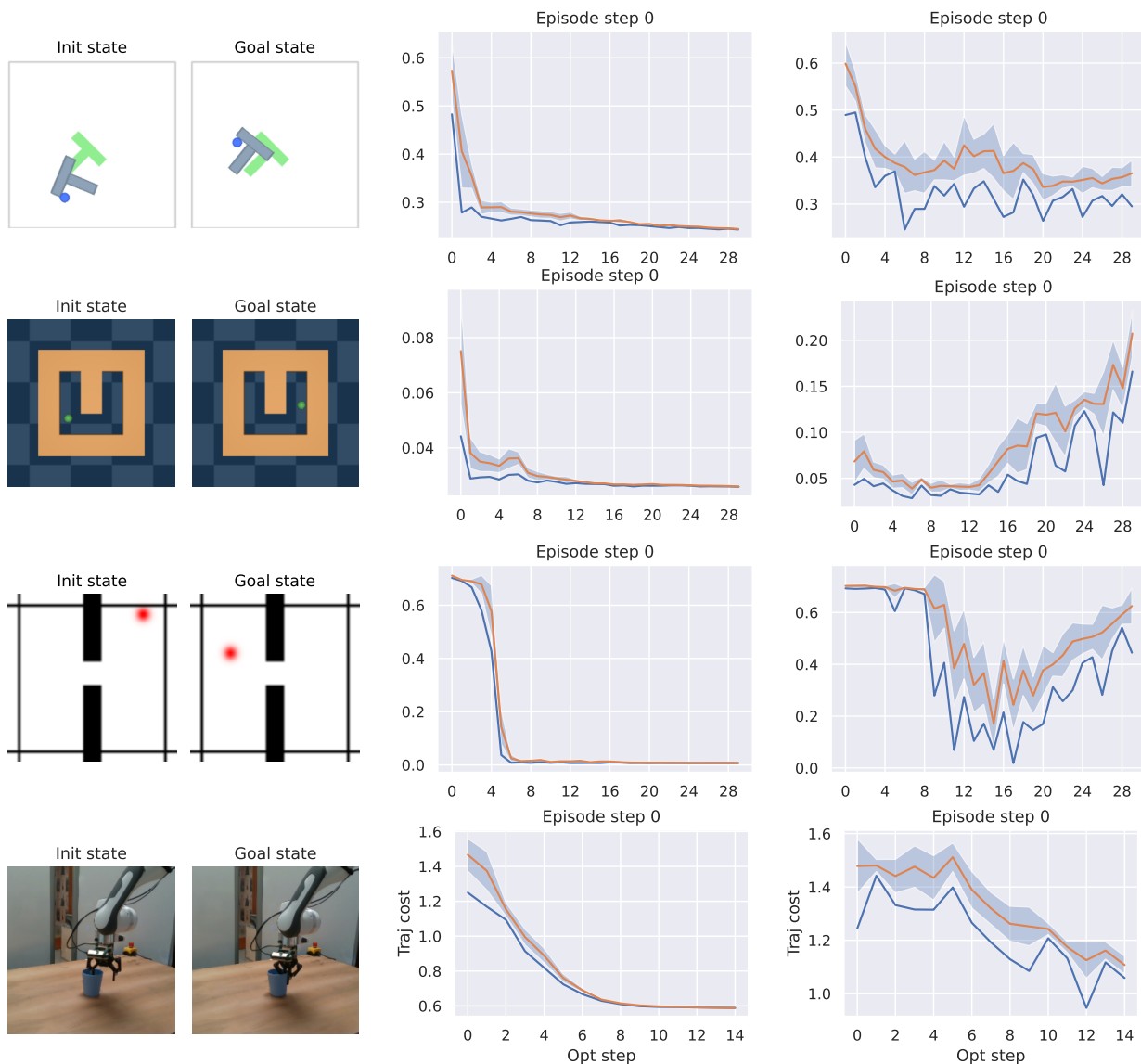

Figure 17: Convergence of planning optimization for all 2D environments and DROID. The model evaluated is the base WM model, at the end of training. Left: episode initial and goal state. Center: CEM planner. Right: NG planner. We display the planning cost of the best trajectory throughout the optimization steps.

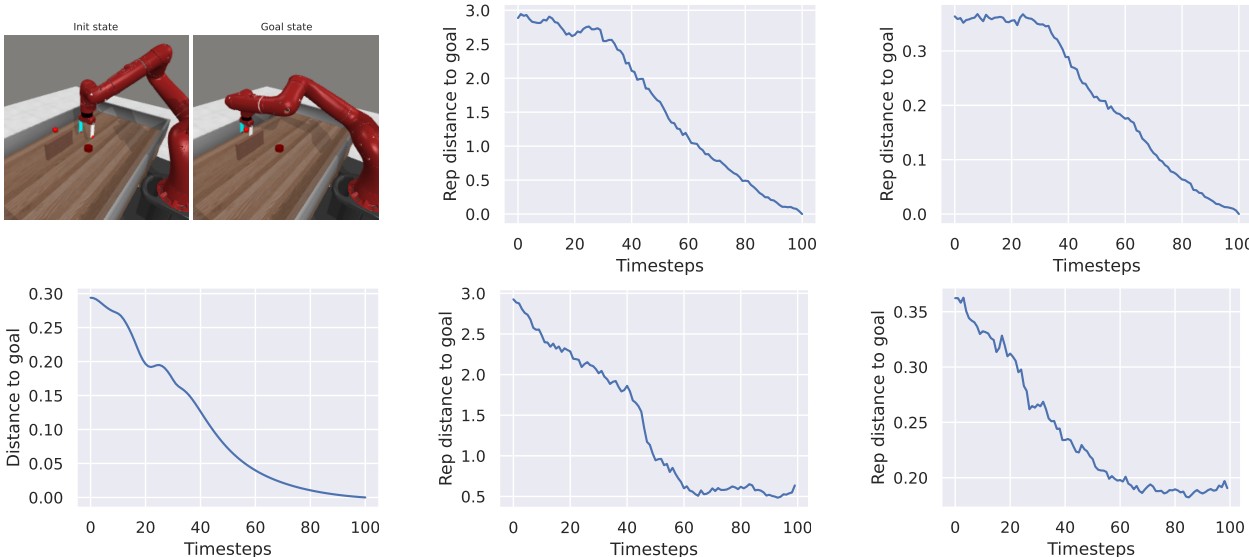

Figure 18: Same Metaworld reach-wall task setup: trajectory of the Base model (bottom center), the Large model (bottom right) and the expert policy (all other subfigures). Left: expert's executed episode first and last state at top, expert's distance of arm to goal position in the simulator space at bottom. Center: WM ViT-S encoder embedding space $L_2$ distance to goal, expert trajectory on top, WM planned episode at bottom. Right: WM-L encoder embedding space, expert trajectory on top, WM planned episode at bottom.

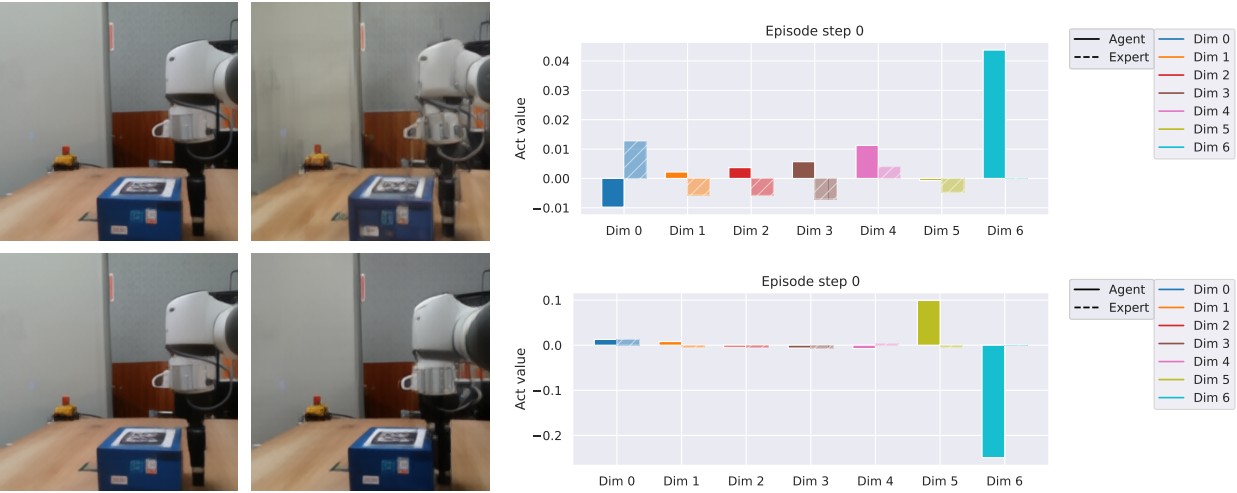

Figure 19: Top row: model at the end of the first training epoch. Bottom row: model at the end of training. Left: visual decoding of the horizon 1 plan. Right: comparison of the actions outputted by the planner (blue) and the groundtruth actions (orange) for the 7 action dimensions. The first three dimensions correspond to end-effector position control, the three next to end-effector orientation control, the last one to the gripper closure. The action error mostly comes from the gripper and orientation control dimensions. Hence, although only the model at the end of training correctly plans to approach the gripper from the box, its total action error is higher than at the beginning of training, if we consider all 7 dimensions.

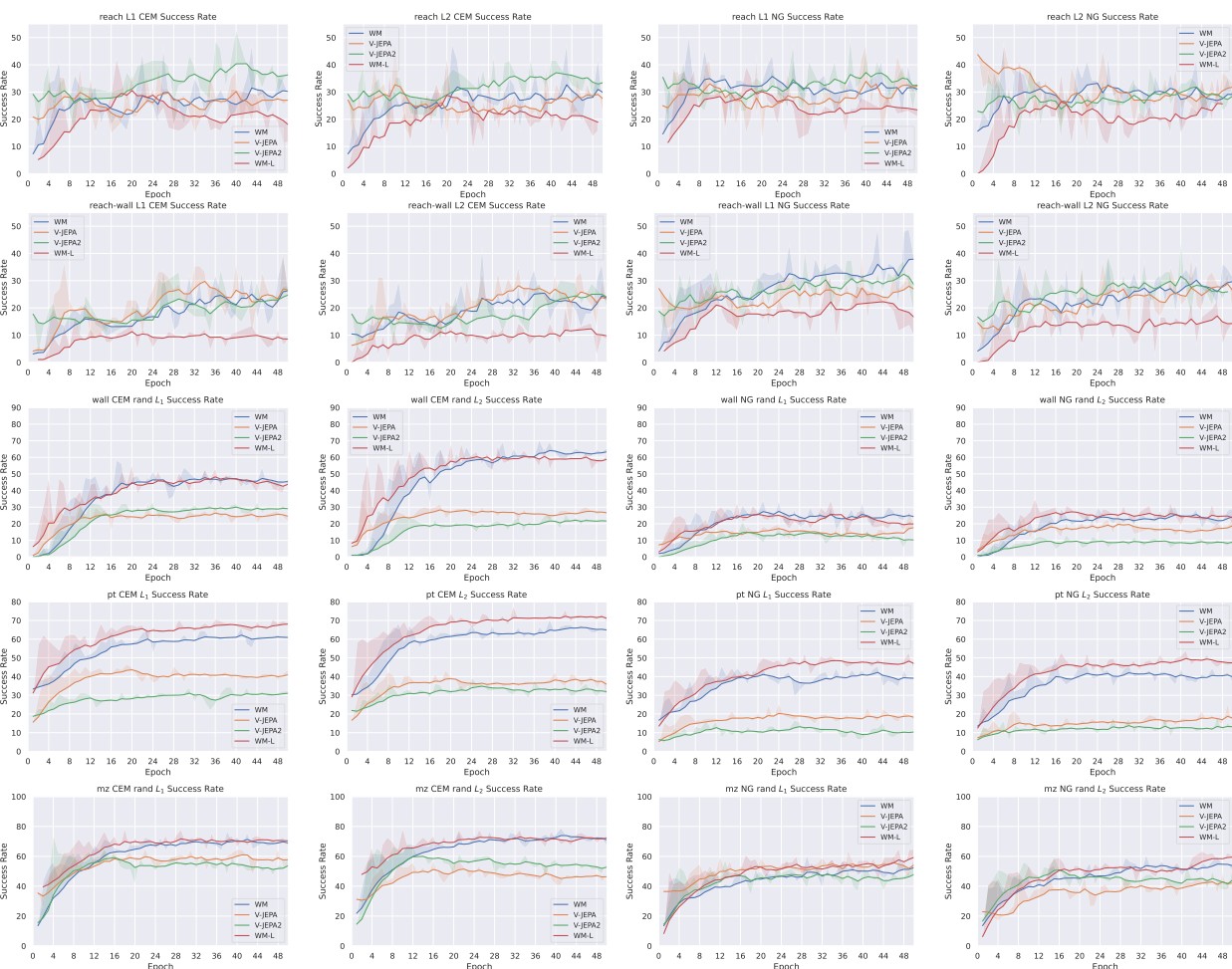

Figure 20: Success rate evolution for several evaluation setups on all tasks, comparing image and video encoders. At each epoch, we evaluate the success rate on 96 independent episodes and report the average. We denote WM the base model for the design choice study, namely DINO-WM (Zhou et al., 2024a) without proprioception, and WM-L its Vit-L version. We display the results for the models learned on top of V-JEPA and V-JEPA2.

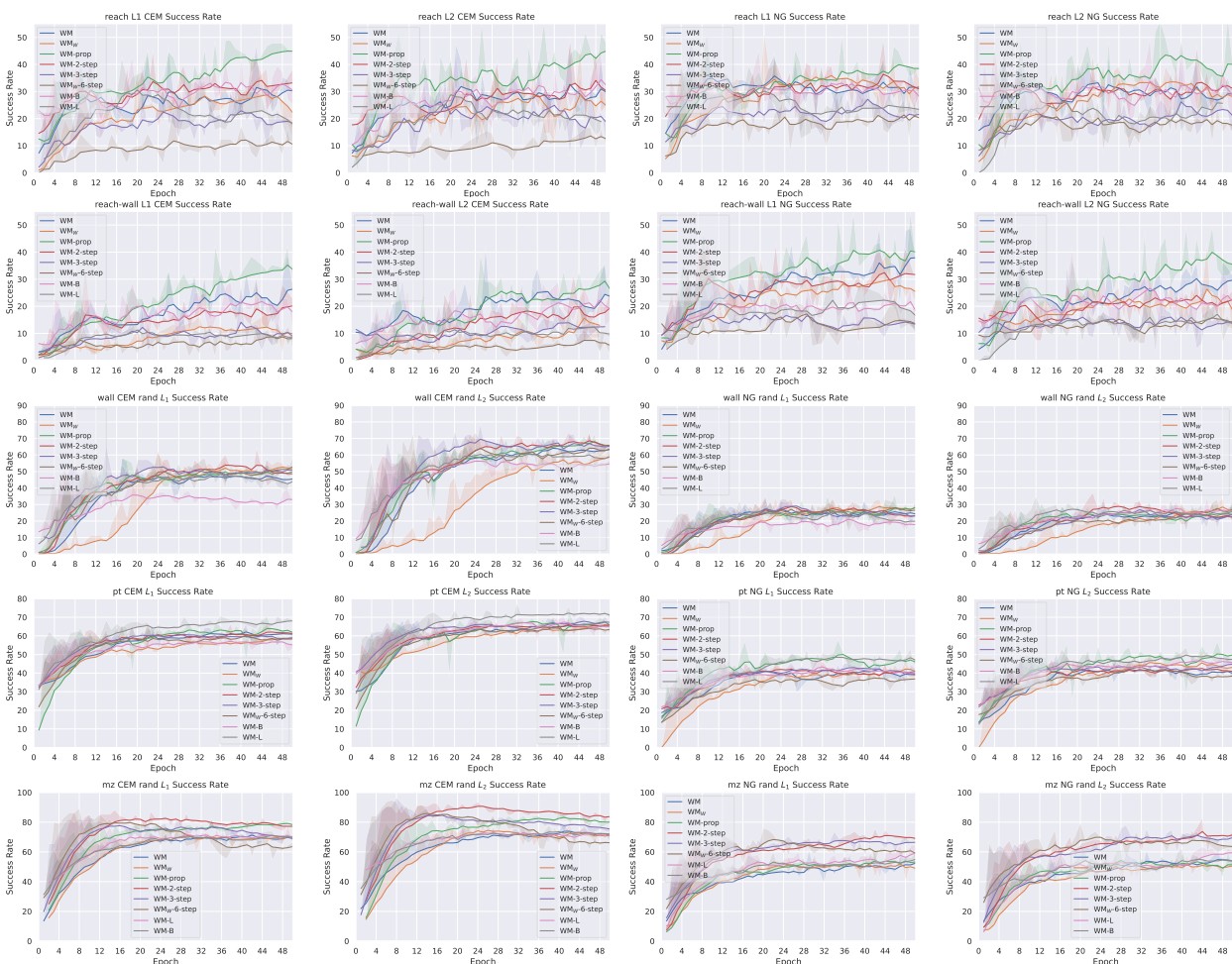

Figure 21: Success rate evolution for several evaluation setups on all tasks, comparing multistep rollout, proprioception and model size. We denote WM-B, WM-L the variants of the base model with size ViT-B and ViT-L, WM-prop the variant with proprioception, and the multistep rollout models as WM-$k$-step. Row 1: Metaworld reach, row 2: Metaworld reach-wall, row 3: Wall, row 4: Push-T, row 5: Point Maze. At each epoch, we evaluate the success rate on 96 independent episodes and report the average.

