# OpenReview forum: "What Drives Success in Physical Planning with Joint-Embedding Predictive World Models?"
_TMLR — Accepted by TMLR_

### Review · Reviewer_C7V4 · 2026-02-17

**Summary Of Contributions:**

The paper presents an empirical study of various factors and decisions that affect planning with latent space prediction world models which they term as jepa-wms. The factors they study include - various planning algorithms (gradient based and search based), predictor architecture, encoder type (DINO, JEPA,...), impact of proprioception, impact of multi-step rollout training, etc. Each factor is evaluated in isolation. After finding the best option per factor, the construct an optimal setup with the chosen best design decisions and show that the resulting models outperforms DINO-WM and VJEPA-2-AC - two models in the same class of approaches (planning in latent space).

Strengths:

1. The paper presents a unified framework for latent prediction based planning approaches by defining common components such as encoders and predictors.
2. The presented study is comprehensive considering a good range of environments (synthetic and real world) and design decisions including exhaustive number of implementations per design decision.
3. Even though each design decision is studied in isolation, combining the best choices leads to a superior final model beating the baselines.


Weaknesses:

1. While the study is comprehensive, I don't think most of the results for any design are surprising. For instance, proprioception information helping is expected since more information can only aid in more accurate future predictions and performance. Similarly, multi-step rollouts helping is also expected. Therefore, the I am not convinced about the full value of this study.
2. I am not sure why the authors chose to name these approaches as "JEPA"-WM approaches. Predicting future states in latent space is common technique that has existed prior to JEPA. For example, muzero (Schrittwieser et al., 2019) and PlaNet (Hafner et al 2018) are some examples of works that were already doing latent prediction for future states. I would day the main difference with the class of approaches presented in the paper is that they start with pretrained visual backbones such as DINOv2/v3 and JEPA. Moreover, another reason I don't see how the terminology "JEPA"-WM is apt for this as "JEPA" is a particular ssl technique which mainly trains a model by predicting an image/video from a masked view of that image/video in the latent space.
3. I believe training with rollouts of 3-6 frames can be very restrictive when moving to more complex environments. The authors claim that 3 frames are sufficient to infer accelaration of an object. While this is true, it assumes environments are very simple. For example, effects such as occlusion may require the model to track context of video beyond 3-6 frames.
4. Another critique of these kinds of approaches is that there is no means to account for the stochasticity in the environment. Most environments pose many factors of uncertanity. The class of approaches studied in this paper only perform deterministic predictions of future states. This can be limiting when moving to more real-world environments.

**Audience:**

Yes

**Audience Explanation:**

Yes, I believe this paper would be of interest to some segments of the TMLR audience:

Model-Based Reinforcement Learning (MBRL) Researchers: There is some work in the community already about building world models which predict the future in the latent space as I have highlighted above. This work may be of interest to those communities.

Self-Supervised Learning (SSL) Researchers: This audience would be interested in seeing how frozen representations from large-scale pre-training (like DINOv2 and V-JEPA) can be 'recycled' for complex downstream planning tasks.

Robotics researchers: The paper considers datasets such as DROID and ROBOCASSA and also performs experiments on real world franka arm. Therefore, I think the experiments and findings will be of interest to the robotics community.

**Broader Impact Concerns:**

No concerns

**Claims And Evidence:**

Yes

**Claims Explanation:**

The paper positions itself as an analysis paper to study various design decisions important when planning with latent prediction world models. To this end, it unifies various works under a common framework and comprehensively studies various design decisions and finds the best setup for each. Since the paper positions itself as an analysis paper and backs it up with supporting experiments, I believe the main claim is supported by clear evidence. I do have some issues with the framing and the broader utility of the approach which I have detailed in the above text box.

**Requested Changes:**

Clarify and Justify "JEPA" name: The authors should provide a more rigorous justification for the use of the "JEPA" label. Given that the encoders are frozen and were not necessarily trained with a JEPA objective (e.g., DINOv2), the authors must address how this work specifically aligns with JEPA principles as opposed to standard Latent World Models or State-Space Models.

Acknowledge Deterministic/Stochastic Trade-offs: The authors must include a section in the Discussion or Limitations addressing the deterministic nature of their predictor. Since they use MSE loss in latent space, they should clarify how the model handles (or fails to handle) multi-modal future distributions or environmental stochasticity compared to generative models.

Would strengthen the work:
Longer context training: It would be beneficial to include training runs with temporal contexts longer than 6 steps (e.g., up to 10 or 15 steps) and study how it affects planning performance, particularly in environments where object permanence or long-term momentum is critical.

---

> ### Author Response · Authors · 2026-03-10
> **Rebuttal response 1/n**
>
> Dear Reviewer,
> We sincerely thank you for your thorough review and constructive feedback. Below, we address each point and describe the corresponding changes in the revised manuscript.
>
> ## Change 1: Clarifying the JEPA-WM Terminology
> We fully agree that latent-space prediction predates the JEPA framework: MuZero (Schrittwieser et al., 2020), PlaNet (Hafner et al., 2019), and the Dreamer line (Hafner et al., 2020; 2023) all learn dynamics models in learned latent spaces. The term "JEPA-WM" is not intended to claim novelty over latent prediction in general; rather, it designates a specific class of models characterized by the nature of their learning signal.
>
> To clarify the scope, we note that DINOv2 is widely regarded as a Joint-Embedding Architecture (JEA), as its loss is computed in embedding space via masked-patch prediction. I-JEPA and V-JEPA similarly rely on masked image/video modeling in embedding space, with the additional use of a predictor that makes them JEPAs in the strict sense. Importantly, **JEPA-WMs are not defined by reliance on pretrained visual representations**: the terminology equally covers models where the encoder is learned jointly with the dynamics model, such as PLDM (Sobal et al. 2025) or the action-conditioned video JEPA in EB-JEPA (Terver et al. 2026). As stated in the introduction: "In this paper, we focus on the learning of the dynamics (predictor) rather than of the representation (encoder), as in DINO-WM and V-JEPA-2-AC."
>
> What distinguishes a JEPA-WM is that **the dynamics model is trained solely through a predictive loss in embedding space**. This is the key differentiator from the methods the reviewer mentions:
> - **MuZero** learns its internal state and dynamics via reward, value, and policy prediction losses. Its hidden state is optimized to predict task-relevant quantities (policies, values, rewards), not to capture environmental semantics per se.
> - **PlaNet** primarily relies on a pixel-reconstruction loss (alongside a latent prior) to train its transition models, supplemented by a reward prediction loss.
>
> By contrast, the JEPA-WMs studied in this paper rely exclusively on a next-state embedding prediction loss: no reconstruction, no reward prediction, no value or policy heads.
>
> We have added clarifying sentences in the introduction and related work of the revised manuscript, explicitly acknowledging prior latent-prediction methods and stating the precise criteria that define a JEPA-WM.

---

> > ### Author Response · Authors · 2026-03-10
> > **Rebuttal response 2/n**
> >
> > ## Change 2: Deterministic/Stochastic Trade-offs
> > We appreciate the reviewer raising this important aspect. We have added a dedicated discussion of the deterministic/stochastic trade-off in the introduction and related work sections. We summarize the key arguments below.
> >
> > **Sufficient modeling for targeted benchmarks.** The benchmarks considered in this work (Meta-World, Push-T, PointMaze, DROID, Robocasa) are fundamentally deterministic: an action applied to a given state produces a uniquely determined next state. A world model trained to predict the conditional mean (as optimized by MSE) should therefore be capable of recovering the true dynamics.
> >
> > **Latent abstraction mitigates multi-modality.** The concern that MSE fails to handle multi-modal distributions primarily applies to high-entropy observation spaces such as pixels. By predicting in the latent space of vision foundation models (DINO, V-JEPA) or shallow encoders learned jointly with the predictor (as in PLDM or the action-conditioned video JEPA in EB-JEPA), JEPA-WMs abstract away task-irrelevant stochastic factors. In this semantically rich space, transitions that would be multi-modal at the pixel level often collapse into a unimodal signal, mitigating the traditional drawbacks of MSE-based regression.
> >
> > **Diffusion-based alternatives and their trade-offs.** Large-scale video generation models often referred to as world models (e.g., Cosmos, Genie, Sora) adopt diffusion-based backbones to handle the multi-modal distributions that characterize natural video. However, this expressivity comes at the cost of substantial computational overhead and potential hallucination of task-irrelevant details.
> >
> > **Extension to stochastic environments.** In settings with genuine aleatoric uncertainty, a JEPA-WM would require a stochastic component. This could be achieved via optimization over a latent variable conditioning the predictor, as discussed in "A Path Towards Autonomous Machine Intelligence" (LeCun, 2022), or via diffusion models operating in latent space.
> >
> > **Closed-loop robustness.** Even when true transition multi-modality exists (e.g., an object could slide left or right on a slippery surface), the closed-loop nature of MPC provides a natural correction mechanism: predictions are re-planned at every time step, allowing the controller to compensate for deviations between the predicted and realized state.

---

> > > ### Author Response · Authors · 2026-03-10
> > > **Rebuttal response 3/n**
> > >
> > > ## Change 3: Longer Context Training
> > > We note that, while the reviewer mentions 6 frames of context, our models were trained on trajectory slices of length up to $W+1=8$, as reported in Figure 5b. The parameter $W$ controls the maximum number of frames the predictor sees as context during training. We agree that certain phenomena (object permanence, long-term momentum) may in principle require longer context. However, on DROID and Robocasa, occlusions are rare, as they can only occur behind the manipulated objects. Furthermore, we sample the DROID dataset (natively at 30 fps) at 4 fps, so a slice of $W+1=8$ frames already spans over 2 seconds of video, covering most occlusion events in the dataset.
> > >
> > > For completeness, we have added new experiments in the "Maximum context size" paragraph of subsection 5.2 and Figure 5b, training models with $W$ set to 10 and 15 on all datasets. We also restructured this paragraph for clarity.
> > >
> > > An important consideration is how training trajectories are sliced into non-overlapping segments of length $W+1$. Details are provided in Appendix D. With a fixed computational budget, increasing $W$ yields fewer but longer slices. This leads to a performance decrease beyond the optimal $W$ for datasets with less data or shorter trajectories:
> > > - On DROID, setting $W$ too high causes some videos shorter than $W+1$ to be discarded entirely, amplifying the data reduction: at $W=3$, we retain 99.2% of the dataset; at $W=9$, 96.4%; and at $W=14$, only 86%.
> > > - For the Wall dataset, whose trajectories have length $L=50$, we can only increase $W$ to 9, since slices of length $(W+1)f = 10 \times 5 = 50$ already span the full trajectory, where $f$ is the frameskip.
> > > - For Push-T (longer trajectories) and Maze (more data relative to task complexity), performance remains essentially constant beyond $W=3$. This observation aligns with the finding of PLDM (Sobal et al. 2025) that world models are effective at stitching suboptimal trajectories, compared to GCRL.

---

### Review · Reviewer_uc1S · 2026-02-27

**Summary Of Contributions:**

## Summary

This paper presents a thorough ablation study on the design choices of latent world models.  Specifically, this paper ablates the effectiveness of searching-based and optimization-based planning algortihms, the impact of rollout steps for planning, the usage proprioceptive information, and the effect of context window size and varying encoder/predictor architecture.  Lastly, this paper studies if the scaling law w.r.t model size holds in latent world models.

---

## Strengths

1. This paper presents a systematic study on latent world models, helping the community identify the best design choices for building performant world models
2. The experiments are elegantly designed: by varying one factor at a time, the results clearly isolate the contribution of each component.
3. The paper is well-organized and easy to follow.

---

## Weaknesses

1. Although this paper delivers a great ablation study, these analyses only bring insight in the choice of model architectures and hyper-parameters, rather than answering fundamental questions, such as "what are the representations for encoding environmental dynamics" or "how to learn such representations", for latent world models. In short, this paper offers engineering, but not algorithmic, insights on learning latent world models.
2. Some statements oversell the idea of latent world models without proper / up-to-date reference.  For example, in the introduction, the authors claim "PLDM offer stronger generalization than other Goal-Conditioned Reinforcement Learning (GCRL) methods, especially on suboptimal training trajectories".  It seems that this statement is not properly justified.  First, the experiments conducted in PLDM only consider navigation tasks, while GCRL methods have been demonstrated on a wide range of robotic tasks, including locomotion and manipulation [1].  Secondly, GCRL approaches have gained great progress recently (such as HILP [2] and quasimetric GCRL [3]). There is no comparison against modern GCRL and latent world models.  As a result, the original statement is not properly justified.
3. This paper does not discuss and compare against "implicit" world models (such as TD-Jepa [4]). If "explicit" world model is a better choice is not properly justified.
4. The results on model scaling is disappointing (Fig 6).   Using larger models only brings trivial performance gain.
5. This paper lacks the analysis on data scaling.  It's unclear if these design choices impact the sample efficiency of latent world models.


Reference:

[1] Park, Seohong, et al. "Ogbench: Benchmarking offline goal-conditioned rl." arXiv preprint arXiv:2410.20092 (2024).

[2] Park, Seohong, Tobias Kreiman, and Sergey Levine. "Foundation policies with hilbert representations." arXiv preprint arXiv:2402.15567 (2024).

[3] Myers, Vivek, et al. "Offline goal-conditioned reinforcement learning with quasimetric representations." arXiv preprint arXiv:2509.20478 (2025).

[4] Bagatella, Marco, et al. "Td-jepa: Latent-predictive representations for zero-shot reinforcement learning." arXiv preprint arXiv:2510.00739 (2025).

**Audience:**

Yes

**Audience Explanation:**

As illustrated earlier, this paper offers engineering, but not algorithmic, insights on learning latent world models.

**Broader Impact Concerns:**

The discussion of broader impact concerns is complete

**Claims And Evidence:**

Yes

**Claims Explanation:**

The experiments are conducted by varying one factor at a time, the results clearly isolate the contribution of each component.

**Requested Changes:**

1. Revise the statement of "PLDM offer stronger generalization than other Goal-Conditioned Reinforcement Learning (GCRL) methods, especially on suboptimal training trajectories" in the introduction.  Add the corresponding discussion in the related work.
2. Discuss and compare againt implicit latent world model--TD-Jepa.
3. Discsuss the results of Figure 6.
4. Include analysis on data scaling.

---

> ### Author Response · Authors · 2026-03-09
> **Rebuttal response 1/n**
>
> ## Change 1: JEPA-WMs relative to GCRL
>
> We thank the reviewer for pointing out that our original claims about JEPA-WMs versus GCRL methods were overly broad. We agree that PLDM's comparison was limited to specific 2D navigation tasks and a particular set of GCRL baselines (those of OGBench), and that presenting these results as a general statement was misleading.
>
> In the revised manuscript, we have scoped the claim in the introduction (Section 1) to accurately reflect the experimental setting of PLDM: "PLDM shows that, on 2D navigation tasks, world models learned in a latent space, trained as JEPAs, generalize better than the GCRL baselines considered in that work~\citep{OGBench}, especially on suboptimal training trajectories."
>
> Additionally, we have expanded the GCRL paragraph in Section 2 (Related work) to acknowledge recent advances in GCRL. Specifically, we now discuss HILP, quasimetric GCRL, and OGBench. This provides the reader with a more complete picture of the GCRL landscape and makes clear that our work does not claim superiority over these more recent methods, a fortiori since the scope of our paper is not to compare to such concurrent approaches.

---

> > ### Author Response · Authors · 2026-03-09
> > **Rebuttal response 2/n**
> >
> > ## Change 2: Implicit World Models like TD-JEPA
> >
> > We thank the reviewer for raising this important point and directing us to TD-JEPA. This is a significant recent methodological contribution that prompts us to carefully clarify our terminology and the scope of what we define as a JEPA-WM. We completely agree with the reviewer's proposed distinction between "implicit" and "explicit" world models.
> >
> > We can formalize this dichotomy as follows:
> >
> > * **Explicit World Models (Our Focus):** These are autoregressive world models conditioned on raw actions. The predictor generates an embedding of a valid future physical state conditioned on a context of past states and actions. Because these models are completely agnostic to the downstream task during training, they require explicit test-time planning (e.g., via CEM, MPPI, or gradient descent) to search the action space for a trajectory that minimizes a task-specific cost function.
> > * **Implicit World Models (e.g., TD-JEPA):** Rather than generating sequential future states, implicit models like TD-JEPA learn representations predictive of long-term latent dynamics. TD-JEPA trains explicit state and task encoders, a policy-conditioned multi-step predictor, and a set of parameterized policies directly in latent space. This approach enables zero-shot optimization of any reward function at test time without the need for an explicit planning search. In TD-JEPA, if we set the $\gamma$ discount factor to zero in all loss equations of section 3, we recover an explicit world model loss, trained via next-state latent prediction. This is a degenerate case of the successor measure.
> >
> > A key practical difference between these two paradigms lies in their **computational tradeoffs across training and inference**.
> > When matching architectures for simple 2D tasks from pixels, explicit JEPA-WMs are highly efficient to train. For example, explicit implementations like PLDM or EB-JEPA (using shallow IMPALA or custom ConvNet encoders paired with GRU/Conv predictors) converge in fewer than 10k gradient steps, taking roughly 2 hours on a single GPU. In contrast, TD-JEPA, which uses a comparable DrQ-v2 convolutional encoder and MLP predictor/policies, requires 1 to 2 million gradient steps to adequately learn its policy-conditioned successor features.
> > Thus, implicit models front-load the computational burden onto the training phase to enable near-instantaneous zero-shot retrieval at test time. Conversely, explicit action-conditioned world models train very efficiently on system dynamics but defer the computational burden to test time, where trajectory optimization (e.g., MPPI/CEM) is required.
> >
> > **Explicit world models allow for a broader range of use cases at test time**. With an explicit world model, one can generate counterfactual trajectories in state embedding space by hardcoding action trajectories corresponding to different future scenarios, as we showcase in Figure 2 of our paper. Implicit world models like TD-JEPA are non-generative (in embedding space), hence cannot perform such counterfactual generation.
> > Additionally, TD-JEPA relies on zero-shot RL, which allows the deployment of policies that are optimal for rewards that belong to the span of the learned feature space. With our explicit world models, since the cost functions are independent of the learned world model, we have more freedom in defining cost (reward) functions at test time, which can be any function from the feature space to $\mathbb{R}_+$.
> >
> > Finally, between explicit and implicit world models, the design space is vast and **many methods exist that combine aspects from both**. For example, TD-MPC2 does learn an explicit world model, but not only from latent next state prediction, as it also receives a reward prediction learning signal. This method also combines planning with the use of a policy, which provides candidate actions to the MPPI planning optimizer.
> >
> > Given the vast design space of explicit world models and the specific scaling behaviors we investigate for complex robotic data, our current empirical study is scoped entirely to the explicit world models. We leave a full empirical comparison between explicit and implicit world models for future work. However, we agree that discussing implicit models adds highly valuable context. We have added a dedicated discussion comparing explicit and implicit world models, including TD-JEPA, to the "Related work" section of the revised manuscript.

---

> > > ### Author Response · Authors · 2026-03-09
> > > **Rebuttal response 3/n**
> > >
> > > ## Change 3: Model scaling yields limited gains in simulated environments
> > >
> > > We agree that the scaling result deserves a more thorough discussion. In the revised manuscript, we have substantially expanded the "Model scaling" paragraph in Section 5 to address this point.
> > >
> > > We now explicitly acknowledge that scaling yields limited gains in simulated environments and provide three complementary hypotheses for this behavior:
> > > 1. Simulated tasks are simple enough to saturate at small model sizes, so additional capacity brings no benefit.
> > > 2. Larger embedding spaces make the planning optimization landscape harder to navigate, as the planner must distinguish nearby states in a higher-dimensional space.
> > > 3. With fixed training compute budgets, larger models see fewer gradient updates per parameter, potentially leading to underfitting.
> > >
> > > Importantly, we emphasize the positive counterpart: on DROID, which involves real-world data with complex visual dynamics, we observe a clear and consistent benefit from scaling both encoder size and predictor depth. This contrast is itself a practical finding for practitioners: scaling model capacity is most beneficial when the environment exhibits complex, high-dimensional dynamics (as in real-world robotics), while simulated environments with simple dynamics saturate at small model sizes.

---

> > > > ### Author Response · Authors · 2026-03-09
> > > > **Rebuttal response 4/n**
> > > >
> > > > ## Change 4: Data scaling
> > > >
> > > > We remind the reviewer that data scaling experiments on Push-T are already performed in the DINO-WM paper. Data quality is also the main focus of the PLDM paper, which compares to GCRL baselines, varying the data quality and quantity.
> > > > Yet, we agree that, to make our study more complete, a data scaling study is beneficial. Hence, we add a figure showing how the success rate evolves when training on increasing fractions of the data available, namely 2%, 10%, 50% (and 100%).
> > > >
> > > > These experiments represent a significant amount of FLOPs, as we launch 3 training runs (for 2%, 10%, 50% fractions) per dataset, with 5 datasets, resulting in 15 training runs. To analyse the relation between data scaling and the design choices considered in the study (architecture, training loss, and maximum context size), we would need to launch 15 training runs for each considered design choice’s set of experiments, leading to an unreasonable number of training runs. Hence, we chose to launch the 15 data scaling training runs for our best JEPA-WM, the DINO-WM baseline, and the V-JEPA-2-AC baseline. We add these results in two figures in the main body of the paper (Figure 7), along with a new data scaling paragraph.
> > > > We observe :
> > > > - For all datasets and methods considered, performance clearly increases when scaling data, as the world model captures more diverse dynamics and nuances of the environment, allowing for more accurate predictions and better-informed planner optimization. This is expected, as the three methods rely on the same main learning signal, which is the one-step teacher-forcing loss in state embedding space.
> > > > - Our method outperforms baselines especially on DROID and Wall, where the data scaling seems less saturated. On the Push-T and Metaworld tasks, it also seems like increasing data diversity would increase performance by a large margin, yet our method's advantage is less clear.
> > > > - The results for models trained on DROID are in Figure 7. When evaluating these on offline planning on Franka videos (denoted DROID), we clearly see performance scaling with data quantity. Yet, for Robocasa Place and Reach, scaling is less clear due to the domain gap between DROID and Robocasa.

---

### Review · Reviewer_EuSp · 2026-03-02

**Summary Of Contributions:**

This paper provides a comprehensive and careful empirical evaluation of different technical components in JEPA-based world models, including prediction architectures, planner optimization modules, encoders, training context, and rollout steps. It systematically analyzes what design choices work best across different tasks and benchmarks, and in that sense offers a useful empirical recipe for the community. I believe this is a meaningful contribution to the world model literature.

In general, the main strength of the paper lies in this well-structured evaluation suite, which is practically valuable for researchers and practitioners. Most of the experimental designs also appear careful and largely faithful to the intended comparisons. That said, given the ambitious scope and strong claims of the paper, I feel the evaluation could be further strengthened with a few additional analyses. I list some specific suggestions in the boxes below for the authors’ consideration during rebuttal.

Strengths:

- The scope and motivation are well aligned with the needs of the community. Providing systematic empirical evidence on JEPA-based world models is likely to be useful for researchers who want practical guidance on model design and deployment.

- The open-sourcing effort is commendable. I checked the code and checkpoints, and they appear complete and well organized. The reproduction details are also clear, which significantly improves the credibility and usability of the work.

Weaknesses:

- W1: Regarding evaluation, as the authors note, success rate can be quite noisy. While the paper already reports world model prediction errors in both embedding space and pixel space, I believe additional evaluation metrics could further strengthen the conclusions. I will suggest some concrete options in the later boxes.

- W2: Since the paper mainly presents a large suite of empirical evaluations, it would be valuable to include more in-depth (possibly theoretical or analytical) discussions on key design choices. This could help move from empirical observations to more principled guidelines.

- W3: For the encoder, the current setup seems to implicitly assume that DINO captures all essential representations for downstream control. This assumption may not always hold. It would be interesting to explore whether task-aligned tuning or adaptation of DINO could further improve performance, especially given recent works that show benefits from aligning visual representations with the task space (see references in the later box).

**Audience:**

Yes

**Audience Explanation:**

Yes, I think this provides useful technical information for the community of world models and represntation learning.

**Broader Impact Concerns:**

The current broader impact section is comprehensive IMO.

**Claims And Evidence:**

Yes

**Claims Explanation:**

Yes, the paper presents a broad and well-structured set of evaluations for JEPA-based world models, with clearly defined experimental setups. Based on these systematic comparisons, the authors make concrete claims about which components work best under different tasks and settings. Overall, the evaluation is comprehensive, clearly presented, and largely sound, which makes the conclusions easy to follow and practically useful

**Requested Changes:**

For writing, I would suggest adding a concise summary table in the main paper that lists all candidate design choices for each component in the JEPA-based world model (e.g., encoder, predictor, planner, training context, rollout horizon). Then, a second table could map dataset/task types to the empirically best-performing component choices. This would make the “recipe” aspect of the paper much clearer and easier to use. In addition, a notation table in the appendix would improve readability.

For W1: I would encourage including more world-model-specific evaluations beyond success rate and embedding/pixel prediction errors. In particular, metrics that directly probe action-following ability, physical consistency, and controllability would better reflect the functional quality of the world model, you can refer to [1]. Additionally, I would be very interested in OOD trajectory evaluation. For instance, if the world model is trained mainly on successful demonstrations, how robust is it when rolling out on sub-optimal or partially failed trajectories? One practical setup is to collect diverse trajectories from different training checkpoints (thus inducing varying policy quality), and then evaluate prediction fidelity and planning performance on held-out trajectory distributions. This would better reflect real deployment settings where data is not always optimal.

For W2: More explicit discussion would be helpful.

For W3: I would expect either more empirical analysis or a deeper discussion. In particular, if the method heavily relies on a frozen representation backbone, it would be useful to discuss the limits of this assumption and whether task-aligned adaptation could further improve performance. See a recent work [2] on fine-tuning dino to align with task-specific information.


[1] Shang, Yu, et al. "WorldArena: A Unified Benchmark for Evaluating Perception and Functional Utility of Embodied World Models." arXiv preprint arXiv:2602.08971 (2026).

[2] Lyu, Jiangran, et al. "LDA-1B: Scaling Latent Dynamics Action Model via Universal Embodied Data Ingestion." arXiv preprint arXiv:2602.12215 (2026).

---

> ### Author Response · Authors · 2026-03-11
> **Rebuttal response 1/n**
>
> ## Change 1: Writing
> We implemented all of your suggestions on this point, adding a table summarising the design choices studied, with the optimal choices per dataset / environment (Table 1) as well as a notation table in the appendix (Section A).

---

> ### Author Response · Authors · 2026-03-11
> **Rebuttal response 2/n**
>
> ## Change 2: World-model-specific evaluations
> Thank you for these constructive suggestions. We address both points below.
>
> ### Additional evaluation metrics beyond success rate
>
> Our paper already reports several metrics that probe the functional quality of the world model independently of planning success: **(i)** the counterfactual experiment (Figure 2) directly tests action-following by checking whether the model produces distinct predicted futures for distinct action sequences from the same initial state, which is the latent-space analogue of the "Action Following" metric of WorldArena [1]; the Action Score on DROID (Section 4.1) further quantifies this via $L_1$ error between planned and ground-truth end-effector actions; **(ii)** embedding-space prediction error at horizons $H=1,2,3$ (Appendix G.2) tracks compounding errors across autoregressive unrolling, and the Spearman correlation study (Appendix G.3) systematically relates these to success rate across all models; **(iii)** proprioceptive decoding error throughout unrolling assesses state-level physical consistency, and LPIPS between decoded rollouts and ground-truth frames captures visual prediction quality (Appendix G.2).
>
> **On WorldArena [1].** WorldArena is, to the best of our knowledge, a comprehensive benchmark for embodied world models to date, and a relevant reference for this discussion. That said, all 14 models it evaluates are pixel-level generative video models (both text- and action-conditioned); it acknowledges but does not evaluate latent-space world models. Its 16 metrics (VLM-judged interaction quality, optical-flow-based motion smoothness, depth accuracy, etc.) all require generated video frames as input. Decoding our latent predictions to pixels and applying these metrics would evaluate the joint system (world model + visual decoder), conflating decoder quality with prediction fidelity, which is the pitfall we discuss in Appendix G.2. A WorldArena-style benchmark for latent-space world models would be a valuable contribution, but does not yet exist. We have added a discussion of this point in the paper.
>
> ### OOD trajectory evaluation
>
> **Our datasets span a wide range of data quality regimes.** The training data in our experiments is far from uniformly optimal: Maze uses fully random trajectories; Metaworld data comes from the first iterations of TD-MPC2 RL training (progressively increasing quality); Push-T uses noised expert demonstrations (as in DINO-WM); DROID consists of real-world human teleoperation with inherently varying skill levels. The Maze results (over 80% success from random data, Table 3) are a strong signal of robustness to sub-optimal distributions. This is consistent with the finding of PLDM that latent-space world models are effective at "stitching suboptimal trajectories," which we discuss in Section 4. We also observe and discuss an OOD challenge on DROID (Appendix G.2, Figure 18): the CEM planner can sample actions outside the training distribution, degrading plan accuracy, which we mitigate via action clipping.
>
> **On Parthasarathy et al. [2].** The reviewer's suggestion (evaluating on held-out sub-optimal trajectories) is an evaluation protocol. A recent concurrent paper [2] addresses a related problem from the training side: they propose two techniques to close the train-test gap for gradient-based planning (GBP). "Online World Modeling" finetunes the world model on GBP trajectories corrected by a ground-truth simulator (DAgger-style, requiring simulator access). "Adversarial World Modeling" is simulator-free: it applies FGSM perturbations to latent states and actions during training, smoothing the loss landscape. Both yield large improvements for GBP on PushT, PointMaze, and Wall, matching or surpassing CEM.
>
> Crucially, both methods target the train-test gap specific to GBP, where the planner differentiates through the world model and can exploit inaccuracies. Our primary planner is CEM (sampling-based), which is inherently more robust: it evaluates many candidate action sequences without relying on world model gradients. Our planner comparison (Section 4, Figure 3a) confirms that CEM consistently outperforms GBP, so the OOD vulnerability analyzed by [3] is less acute in our setting. That said, their Adversarial World Modeling could be combined with our recipe to improve GBP performance. We mention this complementarity in the revised paper (Extended Related Work).
>
> [1] Shang, Yu, et al. "WorldArena: A Unified Benchmark for Evaluating Perception and Functional Utility of Embodied World Models." arXiv preprint arXiv:2602.08971 (2026).
>
> [2] Parthasarathy, et al. "Closing the Train-Test Gap in World Models for Gradient-Based Planning." arXiv preprint arXiv:2512.09929, 2025.

---

> > ### Author Response · Authors · 2026-03-11
> > **Rebuttal response 3/n**
> >
> > ## Change 3: Discussion on key design choices
> > We already added significant discussion about model scaling, maximum context size, and the new data scaling results in the revised version (see the updated paragraphs on these three topics in Section 5.1, and the new Table 3 summarizing the recommended recipe per task category).
> >
> > We revised the **multistep rollout training** paragraph in Section 5.1 to provide a principled explanation of the observed behavior. The revised text introduces the accuracy-robustness tradeoff: increasing the number of rollout steps $K$ raises $\delta_K$ (the one-step error on groundtruth inputs) but reduces the effective Lipschitz constant $\Lambda_K$. In simulated environments, the accuracy term dominates and the optimum lies at small $K$; on DROID, reducing $\Lambda_K$ more than compensates the increase in $\delta_K$, shifting the optimum to $K{=}6$. This analysis is backed by a new appendix section (Appendix D, "Error propagation in autoregressive latent prediction") that formalizes error propagation in three propositions: (1) an upper bound showing $\mathcal{O}(\Lambda^H)$ exponential error growth with the horizon, (2) a tightness result proving the bound is achievable, and (3) a generalization to the sliding-window case with a companion-matrix spectral-radius analysis. We conclude with Remark 1 on the $\delta_K$–$\Lambda_K$ tradeoff, connecting the formal analysis to our empirical findings.
> >
> > In addition, we **deepen the discussion of the impact of the encoder type and of using proprioception** on the predictor training and planning success in the main body of the paper.
> >
> > We stress that the paper, including in its original version, already provides principled and actionable guidelines beyond reporting empirical numbers. We enumerate the main ones below:
> > 1. Prefer image encoders with fine-grained spatial features over video encoders. DINOv2's superior object segmentation capabilities directly benefit the tasks studied, which require precise localization of the agent and objects (Section 5.1, encoder type).
> > 2. At training time, three frames allow inferring acceleration; longer context yields diminishing returns on datasets where trajectories are not long enough as it reduces the number of unique training slices. (Revised version: we added a detailed analysis of this trade-off, including the dataset coverage impact of increasing $W$.)
> > 3. Two-step rollout training is the sweet spot for simulated environments. Going beyond introduces diminishing returns, as the model specializes for a rollout regime it will less crucial for success at planning time (Section 5.1). Yet, for real-world data like DROID, dynamics is more complex and noisier, hence training with more rollout steps acts as data augmentation against compounding error, shifting the optimal number of rollout steps to 6.
> > 4. AdaLN conditioning provides the most consistent action information flow. We provide a principled analysis via the "action ratio" concept and explain why per-layer modulation prevents the vanishing of action information through depth, unlike input-only conditioning (Section 5.1, Appendix).
> > 5. CEM with L2 is the best planner; gradient-based planners fail due to local minima. We identify two concrete failure modes (border attraction and wall-bumping, Appendix Figure 10) and explain why the NG planner is a practical alternative for new tasks due to lower hyperparameter sensitivity (Conclusion).
> > 6. Scale model capacity only when the environment demands it. (Revised version: we added three complementary hypotheses explaining this, covering task saturation, harder planning landscape in higher dimensions, and fewer gradient updates per parameter.)
> >
> > We believe these guidelines offer practitioners a principled understanding of why each design choice matters, not just which performs best.

---

> > > ### Author Response · Authors · 2026-03-11
> > > **Rebuttal response 4/n**
> > >
> > > ## Change 4: Encoder adaptation
> > > We thank the reviewer for the insightful point regarding the assumption that DINO captures all essential representations. We entirely agree: while internet-scale Visual Foundation Models (VFMs) provide powerful semantic and physical priors, they are not inherently task-aligned for specific control dynamics.In the revised version of our manuscript, we have added a discussion comparing the limits of frozen backbones with emerging adaptation methodologies in the related work section. Specifically:
> > > 1. Joint Learning Paradigms: We note that models like PLDM (Sobal et al., 2025) and EB-JEPA (Terver et al., 2026) demonstrate that jointly learning task-specific representations from scratch is highly effective for navigation and trajectory stitching. This is particularly true in environments where internet-scale priors are less critical (e.g., as evaluated in their 2D simple navigation tasks).
> > > 2. Task-Aligned Adaptation: We have incorporated a discussion of recent works that adapt frozen backbones for control.
> > >    - Toso et al., 2026 augment the predictive objective with a bisimulation encoder to enforce control-relevant state equivalence. This mechanism filters out "slow features", i.e. task-irrelevant visual variations like background changes, to ensure that states with identical transition dynamics map to the same latent coordinates.
> > >    - Sparse World Models (Zhao et al. 2026), relies on overcomplete sparse autoencoders to translate dense embeddings into a discrete, selectively active vocabulary. This mechanism aligns dense features with interpretable task subspaces, providing parts-based decompositions that improve planning efficiency and robustness against environmental noise.
> > >    - DDP-WM (Yin et al., 2026) decouples latent updates into high-frequency "primary dynamics" and low-rank background adjustments through a cross-attention architecture. By allocating computational resources commensurate with the nature of the dynamics, it achieves a 9x inference speedup and significantly higher planning success.
> > > 3. Clarification on LDA-1B (Lyu et al., 2026): Regarding the specific citation provided by the reviewer, we respectfully note that LDA-1B actually maintains a strictly frozen DINO encoder throughout its pre-training process. The authors unfreeze the VLM during fine-tuning, but DINO remains fixed. In fact, the LDA-1B authors explicitly list their reliance on "fixed DINO visual features" as a limitation of their work. However, we completely agree with the reviewer's underlying premise that moving beyond frozen representations is a critical next frontier.
> > >
> > > Our current study deliberately focuses on the efficiency of dynamics learning atop frozen pre-trained encoders, as their acquired priors are highly beneficial for real-world planning tasks. Looking forward, we envision that general-purpose JEPA-WMs should train large-scale encoders and predictors jointly, maintain these core components frozen to preserve these broad priors, and then allow for rapid task adaptation through the fine-tuning or training of shallow peripheral networks.
> > >
> > > [1] Zhao et al. Sparse World Models: Visual World Modeling with Sparse Representations. 2026. https://openreview.net/forum?id=e0mUayPl40
> > >
> > > [2] Toso et al. Learning Invariant Visual Representations for Planning with Joint-Embedding Predictive World Models. arXiv preprint arXiv:2602.18639 (2026).
> > >
> > > [3] Yin et al. DDP-WM: Disentangled Dynamics Prediction for Efficient World Models. 2026. preprint arXiv:2602.01780 (2026).

---

### Author Response · Authors · 2026-03-13
**Rebuttal submitted**

Dear reviewers,

We would like to thank for your thorough and valuable feedback. We just submitted our rebuttal responses to all your reviews, striving our best to answer your concerns and propose valuable discussion.
Looking forward to reading your answers and having an emulating discussion.

Best,

---

### Decision · Action_Editor_S2E6 · 2026-05-07

**Recommendation:** Accept as is

**Additional Comments:**

While maybe the framing is not exactly as this, I feel this work fits the profile of an amazing reproducibility paper. It is a large scale empirical evaluation of the different component used to construct JEPA-WMs and provides recipe of how to best combine these components. I see the paper as a detailed ablation study, and I think the reproducibility certification fits.

**Audience:**

Yes

**Audience Explanation:**

The breath of experimentation makes this paper valuable to the community and can act as an anchor point or baseline for future exploration of world models (under the JEPA formalism or not).

**Claims And Evidence:**

Yes

**Claims Explanation:**

The reviewers agree the work presents careful and extensive empirical evaluation of Jepa-WM providing an analysis of which components of these architectures make them work.